# Genetic variants in UNC93B1 predispose to childhood-onset systemic lupus erythematosus

Mahmoud Al-Azab[1,2,13], Elina Idiiatullina [1,3,13], Ziyang Liu [1], Meng Lin[1], Katja Hrovat-Schaale[4,5], Huifang Xian[1], Jianheng Zhu[1], Mandy Yang[6], Bingtai Lu[1], Zhiyao Zhao[1,7], Yiyi Liu[1], Jingjie Chang[1], Xiaotian Li[1], Caiqin Guo[1], Yunfeng Liu[8], Qi Wu[1,9], Jiazhang Chen[1], Chaoting Lan[1], Ping Zeng[1], Jun Cui [10], Xia Gao[1], Wenhao Zhou[1], Yan Zhang[1], Yuxia Zhang [1,14] ✉ & Seth L. Masters [1,4,5,11,12,14] ✉

Rare genetic variants in toll-like receptor 7 (TLR7) are known to cause lupus in humans and mice. UNC93B1 is a transmembrane protein that regulates TLR7 localization into endosomes. In the present study, we identify two new variants in UNC93B1 (T314A, located proximally to the TLR7 transmembrane domain, and V117L) in a cohort of east Asian patients with childhood-onset systemic lupus erythematosus. The V117L variant was associated with increased expression of type I interferons and NF-κB-dependent cytokines in patient plasma and immortalized B cells. THP-1 cells expressing the variant UNC93B1 alleles exhibited exaggerated responses to stimulation of TLR7/-8, but not TLR3 or TLR9, which could be inhibited by targeting the downstream signaling molecules, IRAK1/-4. Heterozygous mice expressing the orthologous Unc93b1[V117L] variant developed a spontaneous lupus-like disease that was more severe in homozygotes and again hyperresponsive to TLR7 stimulation. Together, this work formally identifies genetic variants in UNC93B1 that can predispose to childhood-onset systemic lupus erythematosus.

Systemic lupus erythematosus (SLE) is a chronic autoimmune disease that typically develops in adults but can affect around 1 in 100,000 children[1]. There is a significant genetic contribution to the condition, ranging from common variants with small effects, through to fully penetrant disease-causing alleles[2]. One example of this is TLR7, which marks a genetic interval that is a risk factor for developing SLE[3], but can also drive a monogenic form of the disease as a result of gain-of-function mutations[4]. TLR7 typically functions as a sensor of viral single-stranded (ss)RNA in endosomes, to which it is trafficked by the transmembrane protein UNC93B1 (ref. 5). TLR7 then signals through Myd88 to interleukin-1 receptor-associated kinase 1/-4 (IRAK1/-4), leading to nuclear factor κ-light-chain-enhancer of activated B cells (NF-κB) and type I interferon (IFN) expression programs that are typically associated with SLE[6], but critically are required to fight viral infection. Consequently, people with loss-of-function mutations in TLR7 or UNC93B1 are immunodeficient[7,8]. On the other hand, again similar to TLR7, UNC93B1 expression is increased in patients with SLE and active disease[9], and mutations in murine UNC93B1 can cause a lupus-like disease[10,11]. There is also a sporadic lupus-like disease in dogs that is the result of a mutation in UNC93B1 (ref. 12). Despite all of these compelling data linking UNC93B1 to disease, there was still no genetic variant in the human population that formally validated its role in SLE pathogenesis.

## Rare UNC93B1 variants in patients with childhood-onset SLE

Given the recent observation that rare genetic variants in TLR7 can cause childhood-onset SLE[4], we searched in this patient population for

**Fig. 1 | Conservation, structural location and geographical distribution of UNC93B1 SLE variants. a**, Amino acid conservation across five species, as indicated, for residues surrounding the variants of interest in UNC93B1. **b**, UNC93B1 (Protein Data Bank (PDB) 7CYN) shown in light green and green (surface and cartoon representations) with Thr314 shown as blue sticks on

H2. TLR7 protomers are shown in purple and magenta (surface and cartoon representations). Val117 and Leu117 are shown as red sticks between the interface of two UNC93B1 protomers. The structural analysis software used was Pymol v.2.5.8. **c**, UNC93B1 V117L variant local geographical distribution.

variants that may influence the interaction between TLR7 and the transmembrane (TM) protein that regulates its cellular localization, UNC93B1 (ref. 13). We identified a new variant in UNC93B1 (c.A940G p.T314A) using whole-exome sequencing (WES) in patient 1 (P1), which is not present in the general population and is highly conserved (Fig. 1a). Based on a published structure of UNC93B1 in complex with TLR7 (ref. 13), the affected amino acid is at the start of helix 2 (H2) between TM helices TM6 and TM7 of UNC93B1 (Fig. 1b). H2 is positioned on top of the TM helix of TLR7 responsible for the UNC93B1–TLR7 interaction. In addition, Thr314 is in close proximity to the carboxy terminus of the protein, a disordered region that has been shown to have two phosphorylation sites (Ser547 and Ser550) that regulate TLR7 activation[11]. Looking more broadly in a cohort of 272 patients with lupus recruited at the Guangzhou Women's and Children's Medical Centre, we found that seven (P2–P8) encode UNC93B1 (c.G349T p.V117L), a highly conserved variant (Fig. 1a). Structurally, this residue lies at the interface between UNC93B1 subunits (protomers) in the published dimeric assembly (Fig. 1b). Although these protomers are close to each other, there are

no noticeable interactions that would prevent dimerization of TLR7, similar to what occurs for TLR3 (ref. 13). Instead, V117L may act in a similar way to K333R, which is also located at an UNC93B1 protomer interface and results in increased TLR7 activation. However, as Lys333 is ubiquitinated, and not very close to V117L, this would represent a different mechanism of activation[11]. UNC93B1 V117L variant is present in the general east Asian population and is most prevalent in south coast Han individuals (Fig. 1c)[14]. We calculate that it confers a 17.9-fold increased risk of developing childhood-onset SLE (Supplementary Table 1). For all patients with SLE identified as carrying this allele, there were no immediate reports of affected family members. The presence of heterozygous UNC93B1 V117L was confirmed for the father of P2 and the mother of P3, both of whom are currently unaffected. The two UNC93B1 variants were nominated as potential effectors for systemic inflammation in our subjects according to the well-known biological role of UNC93B1 in autoimmunity[15], predicted pathogenicity scores (Supplementary Table 2) and lack of other likely pathogenic variants found in primary immunodeficiency genes based on another early onset lupus cohort[16].

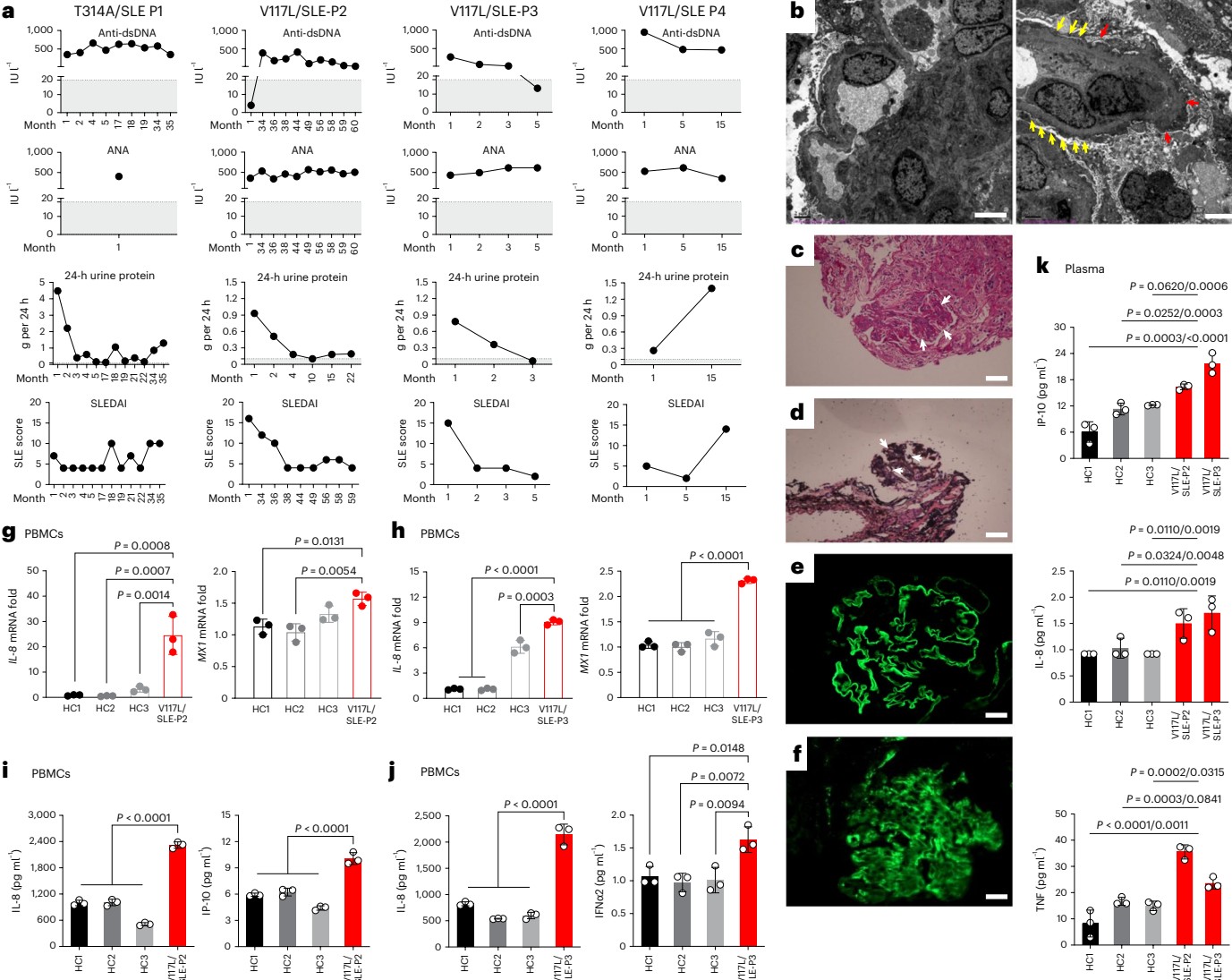

**Fig. 2 | Clinical characteristics related to UNC93B1 T314A and V117L variants. a**, Anti-dsDNA autoantibodies (normal range (NR), <18 IU l⁻¹), ANA autoantibodies (NR, <18 IU l⁻¹), 24-h urinary protein (NR, 0–0.1 g per 24 h) and SLEDAI score of disease for P1 (T314A), P2, P3 and P4 (all V117L). Data are for monthly visits postdiagnosis. NR is highlighted in gray. **b–f**, Kidney pathology for patient with UNC93B1 T314A. **b**, Transmission electron microscopy showing two glomeruli, capillary endothelial cell proliferation, vacuolar degeneration, RBCs, monocytes and neutrophil aggregation in little vascular loops. The podocyte foot process has diffuse effacement (yellow arrows), with the cell swollen and cavitated. The basement membrane has diffuse thickening up to 1,300 nm (red arrows), P1. Scale bar, 4 μm. **c**, Periodic acid−Schiff (PAS) stain showing thickening of the basement membrane and mesangial and endothelial cell proliferation (white arrows), P1. Scale bar, 50 μm. **d**, PAS methenamine (PASM) stain showing thickening of the basement membrane and a small number of platinum ear-like structures (white arrows), P1. Scale bar, 50 μm.

**e**, Immunofluorescence for collagen IV of the glomerular basement membrane showing α3+ve, α5+ve, P1. Scale bar, 20 μm. **f**, Immunofluorescence of kidney tissue showing complement 3 deposition, P1. Scale bar, 20 μm. **g**, RT−qPCR analysis of *IL-8* and *MX1* mRNA expression in the PBMCs of P2 compared with healthy controls. **h**, RT−qPCR analysis of *IL-8* and *MX1* mRNA expression in the PBMCs of P3 compared with healthy controls. **i**, Production of IL-8 and IP-10 in the supernatant of PBMCs isolated from P2 and healthy controls measured by CBA. **j**, Production of IL-8 and IFNα2 in the supernatant of PBMCs isolated from P3 and healthy controls measured by CBA. PBMCs in **g**–**j** were isolated from patients' whole blood and incubated in media for 12 h before analysis without stimulation. **k**, Levels of IP-10, IL-8 and TNF in the plasma from P2 and P3 compared with healthy controls measured by CBA. For **g**–**k**, n = 3 biological samples; indicated P values were determined by two-way ANOVA, multiple comparisons, P_adj value; data are presented as mean with s.d.

Clinical parameters for patients encoding UNC93B1 T314A (P1) and V117L (P2–P7) were available. These include elevated titers of serum autoantibodies to double-stranded DNA (anti-dsDNA) and anti-nuclear antibodies (ANAs), with increased protein in 24 h of collected urine and depleted serum complement 3/4 (Fig. 2a and Extended Data Figs. 1a–g and 2a–f). Comprehensive electron microscopy, light microscopy and immunofluorescence studies for kidney tissues of P1 are consistent with diffuse proliferative lupus nephritis with membranous lupus nephritis, IV −G (A) +V. The mesangial cells are aggregated. Electron densification was deposited in stromal hyperplasia,

in the subepithelial, intrabasement, subendothelial and mesangial zones. In the tubulointerstitium, epithelial cells were cavitated with a small amount of renal tubular atrophy and inflammatory cells infiltrated into the renal interstitium. In renal interstitial vessels, there was red blood cell (RBC) aggregation in the loops of little capillaries and thickening of the arteriole wall (Fig. 2b−f). The complete blood count from noncoagulated whole blood for P1−P3 and P5−P7 was generally normal, with periods of leukocytosis (P1−P3 and P5), neutrophilia and monocytosis, corresponding to periods of lymphocytopenia (Extended Data Fig. 1a−c,e−g). The overall disease score was clinically

evaluated using SLEDAI and SLEDAI-2k scoring systems, in which results ranged from 2 to 20 (Fig. 2a and Extended Data Fig. 1a–d,f). Other clinical parameters and treatments are reported in Supplementary Table 3. We isolated peripheral blood mononuclear cells (PBMCs) from patients with UNC93B1[V117L], P2 and P3, using a Ficoll–Hypaque density gradient separation method and found spontaneous expression of *interleukin (IL)-8, MX1* and *IFN-stimulated gene 15 (ISG-15)* genes in P2 (Fig. 2g and Extended Data Fig. 3a), and *IL-8, MX1, IFN-induced proteins with tetratricopeptide repeat (IFIT)1, IFIT3* and *ISG-15* in P3 (Fig. 2h and Extended Data Fig. 3b). Indeed, we observed induced secretion of typical TLR7-induced cytokines, IL-8, IP-10, IFNγ, IFNλ1/2/3, IL-12p70, IL-10, IFNα, IFNβ and IL-6, in P2 (Fig. 2i and Extended Data Fig. 3c,d), and IL-8, IFNα2, IFNγ, IFNλ1/2/3, IL-12p70 and IL-10 in P3 (Fig. 2j and Extended Data Fig. 3e). Furthermore, we detected elevated levels of IP-10, IL-8 and tumor necrosis factor (TNF) in the plasma of P2 and P3 (Fig. 2k) and IFNα2, IFNγ, IFNλ1/2/3, IL-6, IL-12p70, IL-1β, granulocyte–macrophage colony-stimulating factor (GM-CSF) and IL-10 in plasma of P2 (Extended Data Fig. 3f). These data identify rare UNC93B1 variants in patients with childhood-onset SLE, associated with typical clinical parameters and elevated levels of lupus-associated cytokines, similar to patients with mutations in TLR7 (ref. 4).

## UNC93B1 variants drive inflammation via TLR7/-8 and IRAK1/-4

To study the direct effect of these rare UNC93B1 variants, we created an in vitro model with THP-1 monocytes overexpressing wild-type (WT), V117L or T314A using UNC93B1 lentivirus constructs. The patient mutations trigger spontaneous upregulation of transcripts for *IFNβ* and the ISGs, *IFIT3, ISG-15* and *ISG-20L2* (Fig. 3a and Extended Data Fig. 4a). Upregulated messenger RNA levels of inflammatory cytokines, *IL-8, IL-12a* and *TNF*, were also observed (Extended Data Fig. 4a). In addition, the mutated UNC93B1-induced phosphorylation of IFN regulatory factor (IRF)5, NF-κB and MAPK (JNK and P38) is shown by immunoblotting (Fig. 3b). Overexpression of UNC93B1 V117L and T314A resulted in excess secretion of IFNα, IFNβ and IL-6 as detected by ELISA (Fig. 3c) and IL-8 and IP-10 as detected by cytokine bead array (CBA) (Extended Data Fig. 4b). For an unbiased comparison, we performed RNA sequencing (RNA-seq) in THP-1 cells overexpressing UNC93B1 V117L or T314A. From the most significantly upregulated genes, at least six are known biomarkers for disease activity in lupus (*DEFB1* (ref. 17), *PRLR* (ref. 18), *S100A8/-9* (ref. 19), *FCER2* (ref. 20) and PRG2 (ref. 21)) (Fig. 3d,e). Further gene set and pathway analysis pointed toward programs related to innate immune response, phagosome activity and antigen processing and presentation (Extended Data Figs. 4c–f and 5a,b). Owing to the core role of B cells in the pathogenesis of SLE[22], we generated immortalized B cell lines from patients with UNC93B1[V117L] variants (P2 and P3) and healthy controls. At baseline, B cell lines from the patients revealed induced expression of *IL-6, IL-8, TNF, IL-12a* and *ISG-20L2* genes compared with healthy controls (Fig. 3f and Extended Data Fig. 5c). Increased secretion of IL-6, IL-8, IL-12p70, IL-10, TNF and GM-CSF in the supernatant of the unstimulated B cells from patients with the UNC93B1 V117L variant was also observed (Fig. 3g and Extended Data Fig. 5d,e). Collectively, these results indicate that the UNC93B1 variants identified can intrinsically promote inflammation and immune dysfunction associated with type I IFN and NF-κB signaling pathways.

So far, the main function for UNC93B1 is to regulate trafficking and localization of nucleic acid-sensing TLRs (NAS-TLRs), TLR3, TLR7, TLR8 and TLR9 (refs. 5,23,24). Thus, we next investigated which could be involved in this phenotype by stimulating patients' PBMCs using poly(I:C), R848 and CPG-C, synthetic ligands for TLR3, TLR7/-8 and TLR9, respectively. The results revealed that patient PBMCs stimulated by R848 showed induced expression of *IFIT1* and *TNF* compared with healthy controls, but not when stimulated by poly(I:C) and CPG-C (Fig. 4a). In addition, secretion of TNF, IL-1β and IL-6 in the supernatant of PBMCs from P2 were increased specifically after stimulation by R848,

but not poly(I:C) or CPG-C (Fig. 4b and Extended Data Fig. 6a). These results were independently confirmed using PBMCs of P3 (Fig. 4c and Extended Data Fig. 6b). As R848 is a dual TLR7 and TLR8 synthetic agonist, we next stimulated PBMCs, THP-1 and/or immortalized B cell lines with a variety of specific agonists for TLR7 (guanosine, loxoribine and/or R837), or a specific agonist of TLR8 (TL8-506). Guanosine and TL8-506-stimulated PBMCs reveal that both upregulate IL-6, IL-8, IL-1β, IL-10, IL-12p70 and TNF in patients compared with healthy controls (Fig. 4d and Extended Data Fig. 6c). Although stimulated THP-1 showed the specific upregulation of the TLR7 pathway (guanosine and R848) but not TLR3 (poly(I:C)) or TLR9 (CPG-C) pathways, stimulation by TL8-506 showed induced IL-18 production in V117L and T314A cells, and IL-1β, IL-23 and IL-10 in V117L but not T314A cells (Fig. 4e and Extended Data Fig. 6d). Stimulation of UNC93B1[V117L]-immortalized B cells also demonstrates upregulation of the TLR7 pathway, compared with healthy control cells, whether as a result of dual agonist, R848- or TLR7-specific agonists, guanosine, loxoribine or R837 stimulation. But no obvious stimulation was achieved by the TLR8-specific agonist, TL8-506, except a slight increase in some but not all tested cytokines, because human B cells are not as naturally predominant in expressing TLR8 as myeloid cells[25,26] (Fig. 4f and Extended Data Fig. 6e). Gene variants, V117L and T314A, did not affect the phosphorylation of the UNC93B1 protein (Supplementary Fig. 1), suggesting the presence of another mechanism involved in the upregulation of TLR7/-8 signaling.

Some of the most attractive therapeutic targets downstream of UNC93B1 and the NAS-TLRs are the signaling adapter kinases IRAK1 and IRAK4 (ref. 27). To start, we found that the phosphorylation of IRAK4 was increased owing to overexpression of mutated UNC93B1 (Extended Data Fig. 7a). Transcriptionally, IRAK4 inhibition using zimlovisertib (PF-06650833), a potent selective inhibitor, was efficacious, returning THP-1 R848-induced *TNF, IL-6, IL-8* and the ISGs *IFIT1, IFIT3, ISG-15* and *ISG-54* to baseline in a dose-dependent fashion (Fig. 5a and Extended Data Fig. 7b). In agreement, the downstream cytokines produced by UNC93B1 V117L and T314A R848-stimulated Thp-1 cells (for example, IL-6, IL-8, IL-23, IL-1β and IFNα2) were significantly blunted in a dose-dependent pattern (Fig. 5b and Extended Data Fig. 7c). The pathway was confirmed by immunoblotting cell lysates, where R848-induced phosphorylation of NF-κB and MAPK (ERK and P38) was also downregulated as a result of IRAK4 inhibition dose dependently (Fig. 5c). We also tested a dual targeting IRAK1/-4 inhibitor, a new benzimidazole, which was efficacious in downregulating *ISG-4, TNF, IL-6* and *IL8* transcription in R848-stimulated Thp-1 cells in a dose-dependent fashion (Fig. 5d and Extended Data Fig. 7d). Consistently, production of IL-6, IL-8, IL-23 and monocyte chemoattractant protein 1 (MCP-1) was also downregulated by the IRAK1/-4 inhibitor (Fig. 5e and Extended Data Fig. 7e). We were able to confirm these results in patient cells in which we found that the IRAK4 inhibitor reduced IL-6, TNF, IL-12p70, IFNγ and IL-10 to baseline in UNC93B1 V117L loxoribine-stimulated immortalized B cells, in a dose-dependent manner (Fig. 5f and Extended Data Fig. 7f). The IRAK4 inhibitor was also efficacious in returning IL-6, IL-8 and MCP-1 in P3 R848-stimulated PBMCs to the baseline (Fig. 5g and Extended Data Fig. 7g). In addition, UNC93B1 V117L loxoribine-stimulated, immortalized B cells experienced suppressed IFNγ and IL-8 owing to the IRAK1/-4 inhibitor (Fig. 5h). Overall, these findings point toward IRAK1/-4 as the dominant signaling pathway triggered by variants in UNC93B1 found in patients with childhood-onset SLE. Taken together, genetic variation in UNC93B1 regulates inflammation in patients with childhood-onset SLE through a TLR7/-8–IRAK1/-4 pathway.

## Lupus-like disease in mice with the mutation UNC93B1[V117L]

Given that UNC93B1[V117L] is a highly significant risk factor for childhood-onset SLE, but also present in the general population, we sought to confirm pathogenicity in vivo and created mice with the orthologous

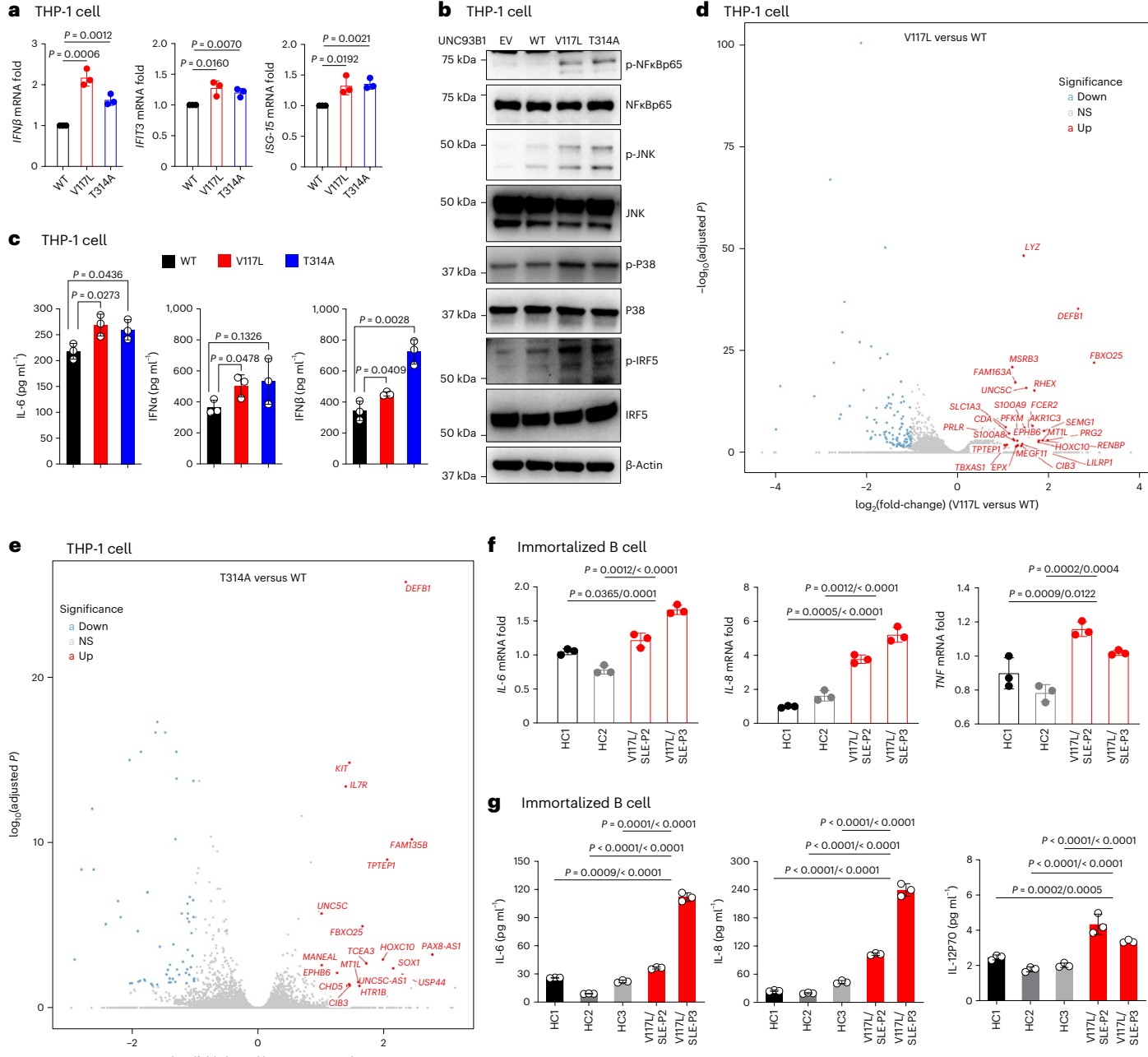

**Fig. 3 | V117L and T314A UNC93B1 variants spontaneously induce IFN and NF-κB signaling pathways. a**, RT–qPCR analysis of *IFNβ*, *IFIT3* and *ISG-15* expression in the indicated THP-1 cell lines (*n* = 3 biological replicates). There were three independent experiments. **b**, Levels of phosphorylated NF-κB, JNK, p38 and IRF5, as measured by immunoblotting, in lysates of the indicated THP-1 cells. Data are representative of three independent experiments. **c**, Production of IL-6, IFNα and IFNβ in the indicated THP-1 cell lines (*n* = 3, three independent experiments measured by ELISA). The indicated *P* values in **a** and **c** were determined using unpaired, two-tailed Student's *t*-test; the data are presented as the mean with s.d.

**d**,**e**, RNA-seq was performed for UNC93B1^V117L (**d**) and UNC93B1^T314A (**e**) THP-1 cells compared with WT. DEGs are presented as a volcano plot and mRNA was extracted from THP-1 of UNC93B1^WT, UNC93B1^V117L and UNC93B1^T314A (*n* = 3 biological replicates). **f**, RT–qPCR analysis of *IL-6*, *IL-8* and *TNF* expression in the indicated immortalized B cell lines (baseline) (*n* = 3 biological replicates). **g**, Production of IL-6, IL-8 and IL-12p70 in the indicated immortalized B cell lines (baseline) (*n* = 3 biological replicates, measured by CBA). The indicated *P* values in **f** and **e** were determined by two-way ANOVA, multiple comparison, *P*_adj value; data are presented as mean with s.d.

mutation V117L. The mutation, UNC93B1^V117L, was introduced to the germline of mice using CRISPR (clustered regularly interspaced short palindromic repeats)–Cas9 genome editing technology (Extended Data Fig. 8). UNC93B1^V117L mice were born at normal Mendelian ratios and initially appear healthy; however, both heterozygous and homozygous mice lose weight over time (Fig. 6a), develop splenomegaly with increased spleen cellularity (Fig. 6b) and have a small kidney size (Fig. 6c). In addition, lupus-associated serum anti-dsDNA and

anti-Smith autoantibodies were increased in knock-in mice (Fig. 6d), but no change was noted for total immunoglobulin G (IgG) (Supplementary Fig. 2). Bone marrow (BM) populations of CD45⁺ immune cells, macrophages and CD11b⁺/Gr-1⁺ cells were increased along with CD16⁺/CD14⁻ cells, regulatory T cells and activated T cells (Fig. 6e). Functionally, the intracellular production of IFNγ in CD4⁺ T cells and CD3⁺/CD4⁻ cells of BM was also upregulated in the mutant mice (Fig. 6f). In the spleen of UNC93B1^V117L mice, there were increased B cells and

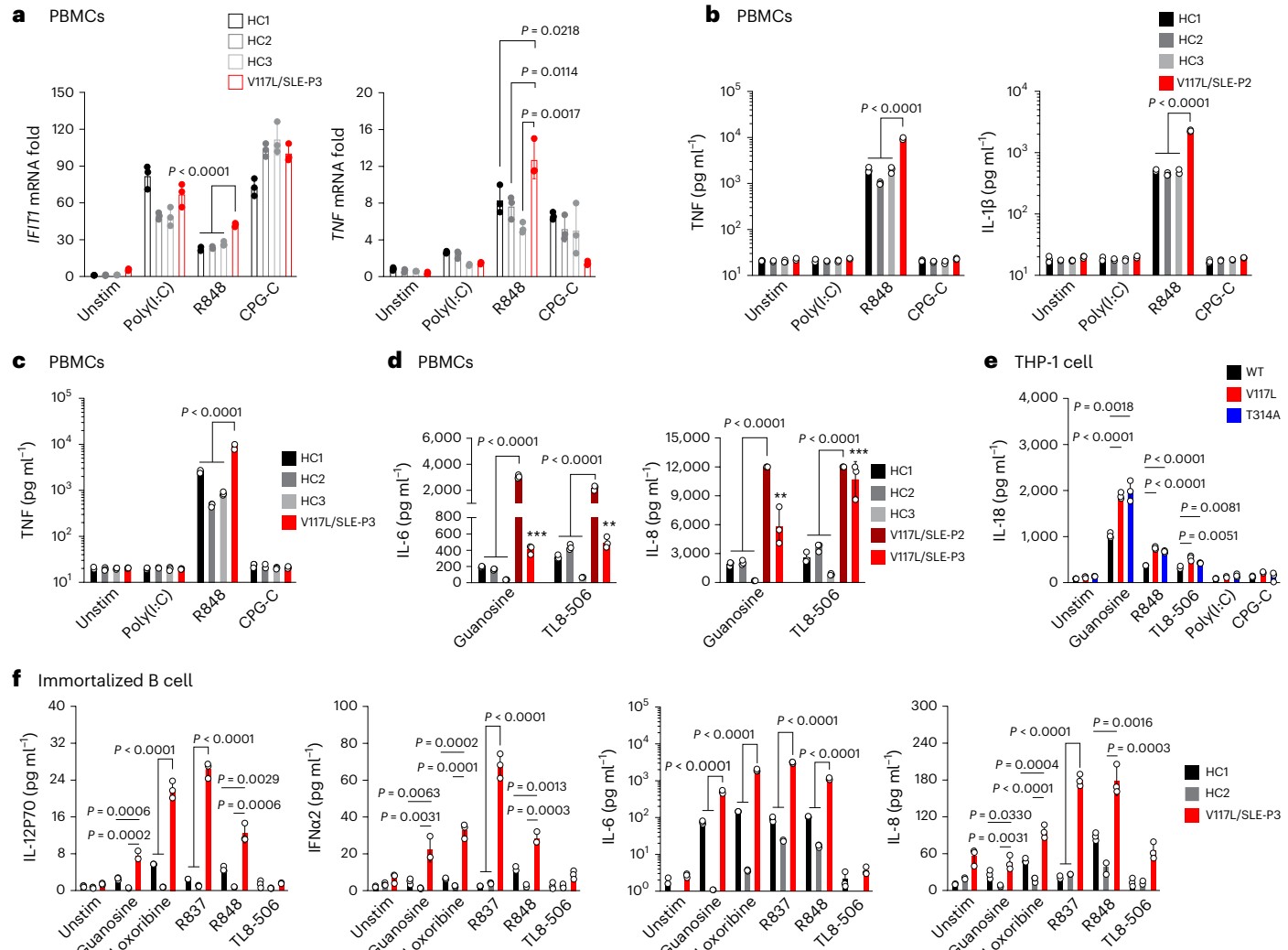

**Fig. 4 | UNC93B1 genetic variation drives inflammation via TLR7/-8. a**, RT–qPCR analysis of *IFIT1* and *TNF* mRNA expression in the PBMCs of P3 compared with healthy controls after stimulation by 10 μg ml⁻¹ of HMW poly(I:C), 1 μg ml⁻¹ of R848 and 4 μg ml⁻¹ of CPG-C for 12 h (*n* = 3 biological samples). Unstim, Unstimulated. **b**, Production of TNF and IL-1β in the supernatant of PBMCs of P2 compared with healthy controls after stimulation by 10 μg ml⁻¹ of HMW poly(I:C), 1 μg ml⁻¹ of R848 and 4 μg ml⁻¹ of CPG-C for 8 h (*n* = 3 biological samples). **c**, Production of TNF in the supernatant of PBMCs of P3 compared with healthy controls after stimulation by 10 μg ml⁻¹ of HMW poly(I:C), 1 μg ml⁻¹ of R848 and 4 μg ml⁻¹ of CPG-C for 12 h (*n* = 3 biological samples). **d**, Production of IL-6 and IL-8 in the supernatant of PBMCs of P2 and P3 compared with healthy controls after stimulation by 1 mM guanosine and 200 ng ml⁻¹ of TL8-506 for 24 h (*n* = 3 biological samples). The *P* value in IL-6, P3 (guanosine set), was 0.0001 compared with HC1 and <0.0001 with HC2 and HC3, and in the TL8-506 set it was 0.0034 compared with HC1 and <0.0001 with HC3. The *P* value in IL-8, P3 (guanosine

set), was 0.0053 compared with HC1, 0.0065 with HC2 and 0.0008 with HC3 and, in the TLR8-506 set, was 0.0001 compared with HC1, 0.0003 with HC2 and <0.0001 with HC3. The indicated *P* values in **a**–**d** were determined by two-way ANOVA, multiple comparisons, $P_{adj}$ value; data are presented as the mean with s.d. **e**, Production of IL-18 in the supernatant of THP-1 of indicated variants after stimulation by 0.7 mM guanosine, 1 μg ml⁻¹ of R848, 200 ng ml⁻¹ of TL8-506, 100 μg ml⁻¹ of HMW poly(I:C) and 5 μg ml⁻¹ of CPG-C for 24 h (*n* = 3 biological replicates); the indicated *P* values were determined by unpaired, two-tailed Student's *t*-test and data are presented as the mean with s.d. **f**, Production of IL-12p70, IFNα2, IL-6 and IL-8 in the supernatant of indicated immortalized B cell lines after stimulation by 0.5 mM guanosine, 0.1 mM loxoribine, 2.5 μg ml⁻¹ of R837, 1 μg ml⁻¹ of R848 and 200 ng ml⁻¹ of TL8-506 for 24 h (*n* = 3 biological replicates). The indicated *P* values in **f** were determined by two-way ANOVA, multiple comparisons, $P_{adj}$ value; data are presented as the mean with s.d. Production of cytokines in the cells supernatant was measured by CBA.

a decreased CD4 T cell:B cell ratio, along with elevated regulatory T cells, activated T cells, germinal center B cells, CD45⁺/CD19⁺/CD3⁻/CD95⁺/CXCR5⁺ cells and CD45⁺/CD4⁺/CD44⁺ cells (Fig. 6g). Serum cytokine analysis demonstrated that UNC93B1^V117L leads to upregulation of IL-12p70, IP-10, IL-6 and IFNγ (Fig. 7a). Histopathological studies were done for hematoxylin and eosin (H&E)-stained tissues sections from kidneys, spleens, lungs and pancreata. Histologically, there was overt pathology in the kidney (Fig. 7b), spleen (Fig. 7c), lung (Fig. 7d) and pancreas (Fig. 7e), where disease scores of mesangial expansion, extramedullary hematopoietic cell hyperplasia and inflammatory cell infiltrate, respectively, were elevated in mice with the UNC93B1^V117L

variant. Molecularly, increased mRNA of the ISGs, *MX1*, *ISG-20L2*, *IFIT1* and *IRF7* were observed in kidney tissue homogenate (Fig. 7f).

BM-derived macrophages (BMDMs) isolated from UNC93B1^V117L mice experienced increased mRNA of the ISGs, *IRF7* and *IFIT1*, when compared with BMDMs of UNC93B1^WT mice (Fig. 8a). Meanwhile, phosphorylation of the UNC93B1/TLR7 signaling pathway, including IRF5, IRF7, NF-κB and MAPK (JNK and P38), was also observed to be activated in UNC93B1^V117L BMDMs (Fig. 8b). In addition, UNC93B1^V117L BMDMs exhibit elevated intrinsic intracellular production of TNF (Extended Data Fig. 9a). Unbiased RNA-seq indicates that all of the most highly upregulated genes are known to be inducible by type I

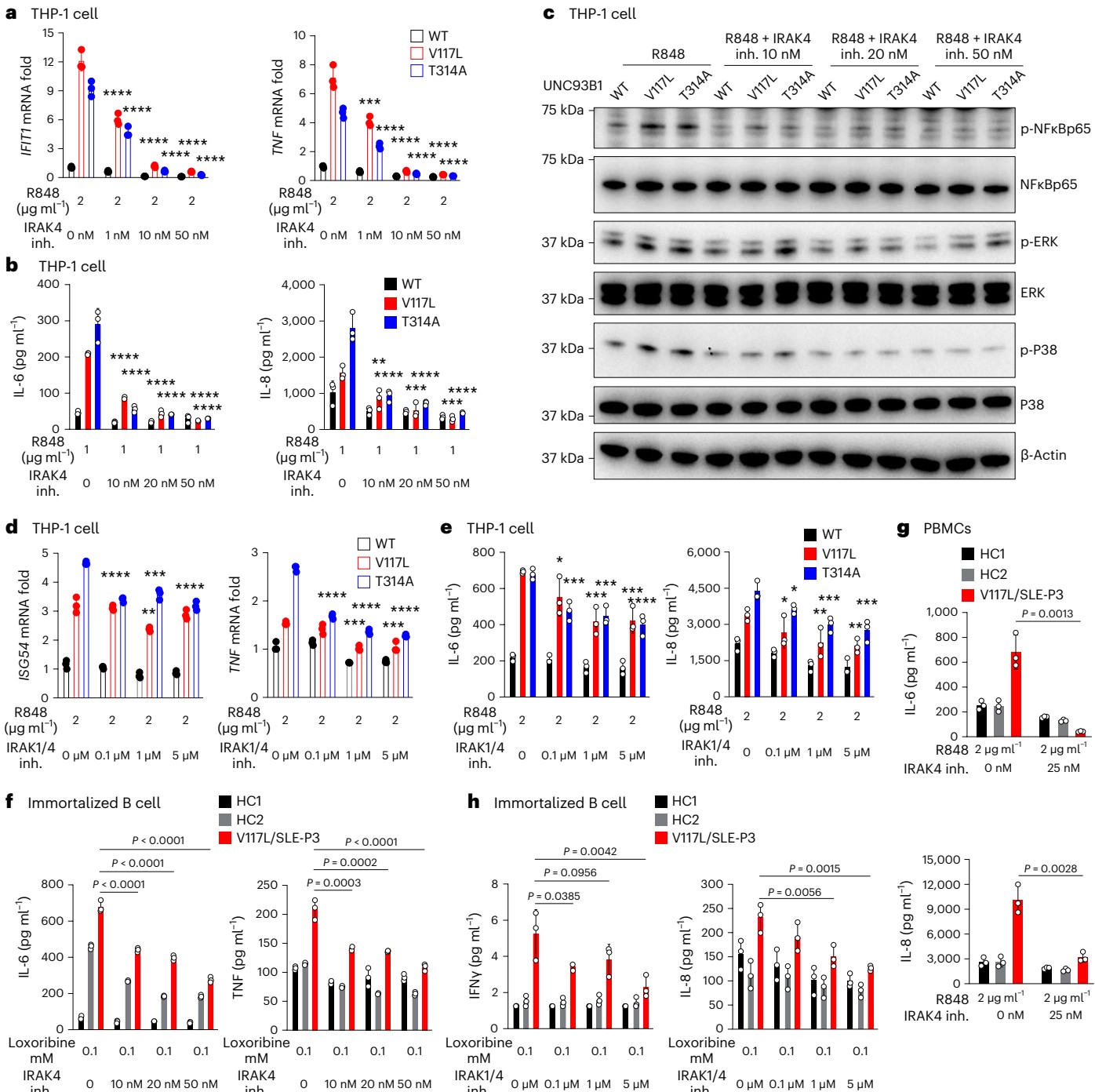

**Fig. 5 | UNC93B1 genetic variation drives inflammation via IRAK1/-4.**
**a**, RT–qPCR analysis of *IFIT1* and *TNF* expression in the indicated THP-1 cell lines, stimulated by R848 for 24 h after being incubated with IRAK4 inhibitor (inh.) for 30 min. ***P = 0.0002, ****P < 0.0001. **b**, Production of IL-6 and IL-8 in the supernatant of the indicated THP-1 cell lines, stimulated by R848 for 24 h after being incubated with IRAK4 inhibitor for 30 min. ****P < 0.0001, **P = 0.0021, ***P = 0.0003 (20 nM set), ***P = 0.0001 (50 nM set). **c**, Levels of phosphorylated NF-κB, ERK and P38, as measured by immunoblotting using lysates of the indicated THP-1 cells, stimulated by 1 μg ml⁻¹ of R848 for 24 h after being incubated with IRAK4 inhibitor for 30 min. Data are representative of three independent experiments. **d**, RT–qPCR analysis of *ISG54* and *TNF* expression in the indicated THP-1 cell lines, stimulated by R848 for 24 h after being incubated with IRAK1/-4 inhibitor for 30 min. In *ISG54*, **P = 0.0025, ***P = 0.0001, ****P < 0.0001, and in *TNF*, ***P = 0.0002, ***P = 0.0003, ****P < 0.0001, respectively. **e**, Production of IL-6 and IL-8 in the supernatant of the indicated THP-1 cell lines, stimulated by R848 for 24 h after being incubated with

IRAK1/-4 inhibitor for 30 min. In IL-6: *P = 0.0212, ***P = 0.0002, ***P = 0.0008, ***P = 0.0001, ***P = 0.0009, ****P < 0.0001, respectively, and in IL-8, *P = 0.0334, *P = 0.0107, **P = 0.0041, ***P = 0.0005, **P = 0.0024, ***P = 0.0002, respectively. **f**, Production of IL-6 and TNF in the supernatant of the indicated immortalized B cell lines, stimulated by loxoribine for 24 h after being incubated with IRAK4 inhibitor for 30 min. **g**, Production of IL-6 and IL-8 in the supernatant of the indicated PBMCs, stimulated by R848 for 24 h after being incubated with and without IRAK4 inhibitor for 30 min. The indicated *P* values were determined by unpaired, two-tailed Student's *t*-test (*n* = 3 biological samples). **h**, Production of IFNγ and IL-8 in the supernatant of the indicated immortalized B cell lines, stimulated by loxoribine for 24 h after being incubated with IRAK1/-4 inhibitor for 30 min. Production of cytokines in the cell supernatant was measured by CBA. For all experiments, data are presented as the mean with s.d. For **a**, **b**, **d**–**f** and **h**, *n* = 3 biological replicates; the indicated *P* values were determined by two-way ANOVA, multiple comparisons, $P_{adj}$ value.

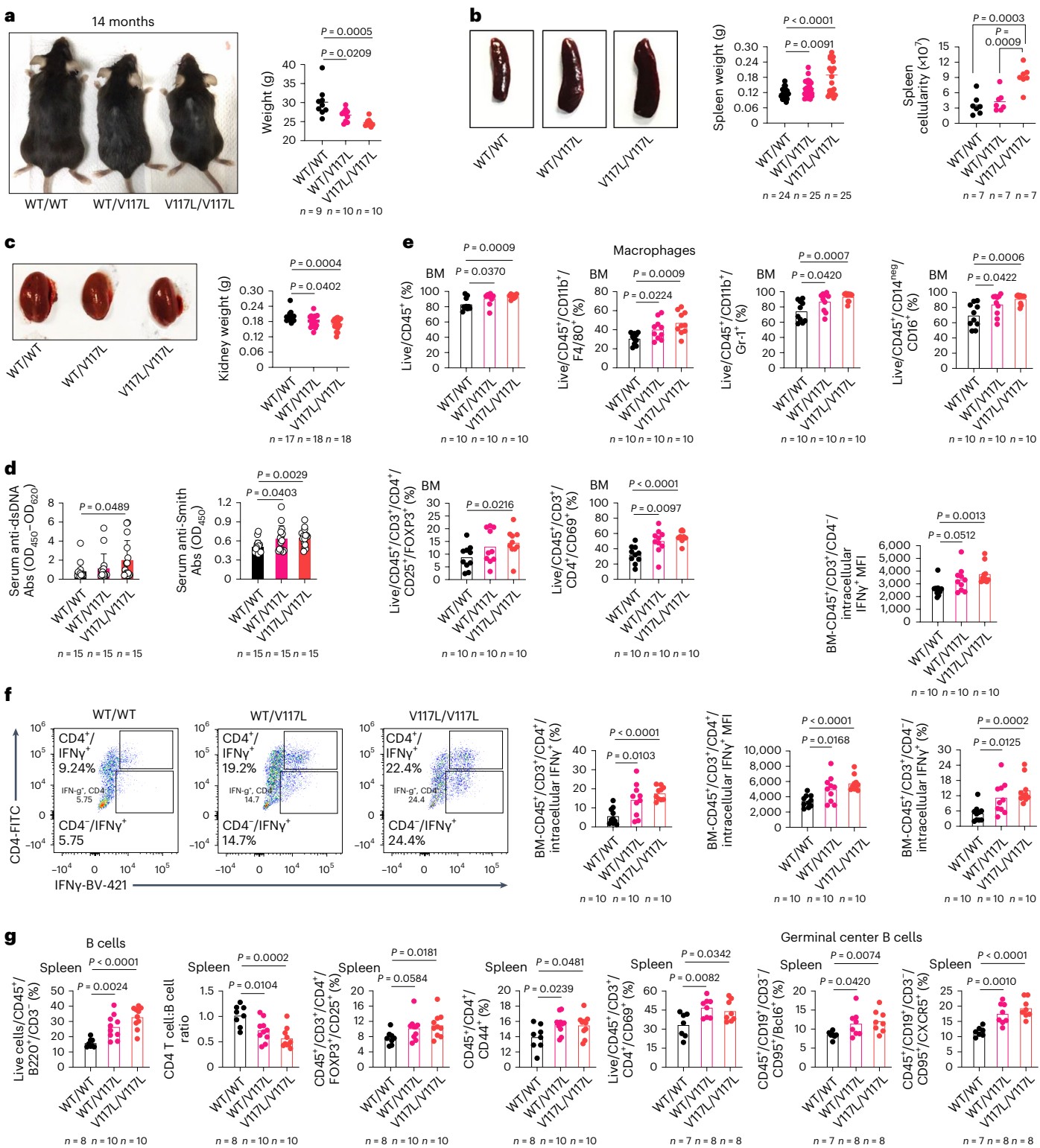

**Fig. 6 | UNC93B1^V117L mice develop lupus-like disease. a**, Appearance and weight of UNC93B1^WT/WT, UNC93B1^WT/V117L and UNC93B1^V117L/V117L mice. **b**, Spleen weight (data pooled from three independent experiments) and splenocyte count of indicated mice. Mice were age matched to 8–14 months. **c**, Kidney weight of indicated mice. Mice were age matched to 8–14 months and the data pooled from two independent experiments. **d**, Serum autoantibodies to dsDNA or Smith for the indicated mice. Mice were age matched to 8–14 months and the data pooled from three independent experiments. **e**, Flow cytometric analysis of indicated immune cell populations in BM of UNC93B1^WT/WT, UNC93B1^WT/V117L and UNC93B1^V117L/V117L mice. Mice were age matched to 8 months. **f**, Intracellular

cytokine staining of IFNγ in BM CD4^+ T cells and CD3^+/CD4^− cells of the indicated mice. MFI, mean fluorescence intensity. Mice were age matched to 8 months. **g**, Flow cytometric analysis of indicated immune cell populations in spleen of UNC93B1^WT/WT, UNC93B1^WT/V117L and UNC93B1^V117L/V117L mice (*n* as indicated). Mice were age matched to 8 months (for B cells, CD4 T cell:B cell ratio, regulatory T cells and CD45^+/CD4^+/CD44^+ cells) or 14 months (for activated T cells, germinal center B cells and CD45^+/CD19^+/CD3^−/CD95^+/CXCR5^+ cells). *n* as indicated. Statistical analysis was done using unpaired, two-tailed Student's *t*-tests and data are presented as the mean with s.d. The exact *P* values are shown. Abs, absorbance; OD, optical density.

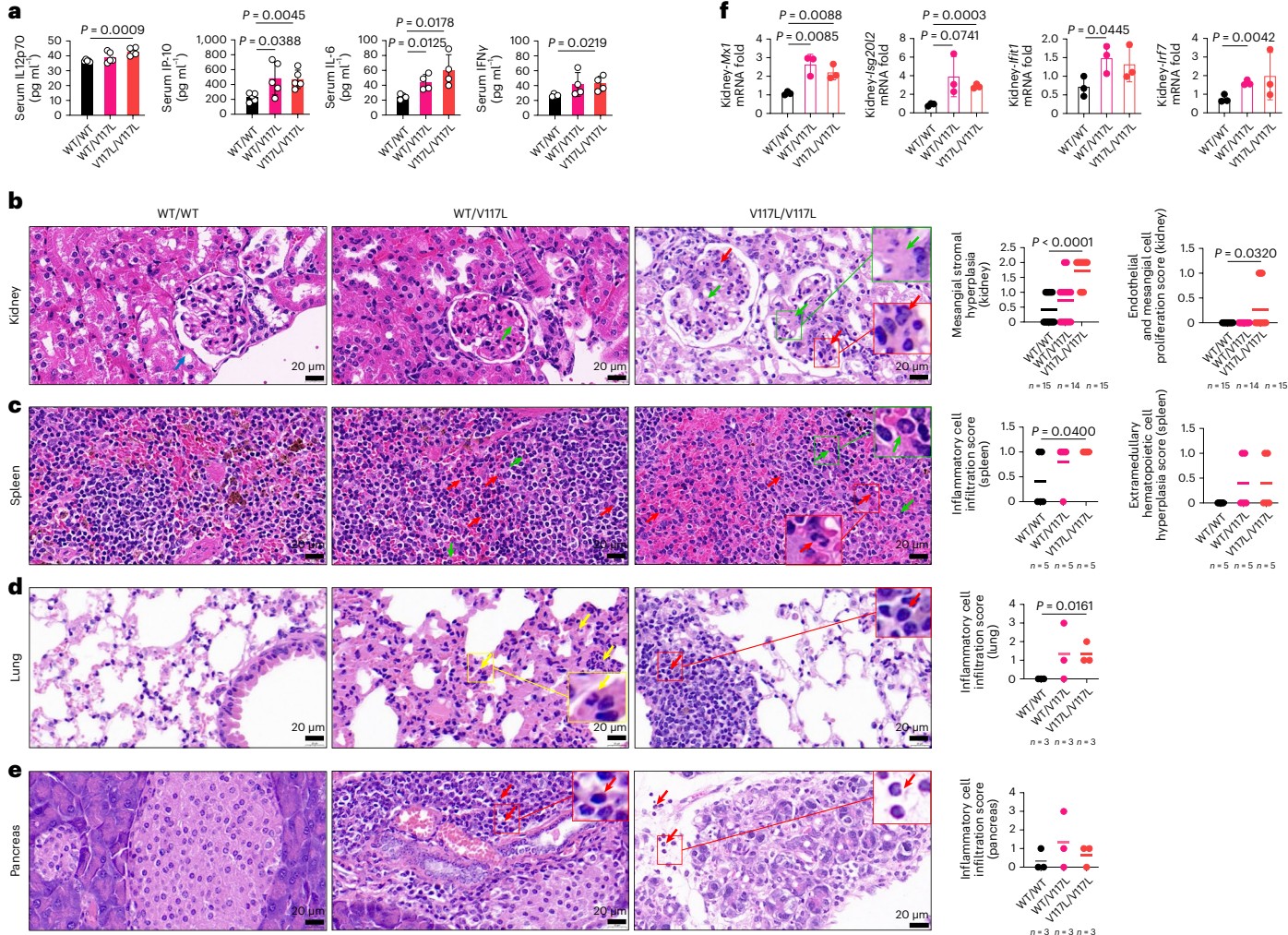

**Fig. 7 | UNC93B1^V117L mice develop systemic inflammation and organ damage.**
**a**, Serum IL-12p70, IP-10, IL-6 and IFNγ of the indicated mice (n = 5 for IL-12p70 and IP-10; n = 4 for IL-6 and IFNγ). IP-10 mice were aged 8 months and IL-12p70, IL-6 and IFNγ mice 10 months. **b**, H&E staining of the kidneys from indicated mice (n as indicated). The green arrows indicated mesangial stromal hyperplasia and the red arrows endothelial and mesangial proliferation. Mice were age matched to 8–14 months and the data pooled from three independent experiments. **c**, H&E staining of the spleens from the indicated mice (n = 5). The green arrows indicated the extramedullary hematopoietic cells and the red arrows the inflammatory cells. Mice were age matched to 8 months. **d**, H&E staining of the lungs from the indicated mice (n = 3). The yellow arrows (granulocytes) and red arrow (lymphocytes) indicated inflammatory cell infiltration; mice were aged 14 months. **e**, H&E staining of the pancreata from the indicated mice (n = 3). The red arrows indicate inflammatory cell infiltration (granulocytes or lymphocytes) and the mice were aged 14 months. The graphs show the pathological disease scores. **f**, RT–qPCR analysis of Mx1, Isg20l2, Ifit1 and Irf7 mRNA in kidney tissues from the indicated mice (n = 3). Mice were age matched to 8 months. Statistical analysis was done using unpaired, two-tailed Student's t-tests and data are presented as mean with s.d. The exact P values are shown.

or type II IFN[28] and include the central TLR signaling molecule IRF7, as well as IRF3 and TBK1 (TANK-binding kinase 1)[29] (Extended Data Fig. 9b,c). Consistently, this was associated with pathway analysis implicating TLR signaling, TNF signaling, antigen processing and presentation (Extended Data Fig. 9d–f). Overall, there is a very significant association with genes listed in the Kyoto Encyclopedia of Genes and Genomes (KEGG) under the disease: Systemic lupus erythematosus (Fig. 8c). To investigate the specificity of NAS-TLR signaling in this mouse model, BMDMs were stimulated by NAS-TLR ligands and tested for inflammatory markers. As in human samples, data from these experiments showed the involvement of TLR7 in the mouse lupus-like phenotype. *IFIT1*, *IRF7*, *ISG-15* and *TNF* genes were more highly expressed in BMDMs of mice with the UNC93B1^V117L variant after stimulation with R848 but not poly(I:C) or CPG-C (Fig. 8d). Also, secretion of CXCL1, CCL5 and IP-10 in the supernatant of UNC93B1^V117L BMDMs was elevated after R848 stimulation but not after stimulation by poly(I:C) or CPG-C (Fig. 8e). These findings implicate enhanced TLR7 signaling as the pathway driving inflammation and a lupus-like phenotype in mice with the equivalent gene variant to humans, contributing to childhood-onset SLE.

Therefore, UNC93B1^V117L drives a lupus-like disease in mice at a cellular level, with relevant end-organ damage. This is associated with activation of the TLR7 signaling pathway and inflammatory gene expression that is consistent with the clinical presentation of patients with the orthologous variant, who suffer from childhood-onset SLE.

## Discussion

Our discovery of rare genetic changes in UNC93B1 that predispose to childhood-onset SLE was facilitated by a large body of work culminating in the discovery of lupus-causing variants in TLR7 (ref. 4). It is interesting that all patients with UNC93B1 variants identified in the present study were female, which could relate to a lack of TLR7 X-chromosome inactivation[30]; however, it could also reflect the strong gender predisposition

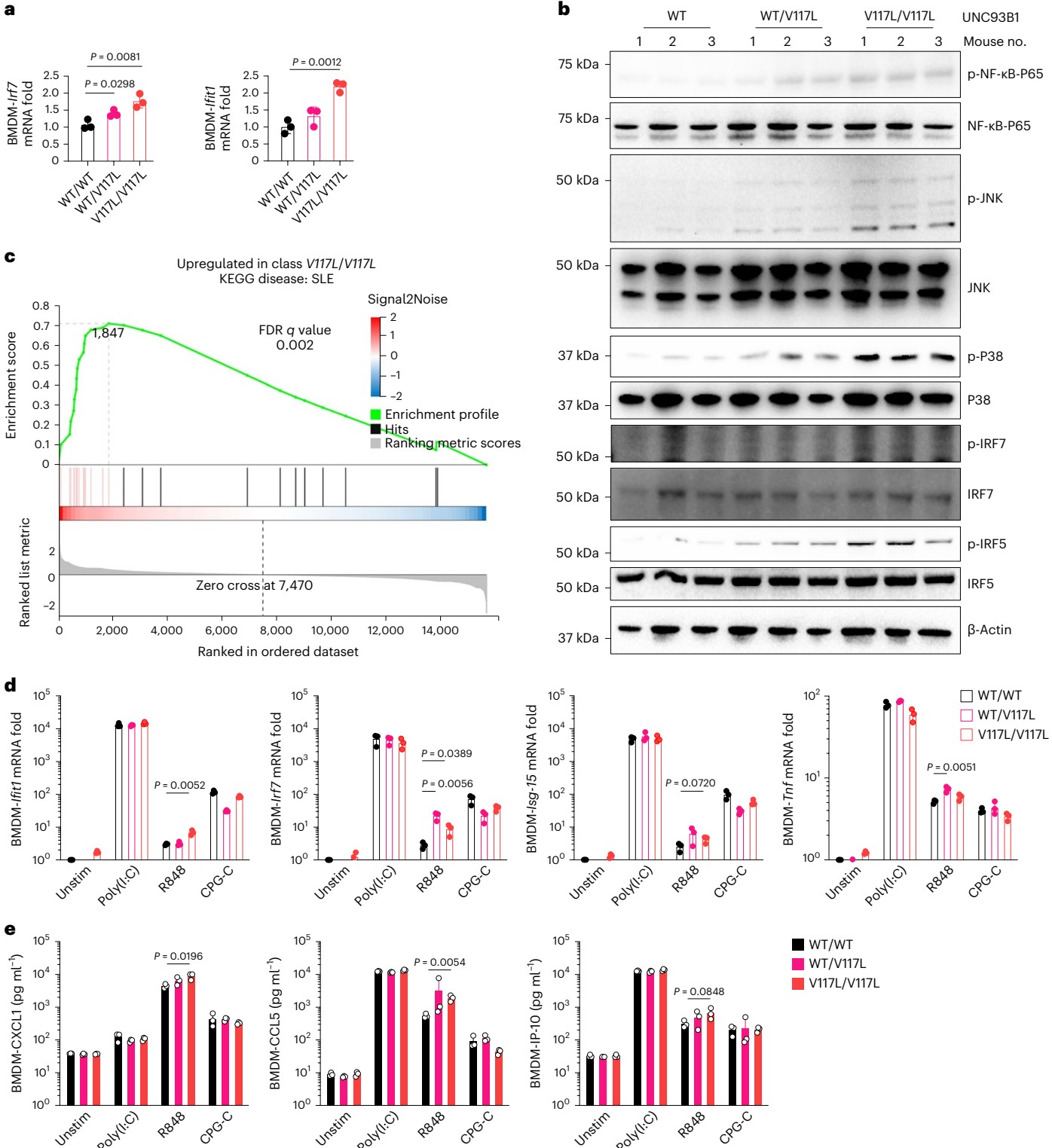

**Fig. 8 | UNC93B1^V117L drives increased inflammation and TLR7 responses in mice. a**, RT–qPCR analysis of *Irf7* and *Ifit1* mRNA in BMDMs from the indicated mice (*n* = 3). Mice were age matched to 8 months. Statistical analysis was done using unpaired Student's *t*-tests. The exact *P* values are shown. **b**, Levels of phosphorylated NF-κB, JNK, P38, IRF5 and IRF7, as measured by immunoblotting, in lysates of BMDMs from the indicated mice (*n* = 3). Mice were age matched to 8 months. **c**, RNA-seq performed for UNC93B1^V117L BMDMs compared with controls (UNC93B1^WT) (*n* = 3 biological replicates). Mice were age matched to 8 months. Gene set enrichment analysis significantly associated with SLE. **d**, RT–qPCR analysis of *Ifit1*, *Irf7*, *Isg-15* and *Tnf* mRNA expression in the BMDMs from the indicated mice (*n* = 3) after stimulation by 40 µg ml⁻¹ of HMW poly(I:C), 2 µg ml⁻¹ of R848 and 10 µg ml⁻¹ of CPG-C for 24 h. Mice were age matched to 14 months. Statistical analysis was done using unpaired Student's *t*-tests. The exact *P* values are shown. **e**, Production of CXCL1, CCL5 and IP-10 in the supernatant of mice BMDMs isolated from the indicated mice (*n* = 3), measured by CBA after stimulation by 40 µg ml⁻¹ of HMW poly(I:C), 2 µg ml⁻¹ of R848 and 10 µg ml⁻¹ of CPG-C for 24 h. Mice were age matched to 14 months. Statistical analysis was done using unpaired, two-tailed Student's *t*-tests and data are presented as the mean with s.d. The exact *P* values are shown.

of this disease in general. We found that the spontaneous inflammatory signaling via IRAK1/-4, due to the variants in UNC93B1 identified here, is associated with increased responses to stimulation of TLR7/-8. This seems logical based on the current literature; however, further work is required to formally delete or inhibit TLR7 and resolve pathology from the UNC93B1[V117L] mouse model that we generated. Theoretically, there should also be an endogenous ligand to stimulate TLR7 in this context, which could be guanosine or another nucleic acid[31]. Furthermore, although the structural location of UNC93B1 T314A presents a logical mechanism to impact TLR7, the molecular effect of UNC93B1 V117L is not yet clear. Given that the phosphorylation of UNC93B1 appeared unaltered, this could also suggest an area in which new findings for the gene variants could be made. Mechanistic insight into this process will be extremely useful and potentially clinically actionable in the future as TLR7 inhibitors are being developed[32].

Not only are TLR7 inhibitors in clinical trials, but also IRAK1/-4 inhibitors could have therapeutic benefit for the patients identified, based on our results. So far, none of the patients characterized in the present study have gone on to receive JAK inhibitors or biological therapeutics that would also target type I IFN signaling. Those approaches should be beneficial and this could be particularly important given that UNC93B1 V117L is present in the general east Asian population, where it could be a contributor to the incidence of SLE[33]. Within this population there is a strong geographical bias (Fig. 1c), for which the underlying basis is unknown. Given that the family members of the affected individuals in the present study were all apparently healthy, we currently consider that UNC93B1 V117L is not a monogenic disease-causing allele with incomplete penetrance, but rather a very strong risk factor for developing childhood-onset SLE. This is consistent with other gene mutations in lupus, for example, DNase1, for which healthy carriers have been observed[34].

Our mouse model represents an important confirmation of the patient findings and shows that a gene dosage effect of the gain-of-function variant is present. Although no homozygous humans have been identified so far, we can speculate that they may have more severe, or a greater likelihood to develop, disease. As UNC93B1 V117L is present in the general population as a rare but highly significant risk factor for disease, the preclinical efficacy of new therapeutic modalities can be accurately modeled for the resulting patient population using our mice avatars. The mouse model should also be useful to determine the cellular contribution of UNC93B1 to lupus-like disease, either intrinsically in B cells or with activation of innate immune signaling from antigen-presenting cells, and the distinct contribution of type I IFNs compared with other inflammatory programs. We would expect this to be similar to other gain-of-function UNC93B1 mouse models of lupus that have been published[35].

Overall these findings, and as published by other groups while this manuscript was under revision[36,37], demonstrate childhood-onset SLE caused by variants in UNC93B1. Additionally, we bridge the gap from rare monogenic diseases to show that this signalling pathway is relevant to the incidence of disease more generally. Moreover, therapeutics targeting this specific pathway are being developed and can be tested first in a mouse model for which there is a corresponding patient population.

## Online content

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

[1]Department of Immunology, Guangzhou Institute of Paediatrics, Guangzhou Women and Children's Medical Centre, and State Key Laboratory of Respiratory Diseases, Guangzhou Medical University, Guangzhou, China. [2]Department of Medical Microbiology, Faculty of Medicine, University of Science and Technology, Aden, Yemen. [3]Department of Therapy and Nursing, Bashkir State Medical University, Ufa, Russia. [4]Inflammation Division, Walter and Eliza Hall Institute of Medical Research, Parkville, Victoria, Australia. [5]Department of Medical Biology, University of Melbourne, Parkville, Victoria, Australia. [6]State Key Laboratory of Respiratory Diseases, School of Basic Medical Sciences, Guangzhou Medical University, Guangzhou, China. [7]Center for Mitochondrial Genetics and Health, Greater Bay Area Institute of Precision Medicine (Guangzhou), Guangzhou, China. [8]Clinical Laboratory, Guangzhou Women and Children's Medical Centre, Guangzhou Medical University, Guangdong, China. [9]National Children Medical Center, Department of Clinical Immunology, Children's Hospital of Fudan University, Shanghai, China. [10]School of Life Sciences, Sun Yat-sen University, Guangzhou, China. [11]Centre for Innate Immunity and Infectious Diseases, Hudson Institute of Medical Research, Clayton, Victoria, Australia. [12]Department of Molecular and Translational Science, Monash University, Clayton, Victoria, Australia. [13]These authors contributed equally: Mahmoud Al-Azab, Elina Idiiatullina. [14]These authors jointly supervised this work: Yuxia Zhang, Seth L. Masters. ✉e-mail: yuxia.zhang@gwcmc.org; seth.masters@hudson.org.au

## Methods

### Patients

Female patients P1–P8, aged 8, 9, 8, 10, 10, 12, 13 and 26 years, respectively, presented symptoms of systemic inflammation associated with lupus nephritis. Patients' blood and other samples were collected at Guangzhou Women and Children's Medical Centre, Guangzhou, China. No compensation was provided. All patients were diagnosed as patients with SLE according to the European League against Rheumatism/American College of Rheumatology (EULAR/ACR) classification criteria. Clinical data for P1–P7 are presented (Fig. 2a–f, Extended Data Figs. 1a–g and 2a–f and Supplementary Table 3). Written informed consent has been obtained from the authorized individual for all participating patients. The Guangzhou Women and Children's Medical Centre Medical Ethics Committee approved the study procedures ((2021)073B00) consistent with the Helsinki Declaration about ethics in using human samples.

### DNA sequencing

WES and Sanger sequencing were performed at Novogene. The frequency of variant V117L in the east Asian population was obtained from PGC.Han 2.0 (ref. 14).

### Human PBMC isolation and plasma separation

Noncoagulated blood samples from the patients with SLE and healthy controls were diluted in an equal volume of sterile phosphate-buffered saline (PBS), and dispensed slowly and gently along the side of a 15-ml conical tube containing 2 ml of Ficoll–Hypaque density gradient. This was followed by centrifugation at 800$g$ for 20 min, 4 Acc (acceleration), 4 Dec (deceleration) and 4 °C. The cloud-like cell layer of cells was collected from the interphase into a new tube and washed twice with sterile PBS by centrifuging at 300$g$ for 5 min. The pellet was resuspended in complete culture medium consisting of 90% Roswell Park Memorial Institute (RPMI)-1640, 10% fetal bovine serum (FBS) and 1% (v/v) penicillin–streptomycin (Gibco) and then incubated at 37 °C, 5% $CO_2$ and 95% humidity. The human PBMCs were used in experiments immediately after isolation or stored FBS with 10% dimethyl sulfoxide (DMSO) at −80 °C. All of the above procedures were performed in an aseptically controlled area using sterilized tools.

For plasma separation, noncoagulated blood samples from the patients with SLE and healthy controls were centrifuged at 1,500$g$ for 10 min; then the upper yellow fluid was separated into a new tube by pipetting without disturbing the layer of buffy coat. Samples were stored at −40 °C up to the time of analysis.

### Immortalized B cell line generation

Immortalized B cells were generated using noncoagulated blood samples from the patients with SLE, P2 and P3 and healthy controls through infection by Epstein–Barr virus according to a previous protocol[38].

### Mice

Animal studies were approved by the Institutional Animal Care and Use Committee of Guangzhou Medical University (protocol no. GY2022-035). Mice lines (*UNC93B1*$^{WT/V117L}$ and *UNC93B1*$^{V117L/V117L}$) were designed, developed and housed by Shanghai Model Organisms Center, Inc. under specific pathogen–free conditions with normal diet, 12-h light:12-h dark cycle, 20–26 °C and humidity of 40–70%. The point mutation mice model at exon3 ($^{117}V$ to $L$) of the *UNC93B1* gene was generated via CRISPR–Cas9 technology. Briefly, Cas9 mRNA and guide RNA were produced by in vitro transcription, an oligonucleotide donor DNA was synthesized and the mixture of Cas9 mRNA, gRNA and donor DNA was microinjected into fertilized eggs (C57BL/6J). Then three positive F0 mice were identified by PCR and sequencing, F0 mice were crossed with WT C57BL/6J mice to generate F1 mice and then four positive F1 mice were identified by PCR and sequencing. The gRNA used was: GTG-TAGAGCAGGGCAGCGATAGG. The recombinant strategy, including the knock-in locus and the oligonucleotide donor DNA sequence, and the

site of target for *UNC93B1*$^{WT/WT}$, *UNC93B1*$^{WT/V117L}$ and *UNC93B1*$^{V117L/V117L}$ genotypes are shown in Extended Data Fig. 8. All mice used for analysis in this project are females within the age indicated for each experiment independently in 'Results'. Within genotypes, animals were randomly allocated in all experiments. Data collection and analysis were not performed blind to the conditions of the experiments, except for histopathology. No animals or data points were excluded except if an animal was injured as a result of fighting or contaminated samples.

### Mouse serum collection

Mice orbital whole-blood samples were collected into tubes without anticoagulant and incubated at 25 °C for at least 15 min to coagulate the blood. The coagulated blood samples were centrifuged at 1,500$g$ for 10 min; then the upper yellow fluid was separated into a new tube using a pipette. Samples were stored at −40 °C up to the time of analysis.

### Tissue digestion and flow cytometry

The whole spleen was minced into tiny pieces in a six-well plate on ice using the plunger end of the syringe and filtered using a 70-μm strainer with rinsing by 5 ml of PBS with 2% FBS. The collected tissue filtrates were centrifuged at 500$g$ for 5 min at 4 °C. The cell pellet was suspended in RBC lysis buffer, incubated for 5 min at 25 °C and then mixed with a double volume of PBS with 2% FBS. The immune cells were collected in the pellet after centrifugation at 500$g$ for 5 min at 4 °C.

BM was collected from mouse femur bones by centrifuging using a technique of two layers of tubes and incubated with RBC lysis buffer for 5 min at 25 °C, and then mixed with a double volume of PBS with 2% FBS. The immune cells were collected in the pellet after centrifugation at 500$g$ for 5 min at 4 °C.

For flow cytometry, 1 million cells were incubated with Zombie Aqua-A (BioLegend) for 15 min in the dark, washed for staining by antibodies for cell-surface markers, incubated for 30 min at 4 °C and then washed. A fixation and permeabilization kit (Thermo Fisher Scientific, cat. no. 88-8824-00) was used for fixation and intracellular permeabilization for intracellular and nuclear markers, according to the manufacturer's protocol. Cells were analyzed using CYTEK NL-CLC. Data were collected using CYTEK NL-CLC software and SpectroFlo and analyzed using FlowJo_v.10.8.1. A gating strategy is shown in Extended Data Fig. 10.

### Isolation of mouse BMDMs

The BM of mice femur bones was collected aseptically as mentioned above. BMDMs were generated by incubating the BM immune cells in a complete medium, RPMI-1640, 10% FBS, 2 mM L-glutamine and 1% (v/v) penicillin–streptomycin, containing 50 ng ml$^{-1}$ of macrophage-CSF (Peprotech) for 6–8 d at 37 °C with 5% $CO_2$ and 100% humidity.

### Intracellular staining for TNF and IFNγ

BMDMs or BM immune cells were stimulated by a 1× cocktail of phorbol 12-myristate 13-acetate, ionomycin, Brefeldin A and monensin from Invitrogen (cat. no. 00-4975-93) for 4–6 h at 37 °C with 5% $CO_2$ and 100% humidity, and then stained using PE-Cy7 TNF and BV-421 IFNγ antibody with a fixation and permeabilization kit (Thermo Fisher Scientific, cat. no. 88-8824-00) according to the manufacturer's protocol. Cells were analyzed using CYTEK NL-CLC. Data were collected using CYTEK NL-CLC software and SpectroFlo and analyzed using FlowJo_v.10.8.1.

### Histopathology

The kidney, spleen, lung and pancreas tissues were preserved in 4% paraformaldehyde. Tissue processing and histopathology reporting were performed by Wuhan Servicebio Technology Laboratory. For disease scoring (blinded) of tissue sections stained by H&E, 0 = within the normal range, 1 = very slight (the changes that appear just exceeded the normal range), 2 = slight (lesions may be observed but not severe), 3 = medium (lesions are obvious and likely to be more severe) and

4 = severe (lesions have taken up the entire tissue and organs). For kidney pathology of mesangial stromal hyperplasia and mesangial and endothelial cell proliferation, the score is calculated for around 50 glomeruli per mouse.

### Antibodies and reagents

For immunoblots, the following antibodies were used: anti-FLAG (Sigma-Aldrich, cat. no. F1804), anti-NF-κB p65 (Cell Signaling, cat. no. 8242S), anti-phospho-NF-κB p65 (Ser468) (Cell Signaling, cat. no. 3039S), anti-JNK (Abcam, cat. no. ab179461), anti-phospho-JNK (Abcam, cat. no. ab124956), anti-P38 (Cell Signaling, cat. no. 8690), anti-phospho-P38 (Cell Signaling, cat. no. 9211S), anti-IRF5 (Cell Signaling, cat. no. 20261), anti-phospho-IRF5 (Ser437) (Invitrogen, cat. no. PA5-64760), anti-IRAK4 (Cell Signaling, cat. no. 4363), anti-phospho-IRAK4 (Thr345/Ser346) (Cell Signaling, cat. no. 11927), anti-p44/42 MAPK (ERK1/2) (Cell Signaling, cat. no. 9102S), anti-phospho-p44/42 MAPK (ERK1/2) (Thr202/Tyr204) (Cell Signaling, cat. no. 9101S), anti-IRF7 (Cell Signaling, cat. no. 39659), anti-phospho-IRF7 (Ser437/438) (Cell Signaling, cat. no. 24129), anti-p-Ser/phosphoserine (Santa Cruz Biotechnology, cat. no. sc-81514), anti-β-actin (ABclonal, cat. no. AC026), anti-rabbit IgG horseradish peroxidase (HRP)-linked antibody (Cell Signaling, cat. no. 7074) and anti-mouse IgG HRP-linked antibody (Cell Signaling, cat. no. 7076). The dilution of anti-β-actin, anti-rabbit IgG HRP-linked antibody, anti-mouse IgG HRP-linked antibody and anti-FLAG was 1:5,000. The dilution of anti-p-Ser/phosphoserine was 1:1,000. The dilution for other primary antibodies was 1:1,500. For flow cytometry, the following antibodies and reagents were used: anti-mouse TNF (BioLegend, cat. no. 506324, 1:200), anti-mouse IFNγ (BioLegend, cat. no. 505830, 1:400), anti-mouse CD45 (BioLegend, cat. no. 103132, 1:200), Zombie Aqua-A (BioLegend, cat. no. 423101, 1:500), anti-mouse F4/80 (BioLegend, cat. no. 123137, 1:200), anti-mouse/human CD11b (BioLegend, cat. no. 101205, 1:500), anti-mouse CD14 (BioLegend, cat. no. 123335, 1:200), anti-mouse CD16 (BioLegend, cat. no. 158004, 1:100), anti-mouse CD3 (BioLegend, cat. no. 100248, 1:100), anti-mouse CD4 (BioLegend, cat. no. 100406, 1:400), anti-mouse FOXP3 (BioLegend, cat. no. 126407, 1:200), anti-mouse CD25 (BioLegend, cat. no. 102043, 1:300), anti-mouse/human CD44 (BioLegend, cat. no. 103059, 1:100), anti-mouse CD69 (BioLegend, cat. no. 104507 or 104510, 1:200), anti-mouse Gr-1 (BioLegend, cat. no. 108416, 1:200), anti-mouse CD19 (BioLegend, cat. no. 159812, 1:400), anti-mouse CD95 (BioLegend, cat. no. 152612, 1:200), anti-mouse/human Bcl-6 (BioLegend, cat. no. 358510, 1:100), anti-mouse/human CD45R/B220 (BioLegend, cat. no. 103222, 1:200), anti-mouse CXCR5 (BioLegend, cat. no. 145517, 1:100), and fixation and permeabilization kit (Thermo Fisher Scientific, cat. no. 88-8824-00). For TLR stimulation, the following ligands were used: high molecular weight (HMW) poly(I:C), a TLR3 ligand (tlrl-pic, InvivoGen), guanosine, a TLR7 ligand (Sigma-Aldrich, cat. no. G6264-1G) resuspended in DMSO (freshly prepared before each experiment), loxoribine, a guanosine analog, a TLR7 ligand (tlrl-lox, InvivoGen), imiquimod (R837), an imidazoquinoline amine analog to guanosine, a TLR7 ligand (tlrl-imqs-1, InvivoGen), TL8-506, a benzazepine compound and an analog of the TLR8 agonist VTX-2337, a TLR8 ligand (tlrl-tl8506, InvivoGen) and R848 (also known as resiquimod, an imidazoquinoline and dual TLR7 and TLR8 synthetic agonist) (tlrl-r848, InvivoGen), CPG-C, synthetic oligonucleotides that contain unmethylated CpG dinucleotides in particular sequence contexts (CpG motifs), a TLR9 ligand (tlrl-m362, InvivoGen) used in human experiments and CPG-C, ODN 2395 VacciGrade (InvivoGen, cat. no. vac-2395-1) (used in mice BMDM stimulation).

### UNC93B1 lentivirus constructs and transduction

For human UNC93B1 (NM_030930) overexpression, constructs of WT or mutated, V117L or T314A UNC93B1 were inserted into Ubi-MCS-3FLAG-SV40-Cherry-IRES-puromycin. Site-directed mutagenesis technology was used to generate c.G349T:p.V117L and c.A940G:p.T314A mutation using a WT UNC93B1 construct. These plasmids, along with

PSPAX2 and pMD2G, were used to produce lentiviruses in the supernatant. The design and synthesis of plasmids and lentivirus production were done at Genechem Laboratory, Shanghai, China. THP-1 cells were transduced by indicated lentiviruses to generate stable cell lines overexpressing UNC93B1 WT, V117L and T314A. Puromycin, 2 μg ml$^{-1}$ (Santa Cruz Biotechnology), treatment and FACS sorting by BD FACSAria lll were performed for positive cell selection lasting 2 weeks after infection. The infection efficiency and stable overexpression were verified by immunoblotting assay for flag expression and flow cytometry for mCherry by CYTEK NL-CLC.

### Cell culture

THP-1 cells were obtained from American Type Culture Collection and cultured accordingly. THP-1 and human PBMCs and immortalized B cells were cultured in RPMI-1640 medium supplemented with 10% FBS, 2 mM L-glutamine and 1% (v/v) penicillin–streptomycin. All cells were cultured at 37 °C with 5% CO$_2$ and 100% humidity. THP-1, immortalized B cells and/or PBMCs incubated with zimlovisertib (PF-06650833) (MedChemExpress, cat. no. HY-19836) (in the present study referred to as IRAK4 inhibitor) and/or with IRAK1/4 inhibitor I, a new benzimidazole (Sigma-Aldrich, cat. no. 15409) (in the present study referred to as IRAK1/-4 inhibitor) as indicated in 'Results'.

### Messenger RNA extraction and qPCR

Extraction of mRNA from THP-1, human PBMCs, immortalized B cells, BMDMs or mice kidney tissue homogenate was performed using EZ-press RNA Purification Kit and converted to complementary DNA using the 4× Reverse Transcription Master Mix Kit (with genomic DNA Remover) from EZ Bioscience and 1,000 ng of mRNA per sample according to the manufacturer's instructions. The quantification of RNA concentration was done using Varioskan LUX (Thermo Fisher Scientific). Quantitative real-time PCR (RT–qPCR) was conducted using the 2× Color SYBR Green qPCR Master Mix (ROX2 plus) from EZ Bioscience, cDNA, RNase-free dH$_2$O and primers shown in Supplementary Table 4 by LightCycler 480 ll (Roche, 96 or 384), according to the manufacturer's instructions.

### Immunoprecipitation

Cell lysates were extracted using a low-salt lysis buffer (50 mM Hepes, pH 7.5, 150 mM NaCl, 1 mM EDTA, 1.5 mM MgCl$_2$, 10% glycerol and 1% Triton X-100) supplemented with 5 mg ml$^{-1}$ of protease and a phosphatase inhibitor cocktail (Roche). Cell lysates were incubated with washed anti-Flag M2 magnetic beads (Sigma-Aldrich, cat. no. M8823) at 4 °C overnight with mixing using an end-over-end rotator. Then, the beads were washed 4× by low-salt lysis buffer. The protein eluted by resuspending washed beads with 2× sodium dodecylsulfate–polyacrylamide gel electrophoresis (SDS–PAGE) sample buffer and boiling for 10 min.

### Immunoblotting

Cell lysates were extracted using a low-salt lysis buffer (50 mM Hepes, pH 7.5, 150 mM NaCl, 1 mM EDTA, 1.5 mM MgCl$_2$, 10% glycerol, 1% Triton X-100) supplemented with 5 mg ml$^{-1}$ of a protease and phosphatase inhibitor cocktail (Roche). Protein quantification was performed using Pierce BCA Protein Assay Kit (Thermo Fisher Scientific, cat. no. 23227) according to the manufacturer's instructions. Then, 10% or 12% SDS–PAGE was used for protein lysates (20 μg) with loading buffer and transferred to polyvinylidene fluoride membrane (Millipore Co.). For the phos-tag gel, 12% SDS–PAGE was prepared by adding 5 mM phosbind acrylamide (ABExBIO, cat. no. F4002) and 10 mM MnCl$_2$ and processed according to the manufacturer's instructions. Membranes were blocked with 5% skimmed milk in Tris-buffered saline–Tween-20 (TBST) and then incubated overnight at 4 °C with primary antibodies. After washing, the membranes were incubated for 1 h at 25 °C with secondary antibody conjugated with HRP. Finally, TBST-washed membranes were treated with enhanced chemiluminescence for detection using a Molecular Imager ChemiDoc XRS+ imaging system (BioRad).

Developed membranes were imaged using the Image Lab detection system (BioRad).

## ELISA and CBA

ELISA and CBA experiments were carried out using serum/plasma and/or cell supernatants according to the manufacturer's instructions. ELISA kits for humans, IFNα (CSB-E08636h), IFNβ (CSB-E09889h) and IL-6 (CSB-E04638h) were purchased from Cusabio. ELISA kits for mice, LBIS mouse anti-dsDNA (FUJIFILM Wako Shibayagi Corp., cat. no. 637-02691), mouse anti-Sm Igs (total (A + G +M)) (Alpha Diagnostic International, cat. no. 5405) and mouse IgG (Alpha Diagnostic International, cat. no. 6320) were purchased as indicated. Absorbance were measured using Varioskan LUX (Thermo Fisher Scientific). Human (cat. no. 740390) and mouse (cat. no. 740621) LEGENDplex 13-plex CBA Anti-Virus Response Panel kits, or LEGENDplex 13-plex CBA Human Inflammation Panel 1 (cat. no. 740809) from BioLegend was used in cytokine quantification using a V-bottomed 96-plate protocol. Beads were analyzed using CYTEK NL-CLC. Data were collected using CYTEK NL-CLC software and SpectroFlo and analyzed using the LEGENDplex Data Analysis Software Suite (BioLegend).

## RNA-seq and analysis

THP-1 cells or mice BMDMs were suspended in Trizol and sent to the Beijing Genomic Institute (BGI) laboratory for RNA-seq. For RNA-seq, total RNA was first extracted from human THP-1 cells or mice BMDMs and the mRNA was first enriched using poly(dT), then fragmented and reverse transcribed into cDNA and subjected to end-repair and adapter ligation processes. The resulting mRNA library was paired-end sequenced (2×150 bp) with the sequencer DNBSEQ-T7 (BGI Genomics).

The sequencing reads for human THP-1 cells were then aligned to human reference genome GRCh38 using STAR aligner (v.2.7.9a)[39] to generate the binary alignment file (.bam), which was then passed to HTSeq for gene expression quantification using the htseq-count function (v.0.11.1)[40]. The resulting gene expression matrix was then normalized and differentially expressed genes (DEGs) were identified using R package DESeq2 (v.1.34.0)[41]. Genes with adjusted $P$ ($P_{adj}$) value <0.05 and absolute fold-change ($\log_2$(space)) > 1 were considered to be DEGs. This analysis pipeline was used for the volcano plots in Fig. 3. Other analysis was performed using BGI software (https://biosys.bgi.com/#/report/login).

## Statistical analysis

Statistical analysis was performed using Prism 9 software (GraphPad Software Inc.). Contingency analysis, Fisher's exact test, Bapista–Pike test, $\chi^2$, unpaired Student's $t$-test or analysis of variance (ANOVA: one-way or two-way) was used to compare differences between groups. $P < 0.05$ was considered significant.

## Reporting summary

Further information on research design is available in the Nature Portfolio Reporting Summary linked to this article.

## Data availability

The RNA-seq raw sequencing data for the THP-1 cell line reported in the present study are deposited under supervision and control of the Genome Sequence Archive (GSA) of the BGI, Chinese Academy of Sciences (https://ngdc.cncb.ac.cn/gsa-human/s/uExs2NxY), under accession no. HRA005592. THP-1 RNA-seq data for this project could be accessed after an approval application. Please refer to the GSA (https://ngdc.cncb.ac.cn/gsa-human/) for detailed application guidance. The RNA-seq raw sequencing data for the mouse model reported in the present study are deposited under the supervision and control of the GSA of the BGI, Chinese Academy of Sciences (https://ngdc.cncb.ac.cn/gsa/s/1dDv8y6X), under accession no. CRA012690. For access to mouse RNA-seq data of this project, please contact the corresponding authors for approval. All other data are available in the main text or supplementary materials. Source data are provided with this paper.

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

## Acknowledgements

We are grateful to all members of the laboratories of Yuxia Zhang and S.L.M. for supplying reagents and technical advice or assistance. We also thank S. Adlat for his kind help during the drafting of this manuscript. The present study is supported by the National Natural Science Foundation of China (grant nos 81873869 to S.L.M., 32250410295 to E.I., 82125015 to Yuxia Zhang and 82101911 to Z.Z.), Guangzhou Science and Technology Key R&D Plan (grant no. 202206010002 to Yuxia Zhang), Guangzhou Science and Technology Bureau (grant no. 2024A03J0806 to P.Z.), Research Fund of Guangzhou Women and Children's Medical Center (grant nos 3001124 to M.A. and 2180129 to S.L.M.) and Research Fund of Guangzhou Women and Children's Medical Center (grant no. 3001032 to Yuxia Zhang).

## Author contributions

S.L.M. and Yuxia Zhang conceived the project. M.A. conducted all experiments with assistance from E.I. M.A. performed data analysis, drafted figures and wrote the first version of the manuscript. K.H.-S. performed structural analysis. Z.L. assisted with bioinformatics analysis. J.Z. assisted with animal experiments. M.L., H.X., M.Y, B.L., Z.Z., Yiyi Liu and Jun Cui were involved in experiments and data analysis. Jingjie Chang, X.L. and C.G. assisted with provision of human samples. C.G. assisted with immortalized B cell generation. P.Z., X.G., Yunfeng Liu, Q.W, Jiazhang Chen and C.L. were involved in clinical management. W.Z. obtained funding to support research. Yan Zhang was involved in WES analysis. S.L.M. supervised the project. M.A. wrote the manuscript with input from all the authors.

## Competing interests

S.L.M. is a scientific advisor for Odyssey therapeutics and NRG therapeutics. The other authors declare no competing interests.

## Additional information

**Extended data** is available for this paper at https://doi.org/10.1038/s41590-024-01846-5.

**Correspondence and requests for materials** should be addressed to Yuxia Zhang or Seth L. Masters.

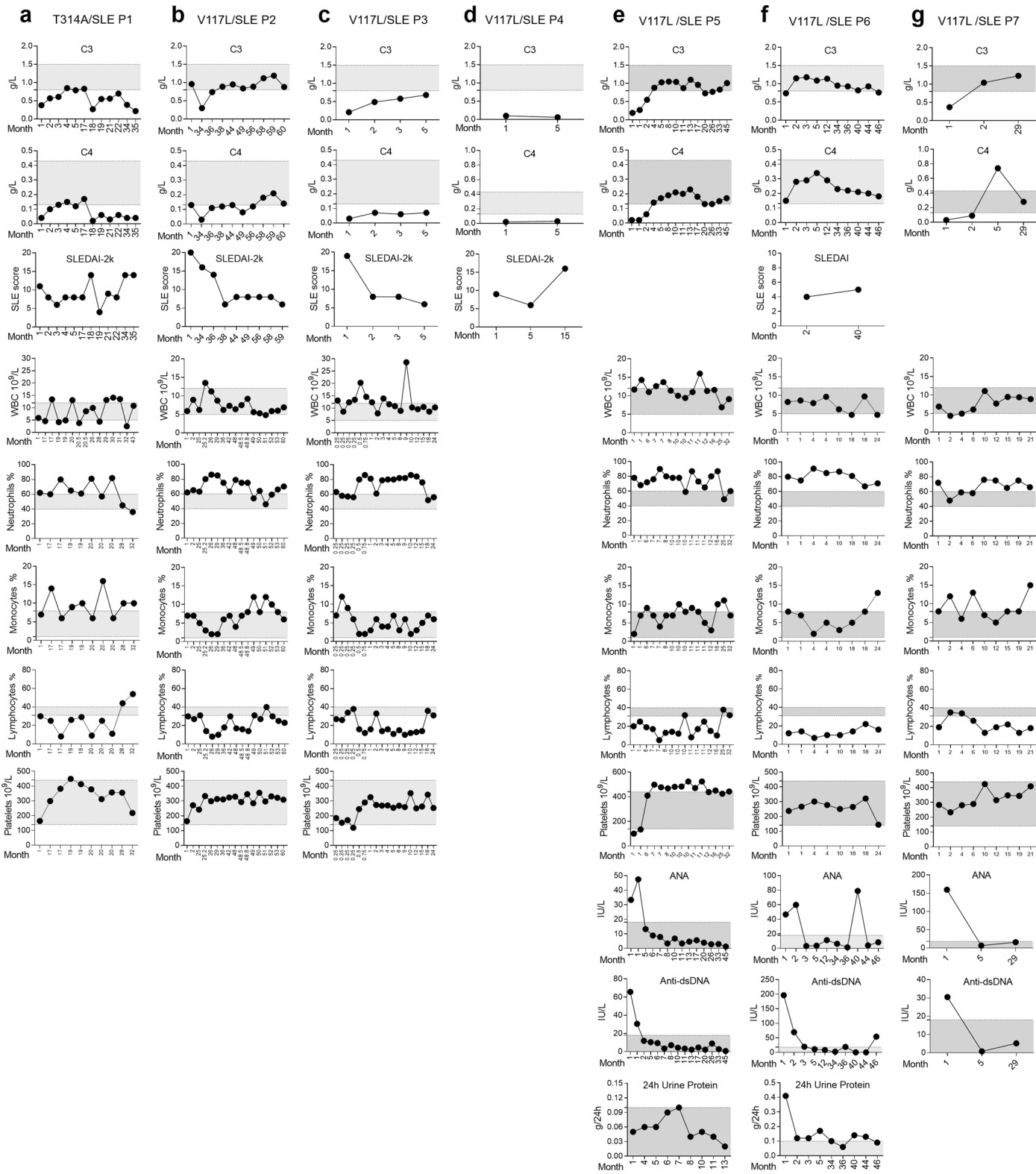

**Extended Data Fig. 1 | Clinical characteristics associated with UNC93B1 T314A and V117 variants. (a–g)** Complement 3 (NR, 0.80-1.50 g/L)/4 (NR, 0.13-0.43 g/L) levels for patients with UNC93B1 T314A, P1 and V117L, P2-P7. **(a-d)** SLE score (SLEDAI-2K), P1-P4. **(f)** SLEDAI score, P6. **(a-c, e-g)** Circulating white blood cells (WBC) (NR, 5-12 × 10⁹/L), neutrophils% (NR, 40-60%), monocytes% (NR, 1-8%), lymphocytes% (NR, 31-40%), and platelets count (NR, 140-440 × 10⁹/L) for patients with UNC93B1 T314A, P1 and V117L, (P2, P3, and P5-P7). **(e-g)** Anti-dsDNA autoantibodies (NR, <18IU/L) and ANA autoantibodies (NR, <18IU/L) for P5-P7 (all V117L). **(e, f)** 24hrs. urine protein (NR, 0-0.1 g/24 h) for P5 and P6 (all V117L). Data is for monthly visits post diagnosis. NR is highlighted in gray.

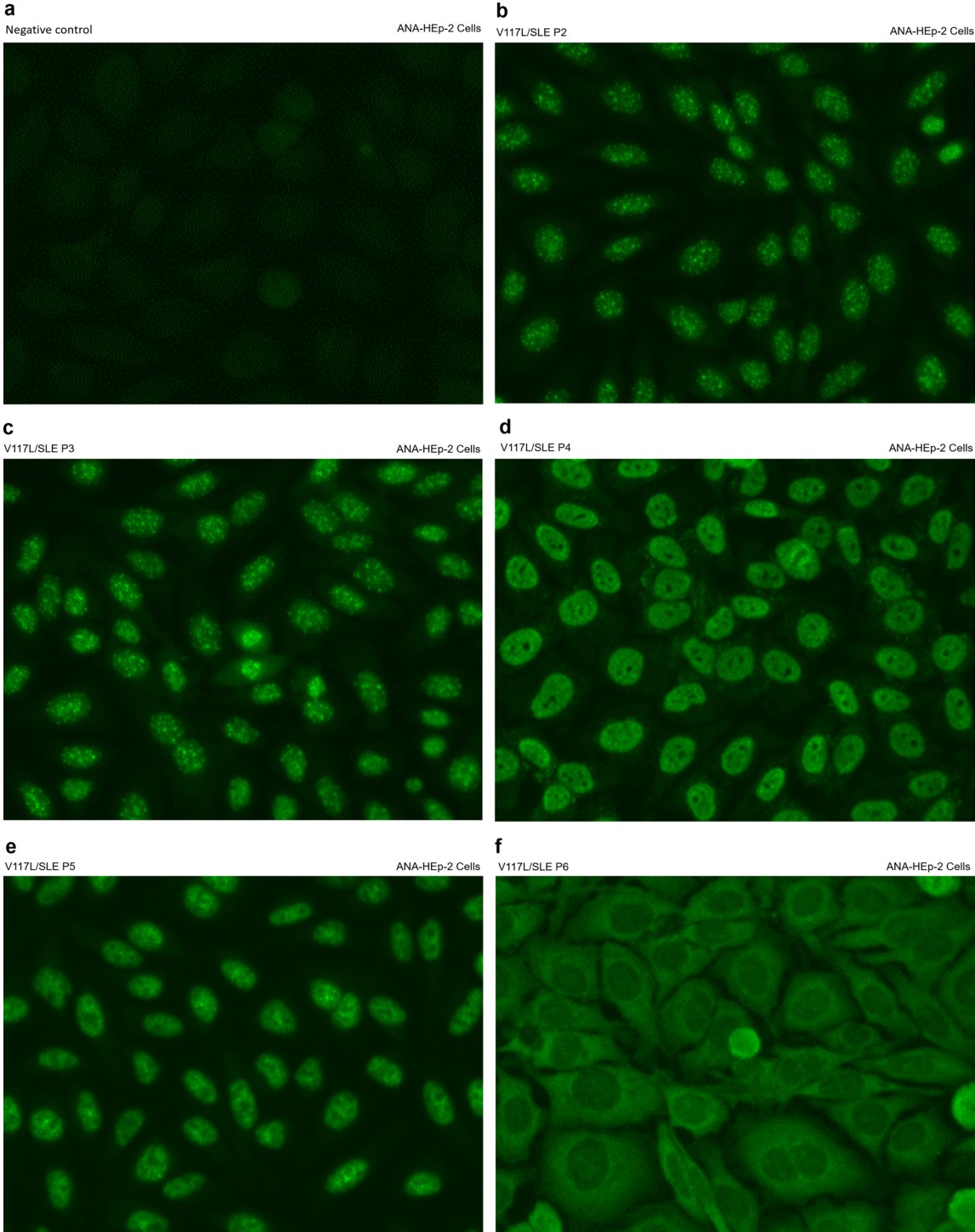

**Extended Data Fig. 2 | Clinical characteristics associated with UNC93B1 V117L variant. (a–f)** ANA autoantibodies by HEp-2 cells, (a) for healthy control, (b, c) for P2 (b) and P3 (c) showing mixed patterns in which the speckled pattern is predominant, (d, e) for P4 (d) and P5 (e) showing speckled pattern, and (f) for P6 showing cytoplasmic pattern. The scale bars were supplied in source data of this figure where these images were magnified from.

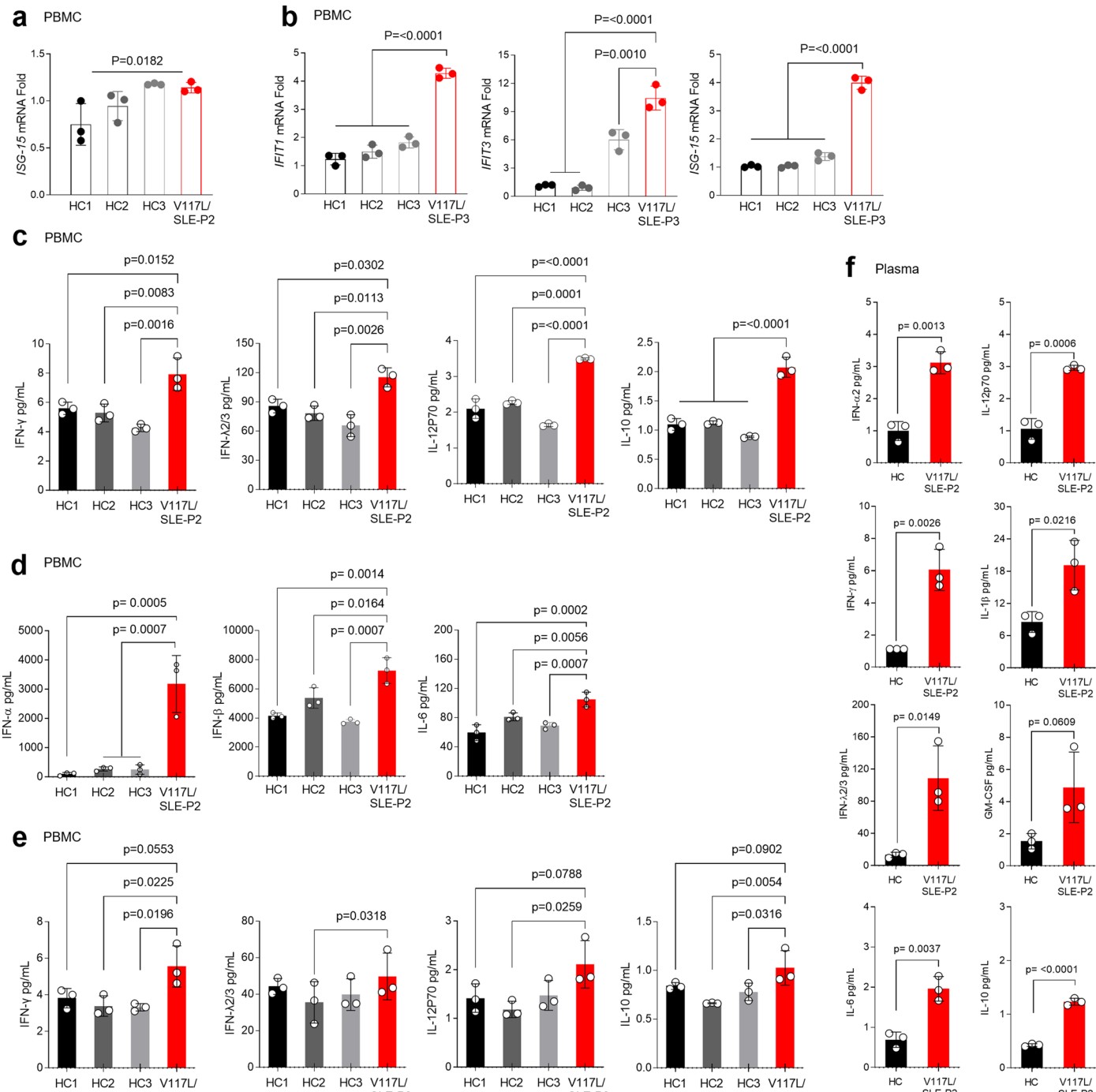

**Extended Data Fig. 3 | Patients with UNC93B1 V117L variant display elevated interferons and inflammatory cytokines. (a)** Quantitative RT-PCR analysis of *ISG-15* mRNA expression in the PBMCs of P2 compared to healthy controls. p value was determined by one-way ANOVA, data are presented as mean with SD, n = 3 biological samples. **(b)** Quantitative RT-PCR analysis of *IFIT1*, *IFIT3*, and *ISG-15* mRNA expression in the PBMCs of P3 compared to healthy controls, n = 3 biological samples. **(c)** Production of IFNγ, IFNλ2/3, IL-12P70, and IL-10 in the supernatant of PBMCs isolated from P2 and healthy controls measured by CBA, n = 3 biological samples. **(d)** Production of IFNα, IFNβ, and IL-6 in the supernatant of PBMCs isolated from P2 and healthy controls measured by ELISA, n = 3 biological samples. **(e)** Production of IFNγ, IFNλ2/3, IL-10, and IL-12P70 in the supernatant of PBMCs isolated from P3 and healthy controls measured by CBA, n = 3 biological samples. Indicated p values in b–e were determined by two-way ANOVA, multiple comparisons, adjusted p value, data are presented as mean with SD. PBMCs in a-e isolated from patients' whole blood and incubated in RPMI 1640 medium supplemented with 10% FBS, 2 mM L-glutamine, and 1% (v/v) Penicillin-Streptomycin for 12hrs before analysis without stimulation. **(f)** Levels of IFN- α2, IFNγ, IFNλ2/3, IL-6, IL-12P70, IL-1β, GM-CSF, and IL-10 in the plasma from P2 compared to healthy control measured by CBA, p values were determined by unpaired t-test, two-tailed, data are presented as mean with SD, n = 3 biological samples.

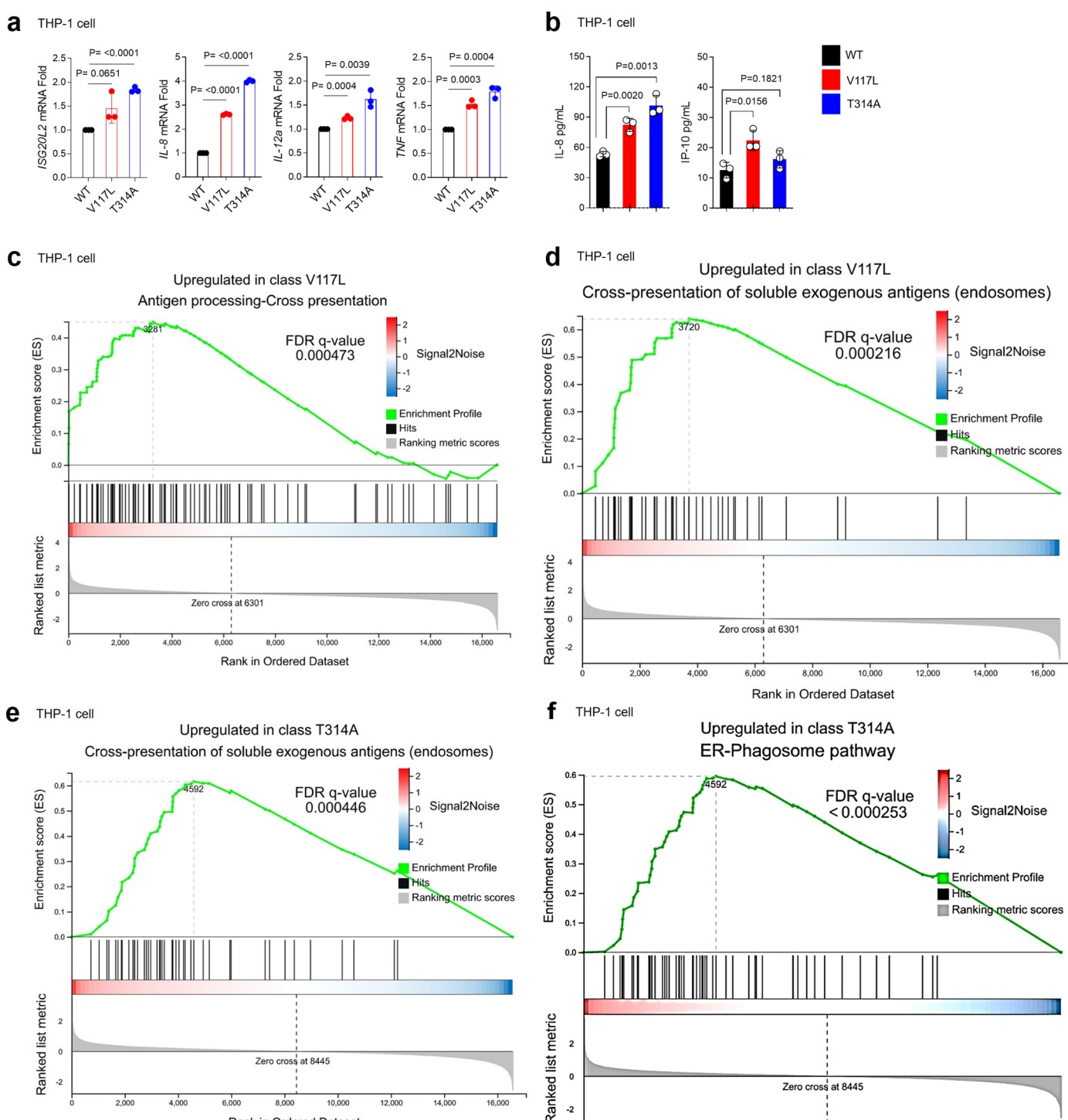

**Extended Data Fig. 4 | Pathway analysis of inflammation in UNC93B1$^{V117L}$ and UNC93B1$^{T314A}$ THP-1 cells.** (**a**) Quantitative RT-PCR analysis of *ISG20L2, IL-8, IL-12a* and *TNF* expression in the indicated THP-1 cell lines, n = 3 biological replicates. Three independent experiments. (**b**) Production of IP-10 and IL-8 in the indicated THP-1 cell lines measured by CBA. n = 3 biological replicates. Indicated p values in a,b were determined by unpaired t-test, two-tailed, data are presented as mean with SD. (**c**–**f**) Gene Set Enrichment Analysis (GSEA) with WT as control groups and UNC93B1$^{V117L}$ or UNC93B1$^{T314A}$ as treatment groups which shows a significant association for genes relating to (**c**) antigen processing and cross presentation, (**d,e**) cross presentation of soluble exogenous antigens and (**f**) ER/Phagosome function. mRNA was extracted from THP-1 of UNC93B1$^{WT}$, UNC93B1$^{V117L}$, and UNC93B1$^{T314A}$, n = 3 biological replicates.

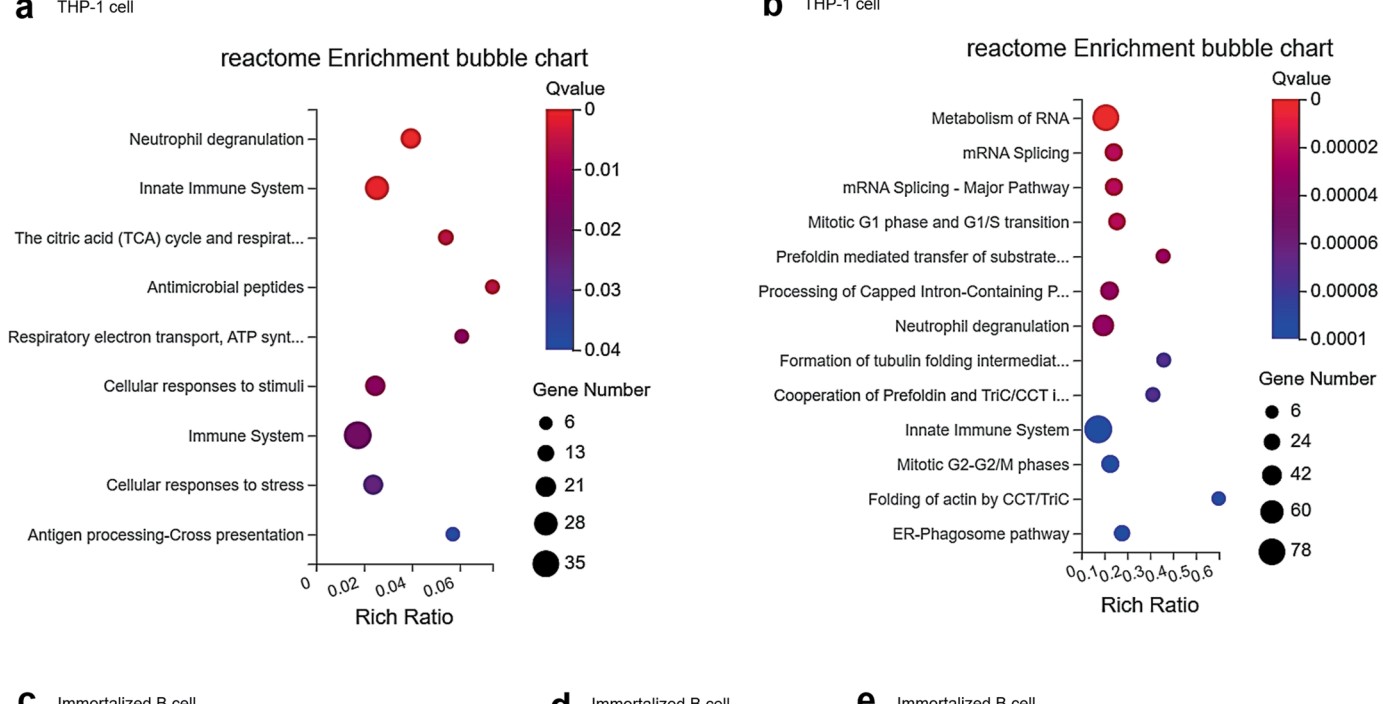

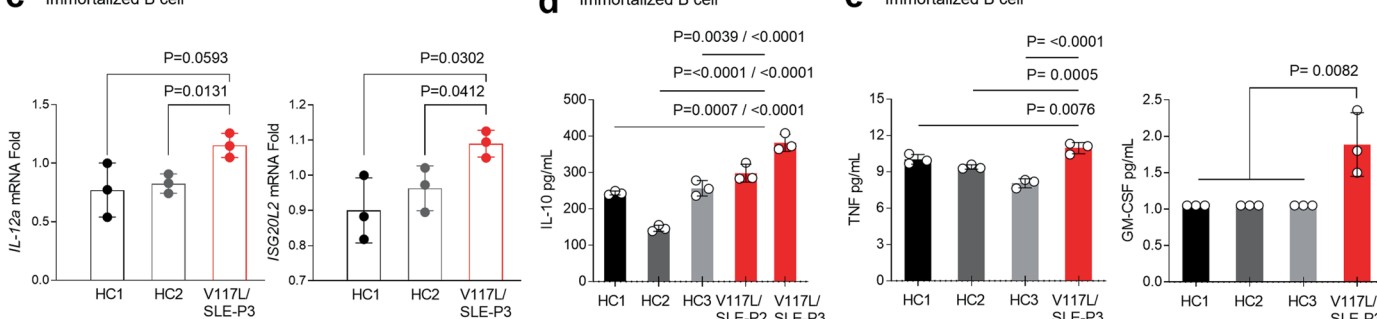

**Extended Data Fig. 5 | V117L and T314A UNC93B1 variants associated inflammation features. (a, b)** DEGs and pathway analysis of inflammation in UNC93B1[V117L] and UNC93B1[T314A] THP-1 cells. (a) RNA sequencing was performed for UNC93B1[V117L] THP-1 cells compared to control (UNC93B1[WT]), reactome Enrichment analysis of significantly upregulated genes highlights innate immune system and antigen processing/presentation. (b) RNA sequencing was performed for UNC93B1[T314A] THP-1 cells compared to control (UNC93B1[WT]), reactome Enrichment analysis of significantly upregulated genes highlights innate immune system and ER/Phagosome pathways. mRNA was extracted from

THP-1 of UNC93B1[WT], UNC93B1[V117L], and UNC93B1[T314A], n = 3 biological replicates. **(c)** Quantitative RT-PCR analysis of *IL-12a* and *ISG20L2* expression in the indicated immortalized B cell lines (baseline), n = 3 biological replicates, indicated p values were determined by unpaired t-test, two-tailed, data are presented as mean with SD. **(d, e)** Production of IL-10 (d), TNF and GM-CSF (e) in the indicated immortalized B cell lines (baseline). n = 3 biological replicates, measured by CBA. Indicated p values were determined by two-way ANOVA, multiple comparisons, adjusted p value, data are presented as mean with SD.

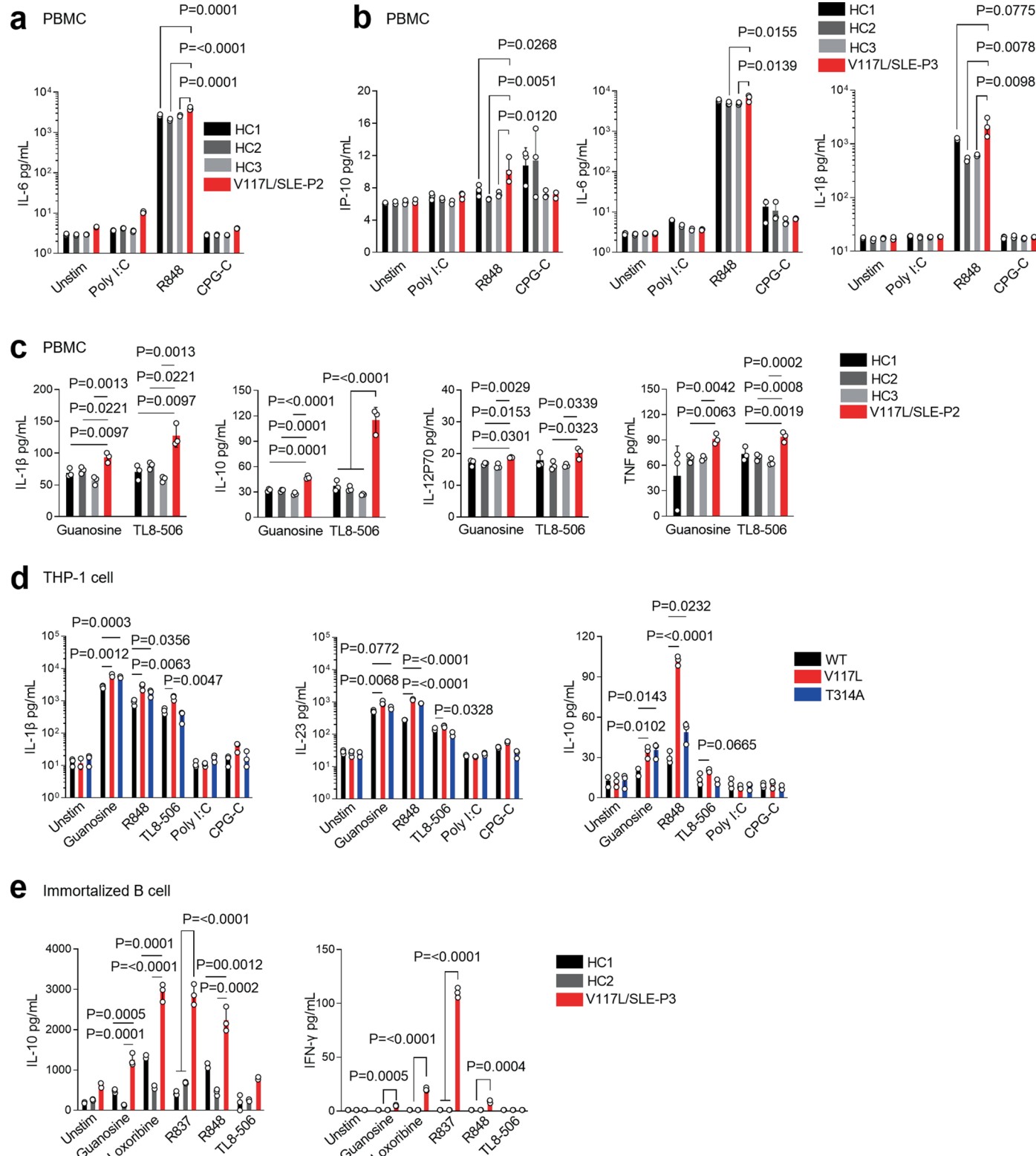

**Extended Data Fig. 6 | See next page for caption.**

**Extended Data Fig. 6 | UNC93B1 genetic variation drives inflammation via TLR7/8.** (**a**) Production of IL-6 in the supernatant of PBMCs of P2 compared to healthy controls after stimulation by 10 µg/mL HMW Poly I:C, 1 µg/mL R848, and 4 µg/mL CPG-C for 8hrs, n = 3 biological samples. (**b**) Production of IP-10, IL-6, and IL-1β in the supernatant of PBMCs of P3 compared to healthy controls after stimulation by 10 µg/mL HMW Poly I:C, 1 µg/mL R848, and 4 µg/mL CPG-C for 12hrs, n = 3 biological samples. Indicated p values in a, b were determined by two-way ANOVA, multiple comparisons, adjusted p value, data are presented as mean with SD. (**c**) Production of IL-1β, IL-10, IL12P70, and TNF in the supernatant of PBMCs of P2 compared to healthy controls after stimulation by 1 mM guanosine and 200 ng/mL TL8-506 for 24hrs, indicated p values were determined by two-way ANOVA, multiple comparisons, adjusted p value, data are presented as mean with SD, p value of TNF (guanosine set) determined by unpaired t-test, two-tailed, data are presented as mean with SD, n = 3 biological samples. (**d**) Production of IL-1β, IL-23, and IL-10 in the supernatant of THP-1 of indicated variants after stimulation by 0.7 mM guanosine, 1 µg/mL R848, 200 ng/mL TL8-506, 100 µg/mL HMW Poly I:C, and 5 µg/mL CPG-C for 24hrs, indicated p values were determined by unpaired t-test, two-tailed, data are presented as mean with SD, n = 3 biological replicates. (**e**) Production of IL-10 and IFN-γ in the supernatant of indicated immortalized B cell lines after stimulation by 0.5 mM guanosine, 0.1 mM loxoribine, 2.5 µg/mL R837, 1 µg/mL R848, and 200 ng/mLTL8-506 for 24 hrs, indicated p values were determined by two-way ANOVA, multiple comparisons, adjusted p value, data are presented as mean with SD, n = 3 biological replicates. Production of cytokines in the cells supernatant was measured by CBA.

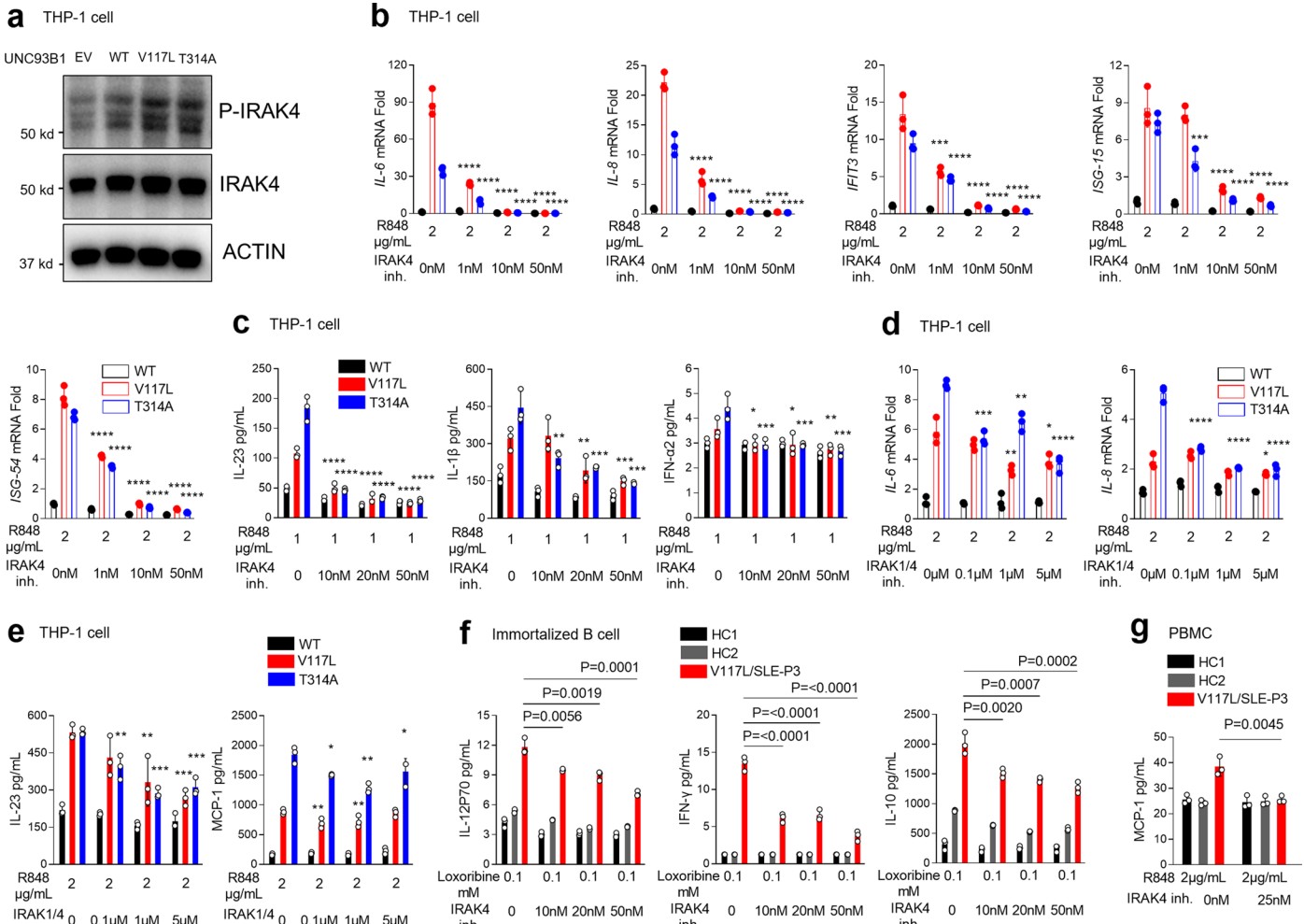

**Extended Data Fig. 7 | IRAK1/4 adaptors involved in UNC93B1 genetic variation-induced inflammation.** (**a**) Levels of phosphorylated IRAK4 as measured by immunoblot using lysates of the indicated THP-1 cells. Data are representative of three independent experiments. (**b**) Quantitative RT-PCR analysis of *IL-6*, *IL-8*, *IFIT3*, *ISG-15*, and *ISG-54* expression in the indicated THP-1 cell lines, stimulated by 2 μg/mL R848 for 24hrs after being incubated with 0, 1, 10, 50 nM IRAK4 inhibitor for 30 minutes, n = 3 biological replicates. ***p = 0.0003, ****p < 0.0001. (**c**) Production of IL-1β, IL-23 and IFN-α2 in the supernatant of the indicated THP-1 cell lines, stimulated by 1 μg/mL R848 for 24hrs after being incubated with 0, 10, 20, 50 nM IRAK4 inhibitor for 30 minutes, n = 3 biological replicates. In IL-1β, **p = 0.0012, **p = 0.0022, ***p = 0.0005, ***p = 0.0005, ***p = 0.0002 respectively. In IFN-α2, *p = 0.0380, ***p = 0.0008, *p = 0.0291, ***p = 0.0007, **p = 0.0092, ***p = 0.0003 respectively. (**d**) Quantitative RT-PCR analysis of *IL-6* and *IL-8* expression in the indicated THP-1 cell lines, stimulated by 2 μg/mL R848 for 24hrs after being incubated with 0, 0.1, 1, 5 μM IRAK1/4 inhibitor for 30 minutes, n = 3 biological replicates. In IL-6, *p = 0.0114, **p = 0.0033, **p = 0.0022, ***p = 0.0003, ****p < 0.0001

respectively. In IL-8, *p = 0.0327, ****p < 0.0001. (**e**) Production of IL-23 and MCP-1 in the supernatant of the indicated THP-1 cell lines, stimulated by 2 μg/mL R848 for 24hrs after being incubated with 0, 0.1, 5 μM IRAK1/4 inhibitor for 30 minutes, n = 3 biological replicates. In IL-23, **p = 0.0023, ***p = 0.0001, ***p = 0.0005, ***p = 0.0002 respectively. MCP-1, **p = 0.0037, *p = 0.0131, **p = 0.0089, **p = 0.0011, *p = 0.0296 respectively. (**f**) Production of IL-12P70, IFN-γ and IL-10 in the supernatant of the indicated immortalized B cell lines, stimulated by 0.1 mM loxoribine for 24hrs after being incubated with 0, 10, 20, 50 nM IRAK4 inhibitor for 30 minutes, n = 3 biological replicates. Indicated p values in b-f were determined by two-way ANOVA, multiple comparisons, adjusted p value, data are presented as mean with SD. (**g**) Production of MCP-1 in the supernatant of the indicated PBMCs, stimulated by 2 μg/mL R848 for 24hrs after being incubated with and without 25 nM IRAK4 inhibitor for 30 minutes, indicated p values were determined by unpaired t-test, two-tailed, data are presented as mean with SD, n = 3 biological samples. Production of cytokines in the cells supernatant was measured by CBA.

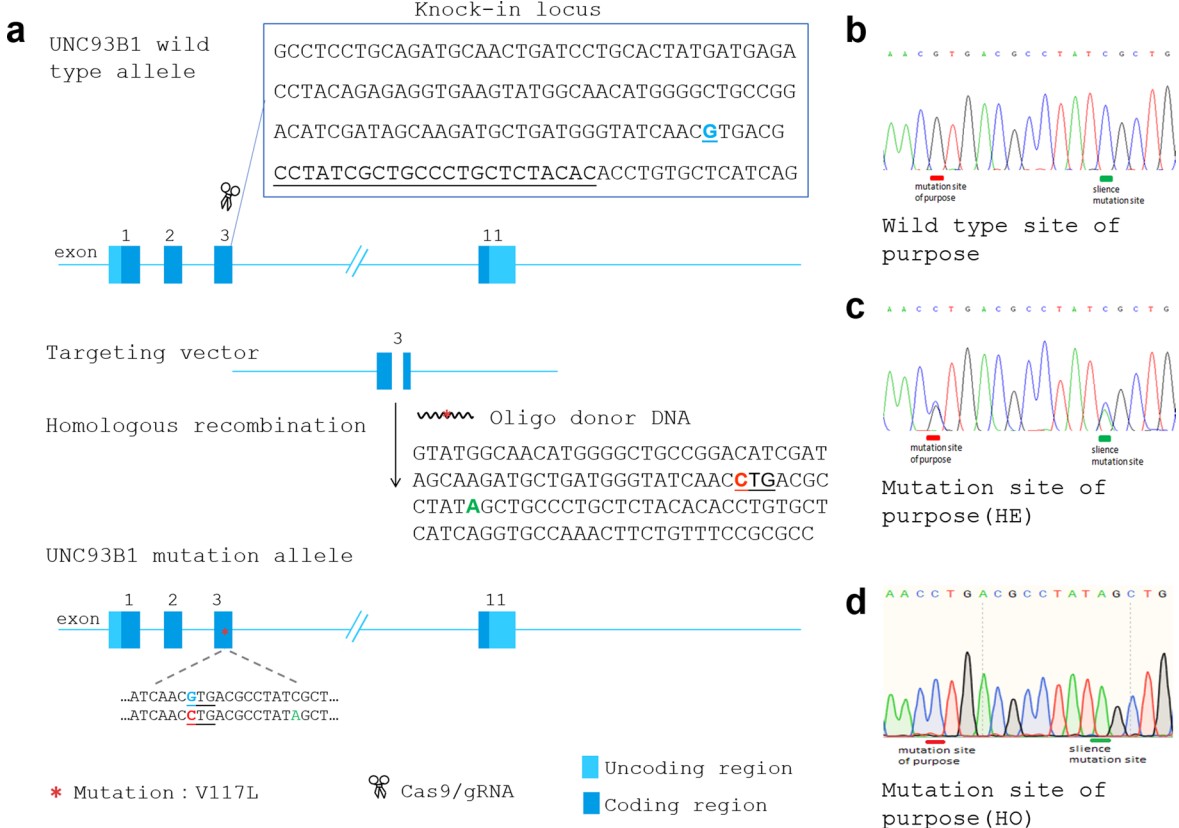

**Extended Data Fig. 8 | Schematic for the CRISPR/Cas9 strategy to generate UNC93B1 V117L knock-in mice.** (**a**) The recombinant strategy, the underlined letters are the GuideRNA target site, and the blue font is the mutant site, the red bold font is the mutant base, and the green bold font is the synonymous mutation made to prevent the gRNA from being cut twice. Target sites for, UNC93B1[WT/WT] (**b**), UNC93B1[WT/V117L] (**c**), and UNC93B1[V117L/V117L] (**d**) mice genotypes. HE, heterozygous; HO, homozygotes.

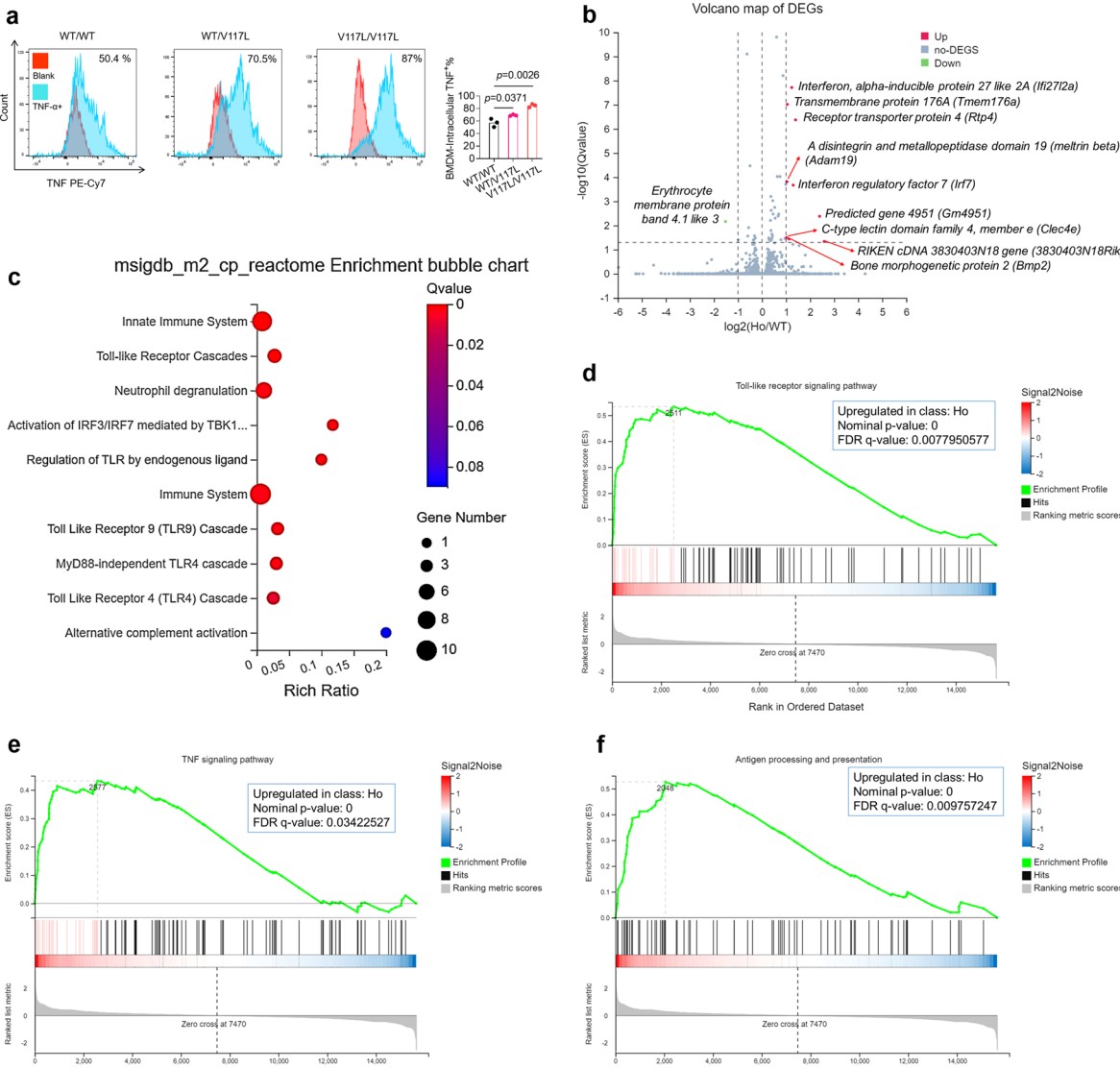

**Extended Data Fig. 9 | DEGs and upregulated pathways of inflammation in UNC93B1$^{V117L}$ homozygous mice compared to wild type.** (**a**) Intracellular cytokine staining of TNF in bone marrow-derived macrophage (BMDM) of indicated mice (n = 3). Red histograms are unstained controls. Statistical analysis was done using unpaired *t*-tests, two-tailed, data are presented as mean with SD. The exact *p* values are shown. (**b**–**f**) RNA sequencing was performed for UNC93B1$^{V117L}$ BMDM compared to control, n = 3 biological replicates. (b) Differentially expressed genes presented as a volcano plot. (c) msigdb_m2_ reactome enrichment analysis of significantly upregulated genes highlights innate immune system and toll-like receptor pathways. (d-f) GSEA significantly associated with TLR signaling (d), TNF signaling (e) and antigen processing/presentation (f). Mice were 8 months old. Ho, homozygotes.

Gating strategy for myeloid cells in bone marrow

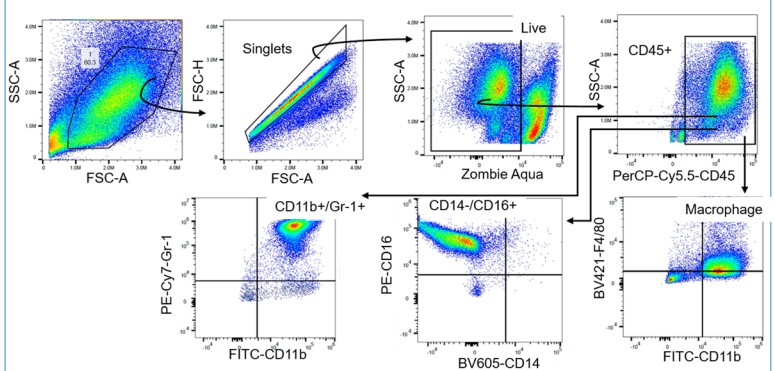

Gating strategy for IFN-γ production in bone marrow CD4 T cells and CD3+/CD4- cells

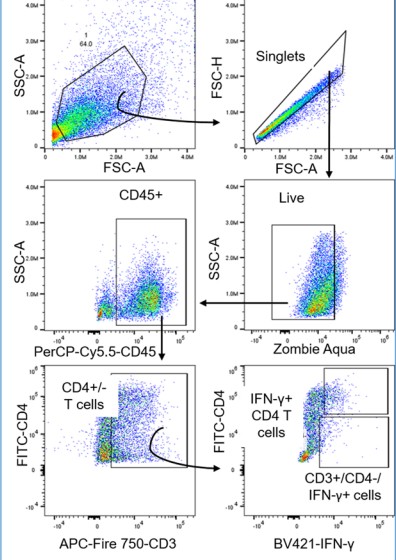

Gating strategy for lymphocytes subsets in bone marrow

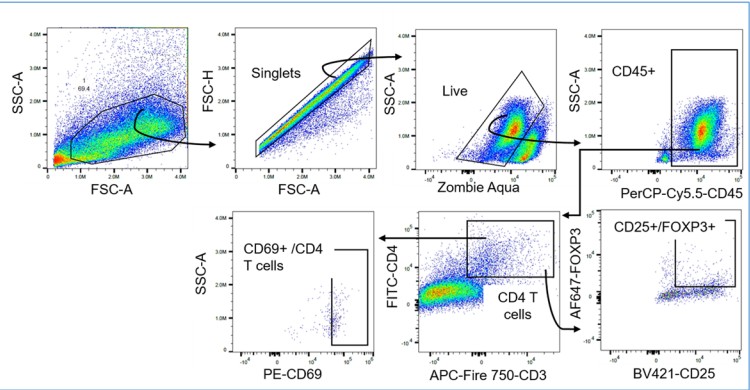

Gating strategy for lymphocytes subsets, CD69+ T cells, germinal center B cells, and CD95+/CXCR5+ B cells in spleen

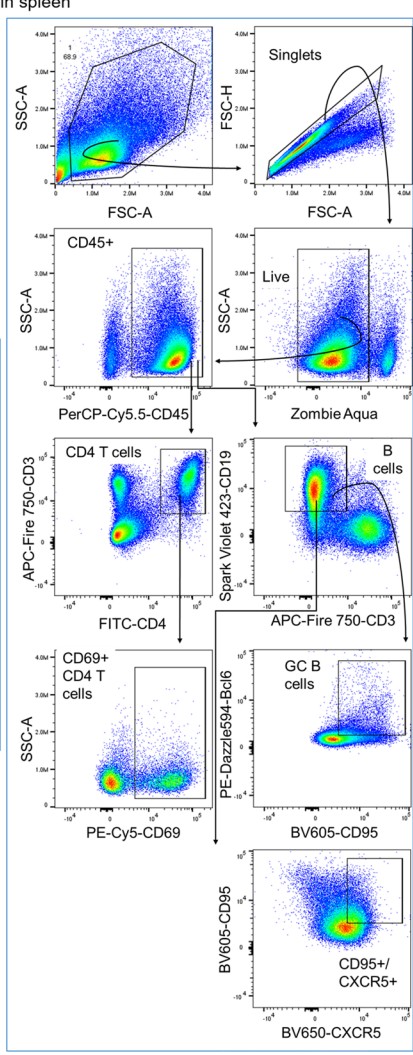

Gating strategy for lymphocytes subsets, B cells, CD4 T cells, T regulatory cells, and CD45+/CD4+/CD44+ cells in spleen

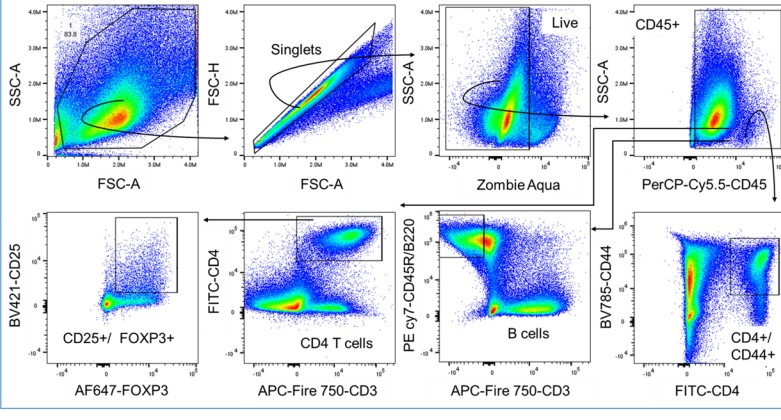

**Extended Data Fig. 10 | Gating strategies.** Representative gating strategies for myeloid cells in bone marrow, macrophage, monocytes, inflammatory Gr-1[+] cells, and IFN-γ producing CD4 T cells and CD3[+]/CD4[−] cells, lymphocytes subsets, B cells, CD4 T cells, T regulatory cells, activated T cells, or/and germinal center B cells in bone marrow or/and spleen. These strategies used for data presented in Fig. 6.

# Reporting Summary

## Statistics

For all statistical analyses, confirm that the following items are present in the figure legend, table legend, main text, or Methods section.

| n/a | Confirmed | |
|---|---|---|
| ☐ | ☒ | The exact sample size (*n*) for each experimental group/condition, given as a discrete number and unit of measurement |
| ☐ | ☒ | A statement on whether measurements were taken from distinct samples or whether the same sample was measured repeatedly |
| ☐ | ☒ | The statistical test(s) used AND whether they are one- or two-sided *Only common tests should be described solely by name; describe more complex techniques in the Methods section.* |
| ☐ | ☒ | A description of all covariates tested |
| ☐ | ☒ | A description of any assumptions or corrections, such as tests of normality and adjustment for multiple comparisons |
| ☐ | ☒ | A full description of the statistical parameters including central tendency (e.g. means) or other basic estimates (e.g. regression coefficient) AND variation (e.g. standard deviation) or associated estimates of uncertainty (e.g. confidence intervals) |
| ☐ | ☒ | For null hypothesis testing, the test statistic (e.g. *F*, *t*, *r*) with confidence intervals, effect sizes, degrees of freedom and *P* value noted *Give P values as exact values whenever suitable.* |
| ☒ | ☐ | For Bayesian analysis, information on the choice of priors and Markov chain Monte Carlo settings |
| ☒ | ☐ | For hierarchical and complex designs, identification of the appropriate level for tests and full reporting of outcomes |
| ☒ | ☐ | Estimates of effect sizes (e.g. Cohen's *d*, Pearson's *r*), indicating how they were calculated |

*Our web collection on statistics for biologists contains articles on many of the points above.*

## Software and code

Policy information about availability of computer code

| Data collection | No special software was used. |
|---|---|
| Data analysis | The sequencing reads for human THP-1 cells were then aligned to human reference genome GRCh38 using STAR aligner (ver. 2.7.9a) 37 to generate the binary alignment file (.bam), which was then passed to HTSeq for gene expression quantification using the htseq-count function (ver. 0.11.1) 38. The resulting gene expression matrix was then normalized and differentially expressed genes (DEGs) were identified using R package DESeq2 (ver. 1.34.0) 39. Genes with adjusted p value less than 0.05 and absolute fold change (log2 space) > 1 were considered as DEGs. This analysis pipeline was used for the volcano plots in Figure 3. Other analysis was performed using BGI software (https://biosys.bgi.com/#/report/login). CYTEK NL-CLC software, SpectroFlo and FlwoJo_v10.8.1 used for analysis of flow cytometry data. Developed membranes of immunoblotting were imaged using the Image Lab detection system (Bio-Rad, USA). Structural analysis software, Pymol V2.5.8. Statistical analysis was performed using Prism 9 software (GraphPad Software Inc.) |

For manuscripts utilizing custom algorithms or software that are central to the research but not yet described in published literature, software must be made available to editors and reviewers. We strongly encourage code deposition in a community repository (e.g. GitHub). See the Nature Portfolio guidelines for submitting code & software for further information.

## Data

Policy information about availability of data

All manuscripts must include a data availability statement. This statement should provide the following information, where applicable:

- Accession codes, unique identifiers, or web links for publicly available datasets
- A description of any restrictions on data availability
- For clinical datasets or third party data, please ensure that the statement adheres to our policy

> The RNA-seq raw sequencing data for THP-1 cell line reported in this study are deposited under supervision and control of the Genome Sequence Archive of the Beijing Institute of Genomics (BIG), Chinese Academy of Sciences (https://ngdc.cncb.ac.cn/gsa-human/), under accession number HRA005592. THP-1 RNA-seq data of this project could be accessed after an approval application. Please refer to GSA (https://ngdc.cncb.ac.cn/gsa-human/) for detailed application guidance. The RNA-seq raw sequencing data for mouse model reported in this study are deposited under supervision and control of the Genome Sequence Archive of BIG), Chinese Academy of Sciences (https://ngdc.cncb.ac.cn/gsa/), under accession number CRA012690. For access to mouse RNA-seq data of this project, please contact the corresponding authors for approval. All other data including raw immunoblot and fluorescence image files, and other in vitro experimental data used to support the findings of this study available in the main text or the supplementary materials.

## Research involving human participants, their data, or biological material

Policy information about studies with human participants or human data. See also policy information about sex, gender (identity/presentation), and sexual orientation and race, ethnicity and racism.

| | |
|---|---|
| Reporting on sex and gender | All 8 patients were female. |
| Reporting on race, ethnicity, or other socially relevant groupings | We found that the gene variant of interest was present in the East Asian population, and is most prevalent in South Coast Han. |
| Population characteristics | Female patients (P) 1-8 at age of 8-26 years respectively presented symptoms of systemic inflammation associated with lupus nephritis. All healthy controls were females with age matched to P2 and P3 age. |
| Recruitment | We retrieved the peripheral blood samples of patients with SLE and healthy control individuals from Clinical Biological Resource Bank of Guangzhou Women and Children's Medical Center. No potential self-selection bias was involved during the sample collection process. |
| Ethics oversight | Written informed consent has been obtained from the authorized individual for all participated patients. The Guangzhou Women and Children's Medical Centre Medical Ethics Committee approved the study procedures ([2021]073B00) in consistence with Helsinki Declaration about ethics in using human samples. |

Note that full information on the approval of the study protocol must also be provided in the manuscript.

# Field-specific reporting

Please select the one below that is the best fit for your research. If you are not sure, read the appropriate sections before making your selection.

☒ Life sciences　　☐ Behavioural & social sciences　　☐ Ecological, evolutionary & environmental sciences

For a reference copy of the document with all sections, see nature.com/documents/nr-reporting-summary-flat.pdf

# Life sciences study design

All studies must disclose on these points even when the disclosure is negative.

| | |
|---|---|
| Sample size | For mouse experiments, at least 3 per group were studied, and more if required to achieve statistical significance. No statistical methods were used to pre-determine sample sizes but our sample sizes are similar to those reported in previous publications. |
| Data exclusions | No data were excluded from the analyses unless technically detected errors. |
| Replication | The experiments were performed by at least three biological replicates. The details were described in the Figure Legends and Methods. |
| Randomization | For animal studies, randomization was performed by the animal facility. |
| Blinding | Blinding was applied for disease scoring of tissues. |

# Behavioural & social sciences study design

All studies must disclose on these points even when the disclosure is negative.

| | |
|---|---|
| Study description | *Briefly describe the study type including whether data are quantitative, qualitative, or mixed-methods (e.g. qualitative cross-sectional, quantitative experimental, mixed-methods case study).* |
| Research sample | *State the research sample (e.g. Harvard university undergraduates, villagers in rural India) and provide relevant demographic information (e.g. age, sex) and indicate whether the sample is representative. Provide a rationale for the study sample chosen. For studies involving existing datasets, please describe the dataset and source.* |
| Sampling strategy | *Describe the sampling procedure (e.g. random, snowball, stratified, convenience). Describe the statistical methods that were used to predetermine sample size OR if no sample-size calculation was performed, describe how sample sizes were chosen and provide a rationale for why these sample sizes are sufficient. For qualitative data, please indicate whether data saturation was considered, and what criteria were used to decide that no further sampling was needed.* |
| Data collection | *Provide details about the data collection procedure, including the instruments or devices used to record the data (e.g. pen and paper, computer, eye tracker, video or audio equipment) whether anyone was present besides the participant(s) and the researcher, and whether the researcher was blind to experimental condition and/or the study hypothesis during data collection.* |
| Timing | *Indicate the start and stop dates of data collection. If there is a gap between collection periods, state the dates for each sample cohort.* |
| Data exclusions | *If no data were excluded from the analyses, state so OR if data were excluded, provide the exact number of exclusions and the rationale behind them, indicating whether exclusion criteria were pre-established.* |
| Non-participation | *State how many participants dropped out/declined participation and the reason(s) given OR provide response rate OR state that no participants dropped out/declined participation.* |
| Randomization | *If participants were not allocated into experimental groups, state so OR describe how participants were allocated to groups, and if allocation was not random, describe how covariates were controlled.* |

# Ecological, evolutionary & environmental sciences study design

All studies must disclose on these points even when the disclosure is negative.

| | |
|---|---|
| Study description | *Briefly describe the study. For quantitative data include treatment factors and interactions, design structure (e.g. factorial, nested, hierarchical), nature and number of experimental units and replicates.* |
| Research sample | *Describe the research sample (e.g. a group of tagged Passer domesticus, all Stenocereus thurberi within Organ Pipe Cactus National Monument), and provide a rationale for the sample choice. When relevant, describe the organism taxa, source, sex, age range and any manipulations. State what population the sample is meant to represent when applicable. For studies involving existing datasets, describe the data and its source.* |
| Sampling strategy | *Note the sampling procedure. Describe the statistical methods that were used to predetermine sample size OR if no sample-size calculation was performed, describe how sample sizes were chosen and provide a rationale for why these sample sizes are sufficient.* |
| Data collection | *Describe the data collection procedure, including who recorded the data and how.* |
| Timing and spatial scale | *Indicate the start and stop dates of data collection, noting the frequency and periodicity of sampling and providing a rationale for these choices. If there is a gap between collection periods, state the dates for each sample cohort. Specify the spatial scale from which the data are taken* |
| Data exclusions | *If no data were excluded from the analyses, state so OR if data were excluded, describe the exclusions and the rationale behind them, indicating whether exclusion criteria were pre-established.* |
| Reproducibility | *Describe the measures taken to verify the reproducibility of experimental findings. For each experiment, note whether any attempts to repeat the experiment failed OR state that all attempts to repeat the experiment were successful.* |
| Randomization | *Describe how samples/organisms/participants were allocated into groups. If allocation was not random, describe how covariates were controlled. If this is not relevant to your study, explain why.* |
| Blinding | *Describe the extent of blinding used during data acquisition and analysis. If blinding was not possible, describe why OR explain why blinding was not relevant to your study.* |

Did the study involve field work? 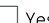 Yes 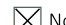 No

# Field work, collection and transport

| | |
|---|---|
| Field conditions | *Describe the study conditions for field work, providing relevant parameters (e.g. temperature, rainfall).* |
| Location | *State the location of the sampling or experiment, providing relevant parameters (e.g. latitude and longitude, elevation, water depth).* |
| Access & import/export | *Describe the efforts you have made to access habitats and to collect and import/export your samples in a responsible manner and in compliance with local, national and international laws, noting any permits that were obtained (give the name of the issuing authority, the date of issue, and any identifying information).* |
| Disturbance | *Describe any disturbance caused by the study and how it was minimized.* |

# Reporting for specific materials, systems and methods

We require information from authors about some types of materials, experimental systems and methods used in many studies. Here, indicate whether each material, system or method listed is relevant to your study. If you are not sure if a list item applies to your research, read the appropriate section before selecting a response.

## Materials & experimental systems

| n/a | Involved in the study |
|---|---|
| ☐ | ☒ Antibodies |
| ☐ | ☒ Eukaryotic cell lines |
| ☒ | ☐ Palaeontology and archaeology |
| ☐ | ☒ Animals and other organisms |
| ☒ | ☐ Clinical data |
| ☒ | ☐ Dual use research of concern |
| ☒ | ☐ Plants |

## Methods

| n/a | Involved in the study |
|---|---|
| ☒ | ☐ ChIP-seq |
| ☐ | ☒ Flow cytometry |
| ☒ | ☐ MRI-based neuroimaging |

## Antibodies

| | |
|---|---|
| Antibodies used | For immunoblots the following antibodies were used: anti-FLAG (F1804, Sigma-Aldrich), anti-NF-κB p65 (8242S, Cell Signaling), anti-Phospho-NF-κB p65 (Ser468) (3039S, Cell Signaling), anti-JNK123 (ab179461, Abcam), anti- Phospho-JNK123 (ab124956, Abcam), anti-P38 (#8690 , Cell Signaling), anti-Phospho-P38 (9211S, Cell Signaling), anti-IRF5 (#20261, Cell Signaling), anti-Phospho- IRF5 (Ser437) (#PA5-64760, Invitrogen), anti-IRAK4 (#4363, Cell Signaling), anti- Phospho-IRAK4 (Thr345/Ser346) (#11927, Cell Signaling), anti-p44/42 MAPK (ERK1/2) (9102S, Cell Signaling), anti- Phospho-p44/42 MAPK (ERK1/2) (Thr202/Tyr204) (9101S, Cell Signaling), anti-IRF7 (#39659, Cell Signaling), anti-Phospho- IRF7 (Ser437/438) (#24129, Cell Signaling), anti-p-Ser/Phosphoserine (sc-81514, Santa Cruz Biotechnology), anti-β-actin (AC026, ABclonal), anti-rabbit IgG, HRP-linked antibody (7074, Cell Signaling), and anti-mouse IgG, HRP-linked antibody (7076, Cell Signaling). The dilution of anti-β-actin, anti-rabbit IgG HRP-linked antibody, anti-mouse IgG HRP-linked antibody, and anti-FLAG was 1:5000. The dilution of anti-p-Ser/Phosphoserine was 1:1000. The dilution for other primary antibodies was 1:1500.<br><br>For flow cytometry the following antibodies were used: anti-mouse TNF-α (506324, 1:200, Biolegend), anti-mouse IFN-γ (505830, 1:400,  Biolegend), anti-mouse CD45 (103132, 1:200, Biolegend), anti-mouse F4/80 (123137, 1:200,  Biolegend), anti-mouse/human CD11b (101205, 1:500, Biolegend), anti-mouse CD14 (123335, 1:200, Biolegend), anti-mouse CD16 (158004, 1:100, Biolegend), anti-mouse CD3 (100248, 1:100 Biolegend), anti-mouse CD4 (100406, 1:400, Biolegend), anti-mouse FOXP3 (126407, 1:200, Biolegend), anti-mouse CD25 (102043, 1:300, Biolegend), anti-mouse/human CD44 (103059, 1:100, Biolegend), anti-mouse CD69 (104507 or 104510, 1:200 , Biolegend), anti-mouse Gr-1 (108416, 1:200, Biolegend), anti-mouse CD19 (159812, 1:400, Biolegend), anti-mouse CD95 (152612, 1:200, Biolegend), anti- mouse/human Bcl-6 (358510, 1:100, Biolegend), anti- mouse/human CD45R/B220 (103222, 1:200, , Biolegend), and anti-mouse CXCR5 (145517, 1:100, Biolegend). |
| Validation | All antibodies used in our study are commercial and any validation statements are noted on manufacture's website. |

## Eukaryotic cell lines

Policy information about cell lines and Sex and Gender in Research

| | |
|---|---|
| Cell line source(s) | THP-1 cells were obtained from American Type Culture Collection (ATCC) and cultured accordingly. |
| Authentication | The authentication procedures for cell line were done by the supplier. |
| Mycoplasma contamination | All cells used in our study were confirmed that had no Mycoplasma contamination. |

| Commonly misidentified lines<br>(See ICLAC register) | No misidentified cell lines were used in our study. |
|---|---|

# Palaeontology and Archaeology

| Specimen provenance | *Provide provenance information for specimens and describe permits that were obtained for the work (including the name of the issuing authority, the date of issue, and any identifying information). Permits should encompass collection and, where applicable, export.* |
|---|---|
| Specimen deposition | *Indicate where the specimens have been deposited to permit free access by other researchers.* |
| Dating methods | *If new dates are provided, describe how they were obtained (e.g. collection, storage, sample pretreatment and measurement), where they were obtained (i.e. lab name), the calibration program and the protocol for quality assurance OR state that no new dates are provided.* |

☐ Tick this box to confirm that the raw and calibrated dates are available in the paper or in Supplementary Information.

| Ethics oversight | *Identify the organization(s) that approved or provided guidance on the study protocol, OR state that no ethical approval or guidance was required and explain why not.* |
|---|---|

Note that full information on the approval of the study protocol must also be provided in the manuscript.

# Animals and other research organisms

Policy information about studies involving animals; ARRIVE guidelines recommended for reporting animal research, and Sex and Gender in Research

| Laboratory animals | C57BL/6J mice lines (UNC93B1WT/WT, UNC93B1WT/V117L, and UNC93B1V117L/V117L). Mice were age matched 8-14 months old. |
|---|---|
| Wild animals | No wild animals were used in the study. |
| Reporting on sex | All mice used for analysis in this project are females. |
| Field-collected samples | No field collected samples were used in the study. |
| Ethics oversight | Animal studies were approved by Institutional Animal Care and Use Committee of Guangzhou Medical University (GY2022-035) |

Note that full information on the approval of the study protocol must also be provided in the manuscript.

# Clinical data

Policy information about clinical studies
All manuscripts should comply with the ICMJE guidelines for publication of clinical research and a completed CONSORT checklist must be included with all submissions.

| Clinical trial registration | *Provide the trial registration number from ClinicalTrials.gov or an equivalent agency.* |
|---|---|
| Study protocol | *Note where the full trial protocol can be accessed OR if not available, explain why.* |
| Data collection | *Describe the settings and locales of data collection, noting the time periods of recruitment and data collection.* |
| Outcomes | *Describe how you pre-defined primary and secondary outcome measures and how you assessed these measures.* |

# Dual use research of concern

Policy information about dual use research of concern

## Hazards

Could the accidental, deliberate or reckless misuse of agents or technologies generated in the work, or the application of information presented in the manuscript, pose a threat to:

| No | Yes | |
|----|-----|---|
| ☐ | ☐ | Public health |
| ☐ | ☐ | National security |
| ☐ | ☐ | Crops and/or livestock |
| ☐ | ☐ | Ecosystems |
| ☐ | ☐ | Any other significant area |

## Experiments of concern

Does the work involve any of these experiments of concern:

| No | Yes | |
|----|-----|---|
| ☐ | ☐ | Demonstrate how to render a vaccine ineffective |
| ☐ | ☐ | Confer resistance to therapeutically useful antibiotics or antiviral agents |
| ☐ | ☐ | Enhance the virulence of a pathogen or render a nonpathogen virulent |
| ☐ | ☐ | Increase transmissibility of a pathogen |
| ☐ | ☐ | Alter the host range of a pathogen |
| ☐ | ☐ | Enable evasion of diagnostic/detection modalities |
| ☐ | ☐ | Enable the weaponization of a biological agent or toxin |
| ☐ | ☐ | Any other potentially harmful combination of experiments and agents |

# Plants

| | |
|---|---|
| Seed stocks | *Report on the source of all seed stocks or other plant material used. If applicable, state the seed stock centre and catalogue number. If plant specimens were collected from the field, describe the collection location, date and sampling procedures.* |
| Novel plant genotypes | *Describe the methods by which all novel plant genotypes were produced. This includes those generated by transgenic approaches, gene editing, chemical/radiation-based mutagenesis and hybridization. For transgenic lines, describe the transformation method, the number of independent lines analyzed and the generation upon which experiments were performed. For gene-edited lines, describe the editor used, the endogenous sequence targeted for editing, the targeting guide RNA sequence (if applicable) and how the editor was applied.* |
| Authentication | *Describe any authentication procedures for each seed stock used or novel genotype generated. Describe any experiments used to assess the effect of a mutation and, where applicable, how potential secondary effects (e.g. second site T-DNA insertions, mosiacism, off-target gene editing) were examined.* |

# ChIP-seq

## Data deposition

☐ Confirm that both raw and final processed data have been deposited in a public database such as GEO.

☐ Confirm that you have deposited or provided access to graph files (e.g. BED files) for the called peaks.

| | |
|---|---|
| Data access links<br>May remain private before publication. | *For "Initial submission" or "Revised version" documents, provide reviewer access links.  For your "Final submission" document, provide a link to the deposited data.* |
| Files in database submission | *Provide a list of all files available in the database submission.* |
| Genome browser session<br>(e.g. UCSC) | *Provide a link to an anonymized genome browser session for "Initial submission" and "Revised version" documents only, to enable peer review.  Write "no longer applicable" for "Final submission" documents.* |

## Methodology

| | |
|---|---|
| Replicates | *Describe the experimental replicates, specifying number, type and replicate agreement.* |
| Sequencing depth | *Describe the sequencing depth for each experiment, providing the total number of reads, uniquely mapped reads, length of reads and whether they were paired- or single-end.* |
| Antibodies | *Describe the antibodies used for the ChIP-seq experiments; as applicable, provide supplier name, catalog number, clone name, and lot number.* |
| Peak calling parameters | *Specify the command line program and parameters used for read mapping and peak calling, including the ChIP, control and index files used.* |

| Data quality | *Describe the methods used to ensure data quality in full detail, including how many peaks are at FDR 5% and above 5-fold enrichment.* |
|---|---|
| Software | *Describe the software used to collect and analyze the ChIP-seq data. For custom code that has been deposited into a community repository, provide accession details.* |

# Flow Cytometry

## Plots

Confirm that:

☒ The axis labels state the marker and fluorochrome used (e.g. CD4-FITC).

☒ The axis scales are clearly visible. Include numbers along axes only for bottom left plot of group (a 'group' is an analysis of identical markers).

☒ All plots are contour plots with outliers or pseudocolor plots.

☒ A numerical value for number of cells or percentage (with statistics) is provided.

## Methodology

| Sample preparation | The whole spleen was minced into tiny pieces in 6-well plate on ice using plunger end of the syringe and filtered by 70 µm strainer with rinsing by 5mL PBS with 2% FBS. The collected tissue filtrates were centrifuged at 500g /5 minutes at 4°C . The cells pallet suspended in RBC lysis buffer and incubated for 5 minutes at 25°C and then mixed with double volume of PBS with 2% FBS. The immune cells were collected in the pellet after centrifugation at 500g /5 minutes at 4°C .<br>Collected bone marrow from mouse femur bones by centrifuging using technique of two layers of tubes were incubated with RBC lysis buffer for 5 minutes at 25°C and then mixed with double volume of PBS with 2% FBS. The immune cells were collected in the pellet after centrifugation at 500g /5 minutes at 4°C.<br>For flow cytometry, 1 million cells were incubated with Zombie Aqua-A (Biolegend) for 15 min in dark, then washed to be stained by antibodies for cell surface markers and incubated for 30 min at 4°C and then washed. Fixation and permeabilization kit (88-8824-00, Thermofisher) was used for fixation and intracellular permeablization for intracellular and nuclear markers according to manufacturer's protocol. |
|---|---|
| Instrument | CYTEK NL-CLC |
| Software | SpectroFlo and FlwoJo_v10.8.1. |
| Cell population abundance | Gating strategies in the Extended Data Fig. 10 |
| Gating strategy | The Extended Data Fig. 10 is the Representative gating strategies for myeloid cells in bone marrow, macrophage, monocytes, inflammatory Gr-1+ cells, and IFN-γ producing CD4 T cells and CD3+/CD4- cells, lymphocytes subsets, B cells, CD4 T cells, T regulatory cells, activated T cells, or/and germinal center B cells in bone marrow or/and spleen. These strategies used for data presented in figure 6. |

☒ Tick this box to confirm that a figure exemplifying the gating strategy is provided in the Supplementary Information.

# Magnetic resonance imaging

## Experimental design

| Design type | *Indicate task or resting state; event-related or block design.* |
|---|---|
| Design specifications | *Specify the number of blocks, trials or experimental units per session and/or subject, and specify the length of each trial or block (if trials are blocked) and interval between trials.* |
| Behavioral performance measures | *State number and/or type of variables recorded (e.g. correct button press, response time) and what statistics were used to establish that the subjects were performing the task as expected (e.g. mean, range, and/or standard deviation across subjects).* |

## Acquisition

| Imaging type(s) | *Specify: functional, structural, diffusion, perfusion.* |
|---|---|
| Field strength | *Specify in Tesla* |
| Sequence & imaging parameters | *Specify the pulse sequence type (gradient echo, spin echo, etc.), imaging type (EPI, spiral, etc.), field of view, matrix size, slice thickness, orientation and TE/TR/flip angle.* |
| Area of acquisition | *State whether a whole brain scan was used OR define the area of acquisition, describing how the region was determined.* |

Diffusion MRI ☐ Used ☐ Not used

## Preprocessing

**Preprocessing software**
*Provide detail on software version and revision number and on specific parameters (model/functions, brain extraction, segmentation, smoothing kernel size, etc.).*

**Normalization**
*If data were normalized/standardized, describe the approach(es): specify linear or non-linear and define image types used for transformation OR indicate that data were not normalized and explain rationale for lack of normalization.*

**Normalization template**
*Describe the template used for normalization/transformation, specifying subject space or group standardized space (e.g. original Talairach, MNI305, ICBM152) OR indicate that the data were not normalized.*

**Noise and artifact removal**
*Describe your procedure(s) for artifact and structured noise removal, specifying motion parameters, tissue signals and physiological signals (heart rate, respiration).*

**Volume censoring**
*Define your software and/or method and criteria for volume censoring, and state the extent of such censoring.*

## Statistical modeling & inference

**Model type and settings**
*Specify type (mass univariate, multivariate, RSA, predictive, etc.) and describe essential details of the model at the first and second levels (e.g. fixed, random or mixed effects; drift or auto-correlation).*

**Effect(s) tested**
*Define precise effect in terms of the task or stimulus conditions instead of psychological concepts and indicate whether ANOVA or factorial designs were used.*

Specify type of analysis: ☐ Whole brain ☐ ROI-based ☐ Both

**Statistic type for inference**
*Specify voxel-wise or cluster-wise and report all relevant parameters for cluster-wise methods.*

(See Eklund et al. 2016)

**Correction**
*Describe the type of correction and how it is obtained for multiple comparisons (e.g. FWE, FDR, permutation or Monte Carlo).*

## Models & analysis

| n/a | Involved in the study |
|---|---|
| ☐ | ☐ Functional and/or effective connectivity |
| ☐ | ☐ Graph analysis |
| ☐ | ☐ Multivariate modeling or predictive analysis |

**Functional and/or effective connectivity**
*Report the measures of dependence used and the model details (e.g. Pearson correlation, partial correlation, mutual information).*

**Graph analysis**
*Report the dependent variable and connectivity measure, specifying weighted graph or binarized graph, subject- or group-level, and the global and/or node summaries used (e.g. clustering coefficient, efficiency, etc.).*

**Multivariate modeling and predictive analysis**
*Specify independent variables, features extraction and dimension reduction, model, training and evaluation metrics.*

