## [Peer Review File · Nature Immunology]

Peer Review Information

Journal: Nature Immunology

Manuscript Title: Genetic variants in UNC93B1 predispose to childhood-onset systemic lupus erythematosus

Corresponding author name(s): Dr Seth Masters, Dr Yuxia Zhang

Reviewer Comments & Decisions:

Decision Letter, initial version:
--

25th Oct 2023

Dear Seth,

Thank you for providing a point-by-point response to the referees' comments on your manuscript entitled, "Genetic variants in UNC93B1 predispose to childhood-onset systemic lupus erythematosus". As noted previously, while they find your work of considerable potential interest, they have raised quite substantial concerns that must be addressed. In light of these comments, we cannot accept the current manuscript for publication, but would be very interested in considering a revised version that addresses these serious concerns.

We invite you to submit a substantially revised manuscript, however please bear in mind that we will be reluctant to approach the referees again in the absence of major revisions.

Specifically, the revision should include new experiments to address:

- (1) examine p-Ser phosphorylation status of UNC93B
- (2) please examine proteinuria/albumin, blood urea concentrations, & serum creatinine in mice
- (3) please report inflammatory cytokine expression (baseline and TLR ligand-stimulated) for patient B cells & healthy controls
- (4) examine protein expression and phosphorylation status of IRF3 and IRF7
- (5) perform dose titrations for IRAK4 inhibitors used in this study and measure p-IRAK4 by immunoblot

Would be of interest to follow-up with patient family members give that in the mouse model heterozygotes have disease manifestations.

ED notes:

- (a) please keep Figures 5 and 6 separate (overruling referee #1).

- (b) Figure 6b, need higher magnification insets for areas where green and red arrows are depicted. Suggest Fig. 6b should be broken up into separate panels for each tissue type.
- (c) Reviewer figure 3 supplied in response letter - can immunohistochemistry of tissue sections be depicted instead for complement and IgG deposition?? Worried about autofluorescence in panels a,b & contrast settings as the images appear to be quite variable, especially for the hets.
- (d) Figs. 3b, 4b & 7b, should depict NF-kBp65 window as same region as the p-NF-kBp65 as shown in the gel slice above, ie include region of 75 kDa. Please provide full blots as source data – all immunoblots.
- (e) please show a negative control using healthy human sera for staining HEp2 cells (for reviewer figure 10).

Please include the additional textual clarifications as indicated in your response letter.

When you revise your manuscript, please take into account all reviewer and editor comments, please highlight all changes in the manuscript text file in Microsoft Word format.

- * Include a "Response to referees" document detailing, point-by-point, how you addressed each referee comment. If no action was taken to address a point, you must provide a compelling argument. This response will be sent back to the referees along with the revised manuscript.
- * If you have not done so already please begin to revise your manuscript so that it conforms to our Article format instructions at <http://www.nature.com/ni/authors/index.html>. Refer also to any guidelines provided in this letter.
- * Include a revised version of any required reporting checklist. It will be available to referees (and, potentially, statisticians) to aid in their evaluation if the manuscript goes back for peer review. A revised checklist is essential for re-review of the paper.

The Reporting Summary can be found here:

When submitting the revised version of your manuscript, please pay close attention to our [href="https://www.nature.com/nature-portfolio/editorial-policies/image-integrity">Digital Image Integrity Guidelines](https://www.nature.com/nature-portfolio/editorial-policies/image-integrity). and to the following points below:

[REDACTED]

If you wish to submit a suitably revised manuscript we would hope to receive it within 6 months. If you cannot send it within this time, please let us know. We will be happy to consider your revision so long as nothing similar has been accepted for publication at Nature Immunology or published elsewhere.

Nature Immunology is committed to improving transparency in authorship. As part of our efforts in this direction, we are now requesting that all authors identified as 'corresponding author' on published papers create and link their Open Researcher and Contributor Identifier (ORCID) with their account on the Manuscript Tracking System (MTS), prior to acceptance. ORCID helps the scientific community achieve unambiguous attribution of all scholarly contributions. You can create and link your ORCID from the home page of the MTS by clicking on 'Modify my Springer Nature account'. For more information please visit www.springernature.com/orcid.

Thank you for the opportunity to review your work.

Kind regards,

Laurie

Laurie A. Dempsey, Ph.D.
Senior Editor
Nature Immunology
l.dempsey@us.nature.com
ORCID: 0000-0002-3304-796X

Referee expertise:

Referee #1: Human autoimmune diseases/SLE

Referee #2: TLR signaling

Referee #3: Inflammation & autoimmunity

Reviewers' Comments:

Reviewer #1:

Remarks to the Author:

A new variant is presented that leads to lupus-like disease in people. The variant enables TLR signaling. Inserted in to mice it causes systemic inflammation and features of systemic autoimmunity.

The information is exciting as a new variant enters the expanding list of the so-called "monogenic lupus". A number of points require clarification.

1. The authors should confirm data reported in ref 9 (UNC93B1 expression is increased in SLE patients with active disease).
2. The novel variant "...novel variant UNC93B1 (c.A940G p.T314A)" should be explained. What is c and p.
3. Does T314A alter S547 and S550 phosphorylation.
4. p.3 "we found that 7 encode UNC93B1 (c.G349T p.V117L) which lies at the interface between UNC93B1 protomers in the reported structure" is not clear.
5. p.4 this text "Although these protomers are close to one another, there are no noticeable interactions that would prevent dimerisation of TLR7, similar to what occurs for TLR3 13. Instead, V117L may act in a similar way to K333R, which is not proximal, but is also located at an UNC93B1 protomer interface and results in increased TLR7 activation. However as K333 is ubiquitinated this could represent a different mechanism of activation" is obscure.
6. p.3 "We first identified the novel variant UNC93B1 (c.A940G p.T314A), which is not present in the general population, and is highly conserved (Fig. 1a)" the 314 variant was identified first. In the next page it is indicated that this was present in the fourth patient.
7. "For the seven SLE patients identified to carry this allele, there were no immediate reports of affected family members, although follow-up studies would be required to confirm this" but the discussion refers to 4 patients.
8. Fig 3e and 3f should be replaced with panels from the suppl Fig 3 to clarify the gene sets that are altered. 3e and 3f are not informative as presented.
9. "PBMCs stimulated by R848" should insert one word to inform the readers what R848 is.
10. "Heterozygous and homozygous UNC93B1 V138L mice were born at normal Mendelian ratios and initially appear healthy, however they lose weight (Fig. 5a), develop". Which develop autoimmunity? The hets or the homos develop disease?
11. "increased along with CD16+/CD14- cells, regulatory T cells, and activated T cells (Fig. 5e)" are the Tregs bona fide Tregs? May be they produce proinflammatory cytokines. This can be tested.
12. The Knock in mice develop systemic inflammation with some autoimmune features. Figures 5 and 6 should be merged.
13. Fig 6b. the provided images are not convincing that there is glomerular disease. Also, since the mice develop systemic inflammation, is there inflammation in the interstitium?
14. Did the mice have proteinuria? Was there any Ig or complement deposition?
15. The authors most probably have WGS data and someone wonders whether other variants coexisted.

Reviewer #2:

Remarks to the Author:

The authors have described that two UNC93B1 mutations, found in several SLE patients in China,

spontaneously activated intracellular inflammatory signal cascades and induced the production of inflammatory cytokines and type I IFNs. They used THP-1 cells and the UNC93B1 overexpression system. Furthermore, they established Unc93B1 knock-in mice with orthologous mutation, and the mice were shown spontaneous systemic inflammation. These new findings are very interesting. However, the present manuscript is not sufficient to investigate the pathogenicity of the UNC93B1 mutations in SLE. Furthermore, the reviewer did not understand how to establish the knock-in mice. From the above, it is strongly required to organize a lot of data and reconsider the manuscript to be published in this journal.

The reviewer will seek corrections and reconsiderations on the following specific points.

1. In Fig. 3 c and d, the background quantity (WT expression column) of cytokine production is too high, and the induction level (WT vs mutants) is too low. Could you stimulate the cells with R848 or the other TLR7 ligands, such as Loxoribine and RNA ligands?
2. In Fig. 3 and 4, THP-1 is a monocyte cell line. However, in SLE pathogenesis, plasmacytoid dendritic cells, conventional dendritic cells, and B cells are more important than monocytes. How about these cell lines? Can you get similar results?
3. In Fig. 4e-g, they used R848 as a TLR7 ligand. But R848 could stimulate both TLR7 and TLR8. They should validate which receptor contributed to the immune responses.
4. In the Unc93B1 knock-in mice experiment, why did they choose the 138th amino acid for the mutation? Human and mouse UNC93B1 amino acid sequences are very similar. The percentage of homology is about 90%, and V117 is conserved between humans and mice (Fig. 1a). Moreover, the reviewer could not find V138 on an amino acid sequence of mouse Unc93B1. Is the 138th amino acid methionine? Please confirm the important point.
5. They should show an establishment strategy of Unc93B1 knock-in mice.
6. In Fig. 5-7, is the lupus-like inflammation dependent on TLR7? Please analyze TLR7 deficiency in the knock-in mice.
7. In Fig. 5f, a major producer of IFN- γ is CD4+ or CD8 α + T cells. Please replace it with dot graphs for T cells.
8. In Fig. 5f, how about the percentage of monocytes, macrophages, and dendritic cells in the spleen?
9. They should measure blood urea nitrogen and serum creatinine.
10. They should show urinary albumin and creatinine ratio.
11. In Fig. 6b, how about depositions of immune complexes such as IgG and C3?
12. In Fig. 7b, how about the activation of IRF3 and IRF7? TLR7 could induce both IRFs phosphorylation.

Reviewer #3:

Remarks to the Author:

Title: Genetic variants in UNC93B1 predispose to childhood-onset systemic lupus erythematosus

Al-Azab et al integrated clinical and mechanistical findings to discover and explain two novel mutations directly associated with the development of childhood-onset SLE. Ploegh's lab and others have shown that Unc93b can bind and traffic toll-like receptors (TLRs) from the ER to subcellular locations for ligand sensing. Through human genetic studies (WGS), Al-Azab et al discovered two mutations in the UNC93B1 gene, a TLR7 chaperone: First, the V317A mutation in a single patient with childhood-onset SLE, and later, the V117L mutation in a set of childhood-onset SLE patients from a southeast Asian cohort. They modeled the later mutation in THP1 monocyte cell line, and in a murine model. Authors described an in vitro mechanism of action by which UNC93B1 mutant THP1 cells exhibit increased production of proinflammatory markers in response to ssDNA, a TLR7 ligand. Although the specifics of the molecular mechanism were not fully described, the authors demonstrated that increased TLR7 action maybe due to an increase of TLR7 in the endosomes, and that IRAK1/4 signaling is a key mediator of this inflammatory process. Using a murine genetic model, authors claim that a SLE-like immune-pathology can be manifested in mice.

While aspects of this study advance our understanding of UNC93B1 biology and potentially SLE, however, the study lacks clear mechanistic as well as an accurate representation of genetic/clinical features in this small cohort.

Comment to the authors:

1.- The analysis of patient cohort is limited to 3 patients and apparently all parents were normal with no SLE. UNC93B1 is on X-chromosome which should help to see whether there is a clear track of SLE disease in family. Authors do not have normal controls for several assays in Fig.1 making it difficult to compare normal versus disease differences including Fig.1a. Given the lack of disease in parents, there are clearly other genetic factors that manifest the disease which authors do not comment. The data for serum cytokines/chemokines are limited to only select patients not all patients in the cohort. It is unclear whether increased cytokines are due to this specific mutation versus ongoing fever or other clinical complications in these patients. Authors could have used TLR7 agonist or other stimuli to directly show the sensitivity of this pathway is upregulated via these mutations.

2.- Authors use THP1 monocytic cell line to model and to understand the mechanistic pathways involved in the role of UNC93B1 and TLR7, however, several controls are missing to ensure authors do not over-interpret this line of studies. For example, it is not clear whether mutation or level of expression is the driver of their phenotype as authors do not show any control to evaluate the expression of various mutants. UNC93B1 is expressed in THP1. Authors omit the fact that THP1 cell line was derived from a male patient whereas all of their patients in their cohort were female. A better cell-based system with viral transformation or utilizing CRISPR might have been a better approach to model these mutations.

3. Authors use a non-selective IRAKi at high concentration (5 μ M) to suggest its utility as a therapeutic in SLE. Authors use high concentration even above IC90 with no titration of the compd in the current data. It is hard to conclude that their pathway is directly linked to IRAK1//4. Several IRAK4 selective small molecule inhibitors from Pfizer (PMCID: PMC8671219), Genentech (PMID: 32487715) or BMS (PMCID: PMC5253435) could be considered to ensure this pathway is TLR7/IRAK

dependent. Constitutive IRAK activations could have also been directly documented by looking at pIRAK4 (see above publications). Authors could have also used CRISPR to knockout IRAK4 in their cell line system.

4.- Authors over-interpretate the phenotype of the *unc93b1* mutant mice. In Fig.5 a number of homozygous mutant mice have a normal weight liver, yet authors show an outlier. Authors should compare and report the functional differences in younger age 4-6 weeks old using TLR7/8 agonist stimulations in vivo or in BMDM.

Minor concerns:

1.- In suppl. Fig 1 (b,c), these images would be better interpreted/appreciated if they add a negative control staining and they quantify the magnitude of difference between background and actual signal.

2.- In lines 92-94, the authors claim that supp fig 1A shows the complete blood count for P2,P3 & P4. However, the current version of that figure only shows the complement levels, the SLEDAI-2K, and the ANA antibodies. They are missing the piece of that that they are referring in this section.

3.- In figs 2g-j and associate supplementary figure 2, the authors measured key lupus-associated cytokines produced by P3 and P4-derived PBMCs. Furthermore, they claim in line 97 that these cytokines were spontaneously expressed by the isolated PBMCs, and in line 100, they related some of these cytokines to TLR7-induced cytokines. Could the authors better describe the handling and treatment of these PBMCs. Particularly, did they measure cytokines right after isolation, or did they have to culture them for some time (how long)? In what media? Did they stimulate these PBMCs somehow?

4.- In line 175, the authors refer "supp fig. 3d-f" when I think they refer to "Supp Fig. 4d-f".

Author Rebuttal to Initial comments

Genetic variants in *UNC93B1* predispose to childhood-onset systemic lupus erythematosus

Proposed Response to Reviewers Comments

Referee #1 (comments to authors)

A new variant is presented that leads to lupus-like disease in people. The variant enables TLR signaling. Inserted in to mice it causes systemic inflammation and features of systemic autoimmunity.

The information is exciting as a new variant enters the expanding list of the so-called "monogenic lupus". A number of points require clarification.

1. The authors should confirm data reported in ref 9 (UNC93B1 expression is increased in SLE patients with active disease).

To address this question we generated immortalized B-cells from UNC93B1 patients P2 and P3, then determined the expression of UNC93B1 mRNA and protein by qPCR and western blot respectively. Compared to healthy control B-cells, there was not a significant increase in UNC93B1 expression (Review Fig 1). Given that Nakano et al. found that UNC93B1 expression was only upregulated during active disease, this is likely a result, rather than a cause, of disease.

Review Figure 1. (a) Quantitative RT-PCR analysis of UNC93B1 expression in the indicated immortalized B cell lines. (b) Levels of UNC93B1 as measured by immunoblot, in the lysates of the immortalized B-cells from UNC93B1 patients, P2 and P3, and healthy control. Data are representative of three independent experiments.

2. The novel variant “..novel variant UNC93B1 (c.A940G p.T314A)” should be explained. What is c and p.

We added this explanation to the text (Manuscript New supplementary table 2), c. is the coding change and p. is the protein change.

3. Does T314A alter S547 and S550 phosphorylation.

To examine phosphorylation of UNC93B1 we performed immunoprecipitation of overexpressed UNC93B1 followed by pSer immunoblotting, or Flag immunoblotting on Phos-tag gels for whole cells lysates (WCL) and immunoprecipitates (Review Figure 2a). Also, we ran lysates of patient cells on Phos-tag gels followed by blotting for UNC93B1 (Review Figure 2b). In this analysis we did not see a significant difference of UNC93B1 phosphorylation due to the patient mutations.

Review Figure 2. (a) Immunoprecipitation and immunoblot of UNC93B1 phosphorylation using Ph-tag gel and P-ser antibody as indicated in THP-1 cells. (b) Levels of UNC93B1 as measured by immunoblot, in the lysates of the immortalized B-cells from mutated UNC93B1 patients, P2 and P3, and healthy control using Ph-tag gel.

4. p.3 “we found that 7 encode UNC93B1 (c.G349T p.V117L) which lies at the interface between UNC93B1 protomers in the reported structure” is not clear.

We updated the text to address this comment.

5. p.4 this text “Although these protomers are close to one another, there are no noticeable interactions that would prevent dimerisation of TLR7, similar to what occurs for TLR3 13. Instead, V117L may act in a similar way to K333R, which is not proximal, but is also located at an UNC93B1 protomer interface and results in increased TLR7 activation. However as K333 is ubiquitinated this could represent a different mechanism of activation” is obscure.

We made this more clear in the revised manuscript.

6. p.3 “We first identified the novel variant UNC93B1 (c.A940G p.T314A), which is not present in the general population, and is highly conserved (Fig. 1a)” the 314 variant was identified first. In the next page it is indicated that this was present in the fourth patient.

We re-wrote this section to improve consistency.

7. “For the seven SLE patients identified to carry this allele, there were no immediate reports of affected family members, although follow-up studies would be required to confirm this” but the discussion refers to 4 patients.

We have now included clinical data for all 7 patients that is available to us (Manuscript New Extended Data Figure 1 and Table 3).

8. Fig 3e and 3f should be replaced with panels from the suppl Fig 3 to clarify the gene sets that are altered. 3e and 3f are not informative as presented.

We replaced Fig 3e and 3f as recommended (Manuscript Figure 3d and 3e), thanks.

9. “PBMCs stimulated by R848” should insert one word to inform the readers what R848 is.

We have modified the text (results and methods sections) to state that this is a synthetic ligand for TLR7/8.

10. “Heterozygous and homozygous UNC93B1 V138L mice were born at normal Mendelian ratios and initially appear healthy, however they lose weight (Fig. 5a), develop”. Which develop autoimmunity? The hets or the homos develop disease?

Heterozygous UNC93B1^{WT/V138L} mice and homozygotes both develop autoimmune disease, although this is more severe in homozygotes. We made this clear in the text.

11. “increased along with CD16+/CD14- cells, regulatory T cells, and activated T cells (Fig. 5e)” are the Tregs bona fide Tregs? May be they produce proinflammatory cytokines. This can be tested.

Please see the gating strategy in Manuscript New Extended Data Figure 10, identifying CD3/CD4/CD25/Foxp3 positive cells which appear to be bona fide Tregs. Referee 2 also asked about proinflammatory cytokine production from T-cells, and we now show that this is increased for CD4+ve T cells and CD3+ve/CD4-ve cells in the bone marrow (Manuscript New Figure 5f).

12. The Knock in mice develop systemic inflammation with some autoimmune features. Figures 5 and 6 should be merged.

Thank you for recommendation, we will merge Figures 5 and 6 if the journal can fit them together.

13. Fig 6b. the provided images are not convincing that there is glomerular disease. Also, since the mice develop systemic inflammation, is there inflammation in the interstitium?

For this mouse model we did not see obvious signs of inflammation in the interstitium, based on Haematoxylin and Eosin (H&E) staining of the kidneys (Manuscript Figure 6b).

14. Did the mice have proteinuria? Was there any Ig or complement deposition?

Although some UNC93B1^{V117L} mice do show a small increase in complement 3 (Review Fig. 3a), IgG (Review Fig. 3b), and IgM (Review Fig. 3c) deposition, this was generally not statistically significant. Similarly, there was not a significant increase in proteinuria for this mouse strain at the ages we tested (Review Figure 4).

Review Figure 3. Glomerular deposition of immune complexes in UNC93B1^{V117L} mice. (a) complement 3. (b) IgG. (c) IgM, blue (DAPI), green (IgM), $n=4-10$ as indicated. Mice were age matched 8 months.

Review Figure 4. Levels of serum creatinine and blood urea nitrogen, and urine creatinine, protein, and albumin of indicated mice. Mice were age matched 14 months.

15. The authors most probably have WGS data and someone wonders whether other variants coexisted.

We performed whole exome sequencing on the early-onset patients and examined the genes that were reported as responsible for lupus. For this reason, we do not have data on common lupus-associated risk factor SNPs which typically are not in coding regions of the genome. Instead, we checked for potential incidence of variants co-existing in genes associated with Primary Immune Diseases (PID) based on the findings of an early-onset lupus cohort (PMID: 36586539)¹. We did not observe any likely pathogenic mutations in known PID genes for the patients carrying UNC93B1 mutations reported in this manuscript.

1- Wu CY, Fan WL, Yang HY, Chu PS, Liao PC, Chen LC, Yao TC, Yeh KW, Ou LS, Lin SJ, Lee WI, Huang JL. Contribution of genetic variants associated with primary immunodeficiencies to childhood-onset systemic lupus erythematosus. *J Allergy Clin Immunol.* 2023 Apr;151(4):1123-1131. doi: 10.1016/j.jaci.2022.12.807. Epub 2022 Dec 28. PMID: 36586539.

Referee #2 (comments to authors)

The authors have described that two UNC93B1 mutations, found in several SLE patients in China, spontaneously activated intracellular inflammatory signal cascades and induced the production of inflammatory cytokines and type I IFNs. They used THP-1 cells and the UNC93B1 overexpression system. Furthermore, they established Unc93B1 knock-in mice with orthologous mutation, and the mice were shown spontaneous systemic inflammation. These new findings are very interesting. However, the present manuscript is not sufficient to investigate the pathogenicity of the UNC93B1 mutations in SLE. Furthermore, the reviewer did not understand how to establish the knock-in mice. From the above, it is strongly required to organize a lot of data and reconsider the manuscript to be published in this journal.

The reviewer will seek corrections and reconsiderations on the following specific points.

1. In Fig. 3 c and d, the background quantity (WT expression column) of cytokine production is too high, and the induction level (WT vs mutants) is too low. Could you stimulate the cells with R848 or the other TLR7 ligands, such as Loxoribine and RNA ligands?

Thank you for this point, we have now stimulated the cells with R848 as suggested, which generates a much greater response of the cells (Manuscript New Figure 4 and Extended Data Figure 6).

2. In Fig. 3 and 4, THP-1 is a monocyte cell line. However, in SLE pathogenesis, plasmacytoid dendritic cells, conventional dendritic cells, and B cells are more important than monocytes. How about these cell lines? Can you get similar results?

To address this point, we prepared B cell lines from two UNC93B1 V117L patients. This confirms the data from Thp1 cells, with increased production of inflammatory cytokines at baseline (Manuscript New Figure 3f-g, Extended data Figure 5c-e), and after stimulation with the TLR7 ligands guanosine and R848 (Manuscript New Figure 4l-m, Extended data Figure 6e).

3. In Fig. 4e-g, they used R848 as a TLR7 ligand. But R848 could stimulate both TLR7 and TLR8. They should validate which receptor contributed to the immune responses.

Thank you very much for the comment, we have now stimulated the patient cells with the TLR8-506 specific ligand. This reveals that both the TLR7, and the TLR8 pathways are

upregulated by the UNC93B1 mutations studied (Manuscript New Figure 4d-f, Extended data Figure 6c-e).

4. In the Unc93B1 knock-in mice experiment, why did they choose the 138th amino acid for the mutation? Human and mouse UNC93B1 amino acid sequences are very similar. The percentage of homology is about 90%, and V117 is conserved between humans and mice (Fig. 1a). Moreover, the reviewer could not find V138 on an amino acid sequence of mouse Unc93B1. Is the 138th amino acid methionine? Please confirm the important point.

Thank you very much for pointing this out. After checking, the information of this gene transcript in the mouse source database has changed. In 2020 when we generated this strain, the main mouse source transcript was 619aa, and now the main mouse source transcript in the database is 598aa. Therefore, we have made the conserved mutation V117L in this strain of mice, and updated our nomenclature throughout.

5. They should show an establishment strategy of Unc93B1 knock-in mice.

We have now included the strategy we employed to generate Unc93B1 knock-in mice (Manuscript New Extended Data Figure 8).

6. In Fig. 5-7, is the lupus-like inflammation dependent on TLR7? Please analyze TLR7 deficiency in the knock-in mice.

We agree that this would be an interesting experiment, however the genetic crossing would take a very long time, beyond the scope of the current manuscript.

7. In Fig. 5f, a major producer of IFN- γ is CD4⁺ or CD8 α ⁺ T cells. Please replace it with dot graphs for T cells.

We have now updated this data, and can see that both CD4^{+ve} and CD4^{-ve} T-cells produce increased levels of IFN- γ (Manuscript New Figure 5f).

8. In Fig. 5f, how about the percentage of monocytes, macrophages, and dendritic cells in the spleen?

We found a small increase in macrophages, dendritic cells, and monocytes, in the spleen of heterozygous UNC93B1^{V117L} mice, however this was not consistent in homozygotes (Review Figure 5).

Review Figure 5. Flow cytometry analysis of indicated immune cell populations in spleen of UNC93B1^{WT/WT}, UNC93B1^{WT/V117L}, and UNC93B1^{V117L/V117L} mice (n=indicated).

9. They should measure blood urea nitrogen and serum creatinine.

Please see the response to Referee 1 above, we did not see a significant increase in blood urea nitrogen and serum creatinine for this mouse strain at the ages we tested (Review Figure 4).

10. They should show urinary albumin and creatinine ratio.

Please see the response to Referee 1 above, we did not see a significant increase in urinary albumin and creatinine ratio for this mouse strain at the ages we tested (Review Figure 4).

11. In Fig. 6b, how about depositions of immune complexes such as IgG and C3?

Please see the response to Referee 1 above (Review Figure 3). We do not see a consistent difference in immune complex deposition in this strain of mice at age 8 months. It could be the case that these symptoms take significantly longer to develop.

12. In Fig. 7b, how about the activation of IRF3 and IRF7? TLR7 could induce both IRFs phosphorylation.

In our unbiased RNA sequencing of bone marrow-derived macrophages (BMDM) we found upregulation of genes from the IRF7, IRF3, and TBK1 signaling pathway in homozygous UNC93B1^{V117L} mice (Extended Data Figure 9c). Additionally, we performed

western blots for pIRF3 and pIRF7 directly, and although pIRF3 does not appear to be increased (Review Figure 6), there is upregulation of pIRF7 in the homozygous UNC93B1^{V117L} mice (Manuscript New Figure 7b).

Review Figure 6. Level of phosphorylated IRF3 as measured by immunoblot, in lysates of BMDM of indicated mice (n=3).

Reviewer #3

(Remarks to the Author)

Title: Genetic variants in UNC93B1 predispose to childhood-onset systemic lupus erythematosus

Al-Azab et al integrated clinical and mechanistical findings to discover and explain two novel mutations directly associated with the development of childhood-onset SLE. Ploegh's lab and others have shown that Unc93b can bind and traffic toll-like receptors (TLRs) from the ER to subcellular locations for ligand sensing. Through human genetic studies (WGS), Al-Azab et al discovered two mutations in the UNC93B1 gene, a TLR7 chaperone: First, the V317A mutation in a single patient with childhood-onset SLE, and later, the V117L mutation in a set of childhood-onset SLE patients from a southeast Asian cohort. They modeled the later mutation in THP1 monocyte cell line, and in a murine model. Authors described an in vitro mechanism of action by which UNC93B1 mutant THP1 cells exhibit increased production of proinflammatory markers in response to ssDNA, a TLR7 ligand. Although the specifics of the molecular mechanism were not fully described, the authors demonstrated that increased TLR7 action maybe due to an increase of TLR7 in the endosomes, and that IRAK1/4 signaling is a key mediator of this inflammatory process. Using a murine genetic model, authors claim that a SLE-like immune-pathology can be manifested in mice.

While aspects of this study advance our understanding of UNC93B1 biology and potentially SLE, however, the study lacks clear mechanistic as well as an accurate representation of genetic/clinical features in this small cohort.

Comment to the authors:

1.- The analysis of patient cohort is limited to 3 patients and apparently all parents were normal with no SLE. UNC93B1 is on X-chromosome which should help to see whether there is a clear track of SLE disease in family.

This is an important point and we contacted all of the parents for the UNC93B1 patients, however there was no report of SLE disease in these families.

Authors do not have normal controls for several assays in Fig.1 making it difficult to compare normal versus disease differences including Fig.1a.

The data in the manuscript of Figure 2a and Extended data Figure 1a-g all now have the healthy normal range determined at our hospital highlighted in grey. We also added the details of the normal ranges in the figure legends. For the disease score (SLEDAI and SLEDAI-2k) there is no normal value.

Given the lack of disease in parents, there are clearly other genetic factors that manifest the disease which authors do not comment. The data for serum cytokines/chemokines are limited to only select patients not all patients in the cohort. It is unclear whether increased cytokines are due to this specific mutation versus ongoing fever or other clinical complications in these patients.

We certainly agree with the referee, and there could be other genetic or environmental factors that are associated with the manifestation of disease. Given that we performed whole exome sequencing, we have examined the genes that are reported as responsible for lupus, however there are no variants in those genes for the UNC93B1 patients. Otherwise, at the time of collecting blood samples, patients were stable, without clinical symptoms except renal involvement. Finally, we generated B-cell lines from the patients, which also demonstrated increased inflammatory markers (Manuscript New Figure 3f-g, 4l-m and Extended data Figure 5c-e, 6e), and provides more evidence that the phenotype is due to the specific mutation and not ongoing fever or other complications.

Authors could have used TLR7 agonist or other stimuli to directly show the sensitivity of this pathway is upregulated via these mutations.

This is an important point, and we have now used the TLR7 ligands R848 and guanosine to stimulate PBMCs, THP-1, cells and patient B-cell lines (Manuscript New Figure 4 and

Extended Data Figure 6). In this data we also found that activation with a TLR8 agonist lead to increased responses for UNC93B1 patients, but not for TLR3 and TLR9 agonists.

2.- Authors use THP1 monocytic cell line to model and to understand the mechanistic pathways involved in the role of UNC93B1 and TLR7, however, several controls are missing to ensure authors do not over-interpret this line of studies. For example, it is not clear whether mutation or level of expression is the driver of their phenotype as authors do not show any control to evaluate the expression of various mutants.

We have now carefully addressed the expression of UNC93B1 to ensure that results are not over-interpreted. Specifically, we detected the proper virus quantity by titration (Review Figure 7a), and then quantified UNC93B1-mCherry expression by fluorescence microscopy (Review Figure 7b), flow cytometry (Review Figure 7b), or flag expression by immunoblotting (Review Figure 7c).

Review Figure 7. (a) Titrations of lenti virus from 0-4×10⁷ ITU LV in THP-1 detected by fluorescence microscopy. (b) Quantification of UNC93B1-mCherry expression by imaging and flow cytometry in THP-1

infected by indicated variants of UNC93b1. (c) Levels of UNC93B1-flag as measured by immunoblot, in the lysates of THP-1 infected by indicated variants of UNC93b1.

UNC93B1 is expressed in THP1. Authors omit the fact that THP1 cell line was derived from a male patient whereas all of their patients in their cohort were female. A better cell-based system with viral transformation or utilizing CRSPR might have been a better approach to model these mutations.

In order to provide a more appropriate in vitro model, we now include data from patient B-cell lines. Data from these female cell lines with endogenous expression of mutated UNC93B1 is consistent with the Thp1 cell overexpression data (Manuscript New Figure 3f-g, 4l-m and Extended data Figure 5c-e, 6e).

3. Authors use a non-selective IRAKi at high concentration (5 uM) to suggest its utility as a therapeutic in SLE. Authors use high concentration even above IC90 with no titration of the cmpd in the current data. It is hard to conclude that their pathway is directly linked to IRAK1//4. Several IRAK4 selective small molecule inhibitors from Pfizer (PMCID: PMC8671219), Genentech (PMID: 32487715) or BMS (PMCID: PMC5253435) could be considered to ensure this pathway is TLR7/IRAK dependent. Constitutive IRAK activations could have also been directly documented by looking at pIRAK4 (see above publications). Authors could have also used CRISPR to knockout IRAK4 in their cell line system.

We thank the referee for this suggestion. Please see the new data from dose titrations of the original IRAK1/4 inhibitor we used previously, and additional data using the suggested more selective IRAK4 inhibitor (Manuscript New Figure 4 and Extended Data Figure 7). Additionally, we see upregulation of pIRAK4 directly by western blot, as recommended (Extended Data New Figure 7a).

4.- Authors over-interpretate the phenotype of the unc93b1 mutant mice. In Fig.5 a number of homozygous mutant mice have a normal weight liver, yet authors show an outlier. Authors should compare and report the functional differences in younger age 4-6 weeks old using TLR7/8 agonist stimulations in vivo or in BMDM.

We have now selected more appropriate representative images of the mouse organs. As recommended, we did perform stimulations of UNC93B1 mutant mouse BMDM with TLR7/8 agonists, and found increased inflammatory responses (Manuscript New Figure 7d, e).

Minor concerns:

1.- In suppl. Fig 1 (b,c), these images would be better interpreted/appreciated if they add a negative control staining and they quantify the magnitude of difference between background and actual signal.

Thank you for the comment, we have now included a negative control and improved images that more clearly show the mixed, speckled and cytoplasmic staining pattern (Extended Data New Figure 2). This readout is not typically quantified in a clinical setting, as negative controls have no staining.

2.- In lines 92-94, the authors claim that suppl fig 1A shows the complete blood count for P2, P3 & P4. However, the current version of that figure only shows the complement levels, the SLEDAI-2K, and the ANA antibodies. They are missing the piece of that that they are referring in this section.

Thank you for noting this, we now include the other parameters which were originally missed in the manuscript (Extended Data New Figure 1).

3.- In figs 2g-j and associate supplementary figure 2, the authors measured key lupus-associated cytokines produced by P3 and P4-derived PBMCs. Furthermore, they claim in line 97 that these cytokines were spontaneously expressed by the isolated PBMCs, and in line 100, they related some of these cytokines to TLR7-induced cytokines. Could the authors better describe the handling and treatment of these PBMCs. Particularly, did they measure cytokines right after isolation, or did they have to culture them for some time (how long)? In what media? Did they stimulate these PBMCs somehow?

We added this information to the appropriate figures legends. Specifically, PBMCs isolated from patients' whole blood and incubated in RPMI 1640 medium supplemented with 10% FBS, 2mM L-glutamine, and 1% Penicillin-Streptomycin for 12hrs without stimulation (Manuscript Figure 2 and Extended Data New Figure 3). For stimulated PBMCs, cells were stimulated by 10µg/mL HMW Poly I:C, 1µg/mL R848, 4µg/mL CPG-C, 1mM guanosine and/or 200ng/mL TL8-506 for 8hrs (P2) or for 12hrs or 24hrs (P3) in RPMI1640, supplemented with 10% FBS, 2mM L-glutamine, and 1% Penicillin-Streptomycin (Manuscript New Figure 4a-d and Extended Data New Figure 6 a-c).

4.- In line 175, the authors refer "suppl fig. 3d-f" when I think they refer to "Suppl Fig. 4d-f".

Yes, that is right, we have corrected this.

Decision Letter, first revision:

11th Mar 2024

Dear Seth,

Thank you for submitting your revised manuscript "Genetic variants in UNC93B1 predispose to childhood-onset systemic lupus erythematosus" (NI-A36518A). It has now been seen by the original referees and their comments are below. The reviewers find that the paper has improved in revision, and therefore we'll be happy in principle to publish it in Nature Immunology, pending minor revisions to satisfy the referees' final requests and to comply with our editorial and formatting guidelines.

We will now perform detailed checks on your paper and will send you a checklist detailing our editorial and formatting requirements in about a week. Please do not upload the final materials and make any revisions until you receive this additional information from us.

If you had not uploaded a Word file for the current version of the manuscript, we will need one before beginning the editing process; please email that to immunology@us.nature.com at your earliest convenience.

Thank you again for your interest in Nature Immunology Please do not hesitate to contact me if you have any questions.

Kind regards,

Laurie

Laurie A. Dempsey, Ph.D.
Senior Editor
Nature Immunology
l.dempsey@us.nature.com
ORCID: 0000-0002-3304-796X

Reviewer #1 (Remarks to the Author):

The authors have addressed the comments raised by the reviewers in a satisfactory manner.

It would be helpful for the authors to address the following points:

1. In responding to the first comment of Reviewer 1, the authors demonstrate that the UNC93B1 variants do not cause increased protein expression in B cells. They argue that the reported (Nagano paper) increased expression of UNC93B1 protein in people with active SLE is secondary. This implies that wild type UNC93B1 overexpression is involved in increased TLR7 signaling after it has been upregulated through unknown mechanisms. We do not know (the authors do not report) whether the identified variants cause increased expression in any kind of cells in the 7 reported patients.
2. S547 and S550 phosphorylation is stated to control the function of UNC93B1. If the pathogenic mutations do not alter the phosphorylation of these two Serine residues, then there must be another mechanism which is used to increase TLR7-initiated signaling. This needs, at least, to be discussed.

3. The authors report that family members were reached and they are reported not to have lupus. Have any of the family members been tested of the two variants? If they have and they are positive, it will be important to include the information. As the authors may know, DNASE1 mutations are linked to lupus but there are plenty of family members which have the same mutation but not lupus ((PMID: 15593183)). This is an important point as we need to understand the nature of what we call "monogenic" lupus.

Reviewer #2 (Remarks to the Author):

The authors have firstly reported a genetic variant in UNC93B1 that correlates with the severity of the disease in SLE patients. The fact that a genetic variant (V117L) found in humans is also shown pathogenic in mice is a strong indication that TLRs (especially TLR7) and Unc93B1 are involved in the pathogenesis of SLE. On the other hand, whether this inflammatory pathology is TLR7-dependent needs to be further tested. Mouse Unc93B1 is known to be involved in the subcellular localization and immune response of TLR3, TLR5, TLR7, TLR9, TLR11, TLR12, and TLR13, and especially TLR7 and TLR9 have been reported in spontaneous inflammatory models. The reviewer recommends that the need to crossbreed this inflammation model with TLR7-deficient mice or to experiment with the administration of inhibitory reagent should at least be mentioned in the discussion.

Reviewer #3 (Remarks to the Author):

Authors' hypothesis that UNC93B1/TLR7 interaction controls TLR/IRAK signaling is bolstered by several pieces of data consistent of genetics, molecular mechanism, pharmacological interventions and clinical data, suggesting that a potential and not far-fetched IRAK4i intervention could be available for these patients.

1.- The authors addressed concerns regarding protein expression and phosphorylation of UNC93B, showing that UNC93B1 expression and phosphorylation levels did not change between healthy and control individuals, suggesting that the mutation has a functional role more than just affecting protein expression. Additionally, they expanded their explanation of the human genetics findings as well as the murine model generation as shown in sup figs. 2 & 4, rev. figs. 1 & 5, and ext. data fig. 8.

2.- To clarify the molecular and cellular markers of disease in their model, and the relevant clinical information for patients, the authors updated figs. 3 & 4, and added new ext. data figs. 5 & 10 & rev. fig.

3.- Lastly, and potentially the most controversial but also significant findings are regarding the cellular & molecular MOA presented in this work. Based on literature, they hypothesized that UNC93B1 is a part of a signaling complex with TLR7, restricting signaling through IRAK1/4, and mutations in the interaction between UNC93B1 and TLR7 removes this controlling step, leading to enhanced IRAK1/4 signaling and disease. Authors utilized selective IRAK4i in new fig. 4 and ext. data fig. 7, suggesting a hyperactivation and phosphorylation of IRAK1/4 in response to TLR7 signaling and increased inflammatory signaling shown in several figs (Fig 3 & 5, ext. data figs. 4, 5, 6 & 7).

4- Given the no disease in patient's parents, authors may state this more clearly in the discussion and comment on complexity of SLE with respect to this mutations.

Author Rebuttal, first revision:

Genetic variants in UNC93B1 predispose to childhood-onset systemic lupus erythematosus

Response to Reviewers Comments

Reviewer #1 (Remarks to the Author):

The authors have addressed the comments raised by the reviewers in a satisfactory manner.

It would be helpful for the authors to address the following points:

1. In responding to the first comment of Reviewer 1, the authors demonstrate that the UNC93B1 variants do not cause increased protein expression in B cells. They argue that the reported (Nagano paper) increased expression of UNC93B1 protein in people with active SLE is secondary. This implies that wild type UNC93B1 overexpression is involved in increased TLR7 signaling after it has been upregulated through unknown mechanisms. We do not know (the authors do not report) whether the identified variants cause increased expression in any kind of cells in the 7 reported patients.

Thank you for the comment. The only available cells from patients are B cells from P2 and P3, for which we reported normal levels of UNC93B1 protein expression (Review Figure 1).

Review Figure 1. (a) Quantitative RT-PCR analysis of UNC93B1 expression in the indicated immortalized B cell lines. (b) Levels of UNC93B1 as measured by immunoblot, in the lysates of the immortalized B-cells from UNC93B1 patients, P2 and P3, and healthy control. Data are representative of three independent experiments.

2. S547 and S550 phosphorylation is stated to control the function of UNC93B1. If the pathogenic mutations do not alter the phosphorylation of these two Serine residues, then there must be another mechanism which is used to increase TLR7-initiated signaling. This needs, at least, to be discussed.

We discussed this as recommended, thanks.

3. The authors report that family members were reached and they are reported not to have lupus. Have any of the family members been tested of the two variants? If they have and they are positive, it will be important to include the information. As the authors may know, DNASE1 mutations are linked to lupus but there are plenty of family members which have the same mutation but not lupus ((PMID: 15593183)). This is an important point as we need to understand the nature of what we call “monogenic” lupus.

We did WES for family of P2 and P3. We found that father of P2 and mother of P3 are healthy carriers of the heterozygous V117L variant, and included this data in the manuscript.

Reviewer #2 (Remarks to the Author):

The authors have firstly reported a genetic variant in UNC93B1 that correlates with the severity of the disease in SLE patients. The fact that a genetic variant (V117L) found in humans is also shown pathogenic in mice is a strong indication that TLRs (especially TLR7) and Unc93B1 are involved in the pathogenesis of SLE. On the other hand, whether this inflammatory pathology is TLR7-dependent needs to be further tested. Mouse Unc93B1 is known to be involved in the subcellular localization and immune response of TLR3, TLR5, TLR7, TLR9, TLR11, TLR12, and TLR13, and especially TLR7 and TLR9 have been reported in spontaneous inflammatory models. The reviewer recommends that the need to crossbreed this inflammation model with TLR7-deficient mice or to experiment with the administration of inhibitory reagent should at least be mentioned in the discussion.

Thanks for nice comment, we mentioned this in discussion section.

Reviewer #3 (Remarks to the Author):

Authors' hypothesis that UNC93B1/TLR7 interaction controls TLR/IRAK signaling is bolstered by several pieces of data consistent of genetics, molecular mechanism, pharmacological interventions and clinical data, suggesting that a potential and not far-fetched IRAK4i intervention could be available for these patients.

1.- The authors addressed concerns regarding protein expression and phosphorylation of UNC93B, showing that UNC93B1 expression and phosphorylation levels did not change between healthy and control individuals, suggesting that the mutation has a functional role more than just affecting protein expression. Additionally, they expanded their explanation of the human genetics findings as well as the murine model generation as shown in sup figs. 2 & 4, rev. figs. 1 & 5, and ext. data fig. 8.

2.- To clarify the molecular and cellular markers of disease in their model, and the relevant clinical information for patients, the authors updated figs. 3 & 4, and added new ext. data figs. 5 & 10 & rev. fig.

3.- Lastly, and potentially the most controversial but also significant findings are regarding the cellular & molecular MOA presented in this work. Based on literature, they hypothesized that UNC93B1 is a part of a signaling complex with TLR7, restricting signaling through IRAK1/4, and mutations in the interaction between UNC93B1 and TLR7 removes this controlling step, leading to enhanced IRAK1/4 signaling and disease. Authors utilized selective IRAK4i in new fig. 4 and ext. data fig. 7, suggesting a hyperactivation and phosphorylation of IRAK1/4 in response to TLR7 signaling and increased inflammatory signaling shown in several figs (Fig 3 & 5, ext. data figs. 4, 5, 6 & 7).

4- Given the no disease in patient's parents, authors may state this more clearly in the discussion and comment on complexity of SLE with respect to this mutations.

Thanks for the comment, we made it clear as you can check at the discussion section.

Final Decision Letter:

Dear Seth,

I am delighted to accept your manuscript entitled "Genetic variants in UNC93B1 predispose to childhood-onset systemic lupus erythematosus" for publication in an upcoming issue of *Nature Immunology*.

Over the next few weeks, your paper will be copyedited to ensure that it conforms to *Nature Immunology* style. Once your paper is typeset, you will receive an email with a link to choose the appropriate publishing options for your paper and our Author Services team will be in touch regarding any additional information that may be required.

Please note that *Nature Immunology* is a Transformative Journal (TJ). Authors may publish their research with us through the traditional subscription access route or make their paper immediately open access through payment of an article-processing charge (APC). Authors will not be required to make a final decision about access to their article until it has been accepted. Find out more about Transformative Journals.

Your paper will be published online soon after we receive your corrections and will appear in print in

the next available issue.

Also, if you have any spectacular or outstanding figures or graphics associated with your manuscript - though not necessarily included with your submission - we'd be delighted to consider them as candidates for our cover. Simply send an electronic version (accompanied by a hard copy) to us with a possible cover caption enclosed.

If you have not already done so, we strongly recommend that you upload the step-by-step protocols used in this manuscript to the Protocol Exchange. Protocol Exchange is an open online resource that allows researchers to share their detailed experimental know-how. All uploaded protocols are made freely available, assigned DOIs for ease of citation and fully searchable through nature.com. Protocols can be linked to any publications in which they are used and will be linked to from your article. You can also establish a dedicated page to collect all your lab Protocols. By uploading your Protocols to Protocol Exchange, you are enabling researchers to more readily reproduce or adapt the methodology you use, as well as increasing the visibility of your protocols and papers. Upload your Protocols at www.nature.com/protocolexchange/. Further information can be found at www.nature.com/protocolexchange/about .

Please note that we encourage the authors to self-archive their manuscript (the accepted version before copy editing) in their institutional repository, and in their funders' archives, six months after publication. Nature Portfolio recognizes the efforts of funding bodies to increase access of the research they fund, and strongly encourages authors to participate in such efforts. For information about our editorial policy, including license agreement and author copyright, please visit www.nature.com/ni/about/ed_policies/index.html

Kind regards,

Laurie

Laurie A. Dempsey, Ph.D.
Senior Editor
Nature Immunology
l.dempsey@us.nature.com
ORCID: 0000-0002-3304-796X